# Controlling photothermoelectric directional photocurrents in graphene with over 400 GHz bandwidth

Stefan M. Koepfli [1]✉, Michael Baumann[1], Robin Gadola [1], Shadi Nashashibi[1], Yesim Koyaz [1,2], Daniel Rieben[1], Arif Can Güngör [1], Michael Doderer[1], Killian Keller [1], Yuriy Fedoryshyn[1] & Juerg Leuthold [1]✉

Photodetection in the near- and mid-infrared spectrum requires a suitable absorbing material able to meet the respective targets while ideally being cost-effective. Graphene, with its extraordinary optoelectronic properties, could provide a material basis simultaneously serving both regimes. The zero-band gap offers almost wavelength independent absorption which lead to photo-detectors operating in the infrared spectrum. However, to keep noise low, a detection mechanism with fast and zero bias operation would be needed. Here, we show a self-powered graphene photodetector with a > 400 GHz frequency response. The device combines a metamaterial perfect absorber architecture with graphene, where asymmetric resonators induce photo-thermoelectric directional photocurrents within the graphene channel. A quasi-instantaneous response linked to the photothermoelectric effect is found. Typical drift/diffusion times optimization are not needed for a high-speed response. Our results demonstrate that these photothermoelectric directional photocurrents have the potential to outperform the bandwidth of many other graphene photodetectors and most conventional technologies.

Efficient and fast photodetection of infrared light has an ever-increasing demand. In the near-infrared (NIR) and the telecom window, detectors target a high response while being able to follow frequencies above 100 GHz. In the mid-infrared (MIR) on the other hand, the PDs need to be able to resolve the smallest optical power levels without the signal drowning in noise coming from the materials' low bandgap. Ideally, one would wish for a material system that does not require expensive fabrication (epitaxial crystalline growth, dedicated substrate, flip chip, …) and could serve both the NIR and MIR spectral ranges—a trend also being driven by the need for expanding the optical communication bands[1].

Graphene with its zero bandgap[2,3] acting as a spectrally universal absorber[4,5] could allow to serve both regimes while becoming cost-efficient[6] and potentially co-integratable[7] due to its two-dimensional nature. While some challenges such as scalability, stability and effective passivation remain to this point[8], the hurdle to efficiently couple light to the single layer has been approached by a myriad of different approaches: Gain-medium assisted approaches have been able to reach record high responsivities[9], have demonstrated scalability[10] and were successfully demonstrated to work simultaneously in the NIR and MIR regime[11–13]. However, these systems rely on the spectral absorption characteristics of the gain medium and do not make use of the direct absorption across a large spectral range in graphene. Therefore, they're usually limited to the low-frequency characteristics of the gain medium and the resulting responses in the kHz regime is unsuitable for telecommunication. To exploit the fast carrier dynamics of graphene, efficient direct absorption is needed. By using photonic integrated circuits (PIC) such as silicon[14–16], silicon nitride[17] or even chalcogenide glass[18] one may enable good coupling to graphene. The viability of these approaches in the telecom regime has

[1]ETH Zurich, Institute of Electromagnetic Fields (IEF), Zurich, Switzerland. [2]Present address: EPFL, Photonic Systems Laboratory (PHOSL), Lausanne, Switzerland. ✉e-mail: stefan.koepfli@ief.ee.ethz.ch; juerg.leuthold@ief.ee.ethz.ch

been demonstrated with data transmission experiments[19-25] and even bandwidths in excess of 100 GHz[20,21,26]. However, most of these approaches suffer from early optical power saturation onsets due to a high confinement onto a small active area and restrict the spectral window due to the material basis and PIC elements such as couplers and splitters. Direct illumination of graphene from free space could solve the power saturation issue by distribution of the power, overcome the limited spectral window by leveraging the zero bandgap, and remove the need for dedicated materials. And indeed, a variety of demonstrations achieved photodetection in the NIR[27-29], MIR[30-33], or both regimes[34-36]. The carrier extraction and low absorption of graphene can be tackled by deploying special electrode designs[35,37] and adding resonators or metamaterials[30,33,38]. While the zero bandgap enables broadband operation, the semi-metallic nature is generally also a hurdle to achieve low-noise operation.

Interestingly, different performance metrics can be achieved on graphene by targeting specific operation conditions, as the direct absorption of graphene can lead to different detection mechanisms, mainly[39]. photovoltaic (PV), photoconductive (PC), bolometric (BOL), and photothermoelectric (PTE). As these effects can typically exist simultaneously—and even counteract each other—the design of graphene PDs requires more careful considerations than simply optimizing for the highest absorption in graphene. For instance, when operating a graphene detector at zero bias and exploiting the PV and PTE effects, one may tune out the PC and BOL effects that usually come along with shot noise from the conductive channel. Considering that the PV is localized to the driving fields in the vicinity of the electrodes[39,40], the PTE effect extends over longer distances as it follows the temperature gradient[38]. This may be beneficial in the design of high-saturation power detectors. In this context, asymmetries introduced by directly etching the graphene channel[41] or by including asymmetric resonators[42] have led to zero-bias PDs partially associated to the PTE effect. Especially the latter case with the metallic resonators has not only successfully demonstrated uncooled low-noise equivalent powers (NEP) at 4 μm[42] but was also shown to be useful as linear polarimeter[43] and circular polarimeter[44]. The underlying PTE mechanism was in said papers referred to as directional photocurrent, which reflects the clever arrangement of the PTE elements that

provides a net photocurrent in one direction. And indeed, it could potentially offer a platform for next-generation broadband, high-speed graphene-based detectors[45]. However, the performance of these directional PTE currents in terms of optimized frequency response and the interaction with other effects is not yet understood.

Here in this work, we investigate the photothermoelectric-induced directional currents (PTE-DC) within a metamaterial perfect absorber (MPA) architecture for its viability to serve as infrared photodetector and thereby uncover its quasi-instantaneous frequency response. Firstly, we provide a scheme to incorporate asymmetric resonators into a passivated MPA architecture, reaching more than 80% absorption and by that boost the interaction of the light with the graphene. This metamaterial offers a broad response able to cover several communication bands at once and is easily tunable from the NIR to MIR regime. Secondly, we identify how the PTE-DC interacts with the PC and BOL effects revealing that the effects add up. Thirdly, by testing the frequency response, we find that the PTE-DC in our devices offers a 3 dB bandwidth of 420 GHz—not only outperforming the other effects present in the device, but also almost all other graphene-based PDs and even conventional technologies. The PTE-associated response thereby shows a quasi-instantaneous characteristic as it does not rely on typical drift/diffusion limit optimization. Lastly, we demonstrate the flexibility of the PTE-DC by combining a supercell MPA architecture with a four-electrode design enabling broader spectral absorption, as well as wavelength and polarization multiplexing. We therefore provide insights into the engineering process for such a metamaterial-graphene architecture PD by characterization and multiphysics simulations, and thereby show its potential for next-generation highest-speed infrared PDs.

## Results
### Metamaterial perfect absorber concept
Our graphene metamaterial photodetector combines a metamaterial perfect absorber (MPA) structure with asymmetric resonators and graphene. A visualization of the metamaterial device and its structure is given in Fig. 1a, b. The architecture targets a direct illumination from the top, e.g., by a single-mode fiber. The device itself consists of a gold back reflector, an aluminum oxide spacer layer, a monolayer of CVD-

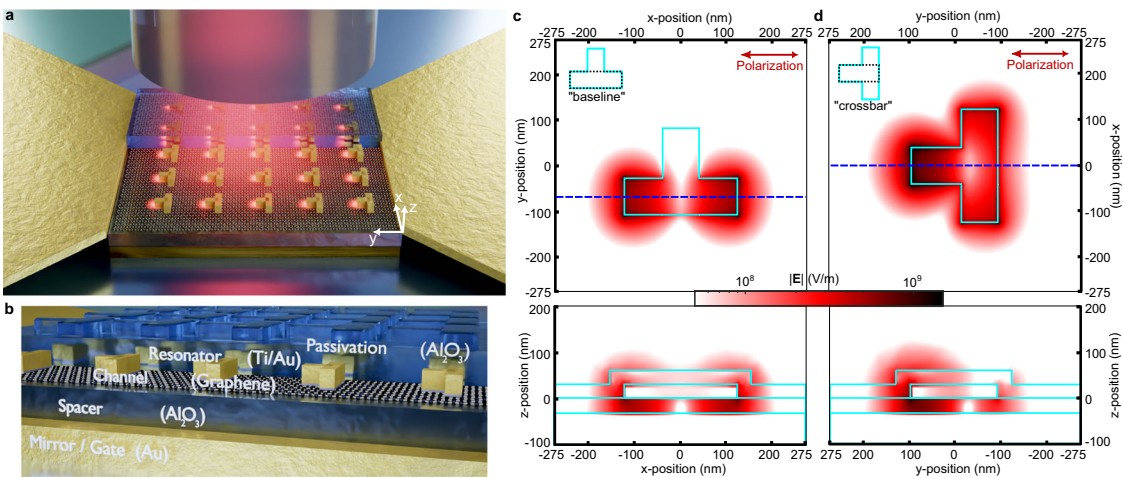

**Fig. 1 | Metamaterial perfect absorber graphene photodetector concept.**
**a** Artistic visualization of the metamaterial-induced photothermoelectric directional photocurrent device. Asymmetric resonators lead to hot spots that induce a net-current along the channel contacted by source-drain pads. **b** Cross-section showing the metamaterial which consists of a metal, insulator, graphene, metal resonator, and insulator layer stack that forms an impedance-matched perfect absorber. **c, d** Simulated absolute electric field depicting the top view and cross-section along the blue dashed line of one unit cell of the metamaterial. The

structure is excited with a linearly polarized plane wave with an orientation as indicated by the red arrow in the top right corner. **c** The baseline-resonator acts as a dipole antenna, as can be seen by the typical field pattern. The cross-sectional view further shows that the field hotspot is around the graphene layer at $z = 0$ nm.
**d** Simulation results for the same unit cell but rotated 90° with respect to (**c**). The crossbar shows a single hotspot at the tip, with weaker fields surrounding the whole structure. The cross-section again reveals the highest field intensity close to the graphene layer at $z = 0$ nm.

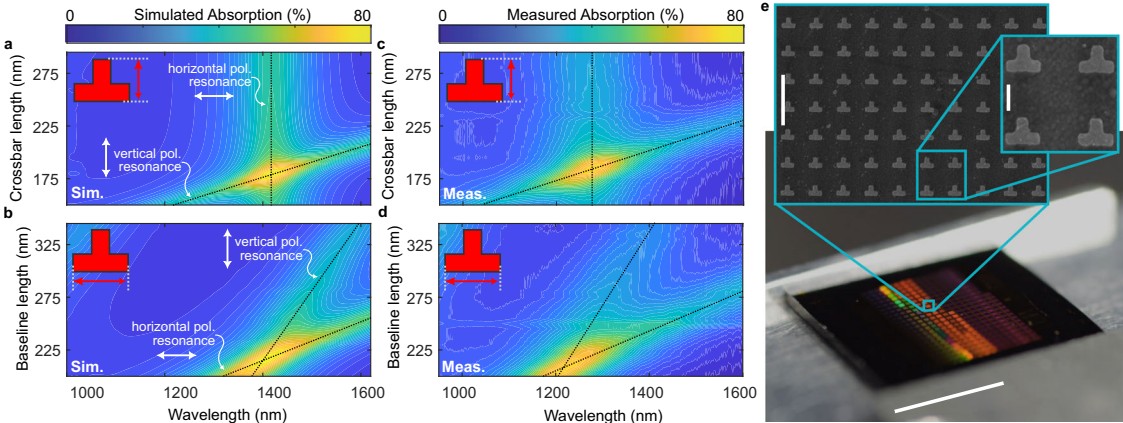

**Fig. 2 | Metamaterial optical simulation and characterization. a, b** Simulations showing the strength of absorption of unpolarized light for structures as a function of **a** crossbar- and **b** baseline length versus wavelength. Changing the geometry versus wavelength reveals two absorption bands. One related to the vertically polarized light and the other to horizontally polarized light. By tuning the length of either the baseline or the crossbar length, a point with maximum absorption for unpolarized light with an 80% absorption efficiency can be found. Such a point for light across a large spectral range. Here, the baseline is 218 nm long in (**a**), whereas the crossbar is 178 nm in (**b**). **c, d** Corresponding measured absorption values of the simulated structures. The same length scaling is extracted validating the simulated tunability of the structure. **e** View on the fabricated spectroscopy sample where each square corresponds to one metamaterial structure with varying designs and parameters (white scalebar 5 mm). The zoom-ins show scanning electron micrographs of an example structure (white scale bars 1 μm and 200 nm, respectively).

grown graphene on which the T-shaped resonators are positioned. The full structure is encapsulated by another aluminum oxide layer serving as a passivation. The "inverse T"-shaped resonators (⊥) are inspired from the previous demonstration of Wei et al.[42] and scaled to operate in the NIR. They can be viewed as an asymmetric modification of a classical cross resonator often used for metamaterial perfect absorbers (MPA)[46,47]. To visualize their electric field response, we simulate such an MPA unit cell optimized for NIR illumination by exciting it with a linearly polarized plane wave. Figure 1c shows the absolute electric field in the graphene layer. A clear dipole-like pattern as typically seen by these types of bar- or cross-resonators is observed. We call the alignment of the resonators such as shown in Fig. 1c the "baseline orientation" of the ⊥-resonator. The cross-section along the blue dashed line is given in the bottom panel, showing that the majority of the field is situated close to the graphene layer at z = 0 nm. If we rotate the structure by 90° while keeping the polarization orientation, we find the resonance for the short arm of the ⊥-structure, which we refer to as "crossbar orientation", see Fig. 1d. Again, a clear hotspot can be observed, and the electric field is strongest close to the tip, further visualized in the cross-section given in the bottom panel.

To find the spectral resonant enhancement as a function of the geometry we further simulated the optical absorption spectrum of the structure by sweeping the different dimensions of the layer stack and the resonator. The resulting absorption for unpolarized light as a function of crossbar or baseline length versus wavelength is given in Fig. 2a, b. One can observe two resonance lines that change when varying the lengths of (a) the crossbar or (b) baseline features. Polarization-resolved simulations (see Supplementary Fig. 9) show that they are related to either the vertically or horizontally aligned polarization of the light. The results in Fig. 2a show that by tuning the length of the crossbar one can tune the resonance peak position associated with the vertical polarization without influencing the resonance of the horizontal polarization. Figure 2b on the other hand shows that by tuning the baseline length both resonances shift. In practice, the plots give us a recipe to find the asymmetric geometry for a polarization-insensitive absorber. For this, one uses Fig. 2b to set the horizontal polarization resonance to the target peak position by changing the baseline length. With Fig. 2a, one is able to move the vertical polarization resonance by changing the crossbar length without shifting the horizontal polarization resonance, thereby enabling to match the peak positions of the resonances. The absorption for both

polarizations thereby adds up to a maximum of 80% for unpolarized light.

To verify our simulation results, we fabricated the corresponding metamaterial structures. A picture and example scanning electron micrographs of the fabricated sample are provided in Fig. 2e. Each brighter, colorful square on the sample corresponds to a different metamaterial design. Figure 2c, d shows the absorption spectroscopy measurement results (see "Methods" for more details). A peak total absorption in the order of 80% is found. This is close to the 80% as predicted by the simulation in Fig. 2a, b. Furthermore, we observe the same scaling behavior with changes in the crossbar (Fig. 2c) and the baseline (Fig. 2d) lengths as indicated by the black dotted lines.

Notably, there is a blue shift between the simulations and the measurement results. We attribute this to the rounded corners and size deviations resulting from the nanofabrication. Also, for the current geometry, the absorption is limited to 80%. Reaching perfect absorption geometry might be possible though by applying further changes to the layer stack and the geometry. A more detailed discussion on the metamaterial design and the corresponding trade-offs is given in Supplementary Note 4.

### Photoresponse as a function of gate voltage and polarization
We next verify the device for its efficiency as a photodetector. For this purpose, we fabricate active devices with electrical contacts. First, we define a coordinate system and define the angular alignment. Figure 3a shows a schematic of the resonator arrangement with respect to the electrodes. The ⊥-shaped resonators are aligned with the baseline pointing along the channel defined by source-drain, as also schematically depicted in the polar plot where we define the polarization angle. We define the drain as the ground terminal. The resulting current convention is given by the arrows below: electrons moving in positive (negative) $x$-direction result in negative (positive) photocurrents $I_{ph}$. We use dark red and dark blue colors for negative and positive photoresponses, respectively, throughout the paper.

To check the graphene's properties and the doping regime, we first sweep the gate voltage. For this, we use the back reflector of the metamaterial as a gate electrode. Figure 3b shows the device resistance as a function of gate voltage $V_G$. We observe the Dirac point close to 0 V ($V_D \approx 0.6 V$). We use this value to offset the gate voltage axis accordingly, i.e., $\overline{V_G} = V_G - V_D$. Furthermore, the curve depicts a back-and-forth sweep of the gate voltage, indicating a small hysteresis.

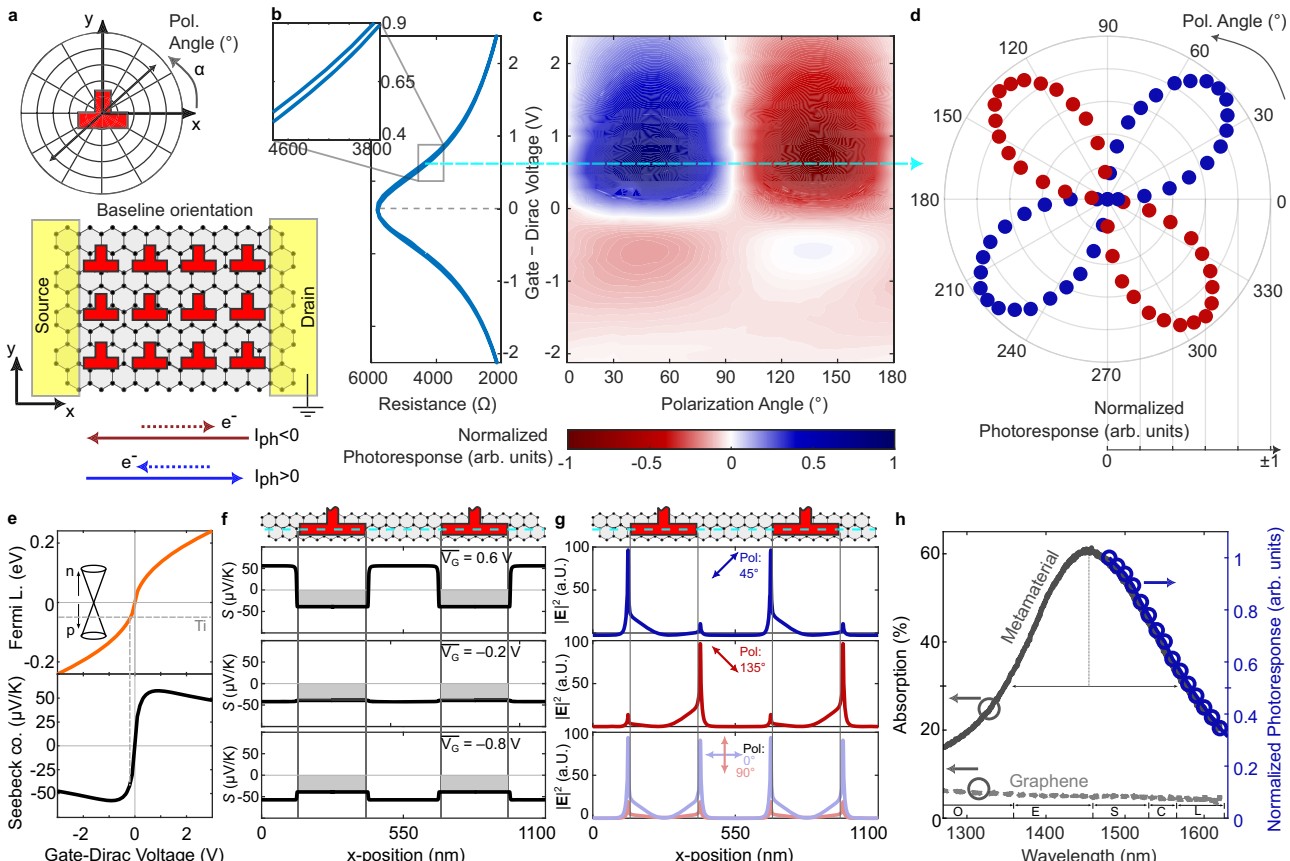

**Fig. 3 | Performance evaluation of active metamaterial-induced PTE-DC device.**
**a** Schematic of the resonator orientation with respect to the source-drain contact and definition of the polarization angle. The dark red and blue arrows represent electron flow directions (dotted) and photocurrent sign convention (solid).
**b** Resistance of the device as a function of gate voltage (back- and forth sweep). The Dirac point is found close at 0.6 V, showing low doping and making the Dirac point accessible with sub-Volt gating. The gate voltage axis has been offset by $V_D = -0.6\,V$ such that the Dirac point is centered at 0 V, i.e., the axis shows $\overline{V_G} = V_G - V_D$. The blowup shows the small hysteresis between back- and forth sweep. **c** Normalized photoresponse as function of gate voltage and polarization. The maximum responses are at 45° and 135°. When changing the gate voltage from positive to negative the photoresponse switches sign. The ideal operation point is at $\overline{V_G} = 0.6\,V$, slightly above the Dirac point (cyan dashed line). **d** Normalized photocurrent of the ideal operation point as function of the polarization angle in polar coordinates. The color indicates the sign of the current. **e** Simulated graphene channel Fermi level and calculated Seebeck coefficient as function of gate voltage. The dashed line

labeled with Ti marks the influence of the contact metal. **f** Seebeck coefficient along the channel in an area with resonators as schematically depicted. The panels correspond to a device gate minus Dirac voltage of 0.6 V, −0.2 V, and −0.8 V. The resonators dope the graphene directly underneath and the local surrounding influencing the Seebeck coefficient. **g** Simulated absolute electric field intensity for the same channel region as in (**f**). The three panels correspond to three different polarization cases leading to (i) maximum positive (45°), (ii) maximum negative (135°) and (iii) minimum (0° and 90°) photoresponse. The electric field intensities correspond to the optical-induced hot spots. The hot spots are largest where the Seebeck coefficients have the largest slope, leading to a maximum in the driving field of the current. **h** Measured optical absorption (left axis) once for graphene with metamaterial and once without. The plot of the photocurrent normalized to absorption (right axis) as a function of wavelength shows how the active photoresponse follows the same trend as the total metamaterial absorption. The measured absorption shows a FWHM of ~200 nm covering several optical communication bands.

The inset with the blowup in the range of 0.4 V and 0.9 V further visualizes this behavior. We ascribe the stable and controlled gating in ambient environment to the aluminum oxide passivation.

To test the photoresponse, we directly illuminate the device from top with a standard single-mode fiber (SMF). Further, to find the optimum working point we control the polarization of the illumination with a polarization synthesizer and simultaneously sweep the gate voltage (see "Methods"). The normalized photoresponse as a function of polarization and gate voltage is given in Fig. 3c. We find the strongest response at 0.6 V away from the Dirac point and observe the previously reported sign switch[42] in the photoresponse showing a directionality of the photocurrent. We further visualize this behavior of the photoresponse in the ideal working point by a polar coordinate plot (Fig. 3d) showing a clear four-lobe pattern with the sign switch. The dark red and dark blue colors correspond to negative and positive photocurrents, respectively.

To understand the photoresponse of Fig. 3c. as a function of polarization and applied gate voltage, we model the driving force $F$ of the PTE-DC by Wei et al.[43],

$$\vec{F}(x,y) \propto \nabla S(x,y) \cdot |\mathbf{E}(x,y)|^2. \tag{1}$$

The electric field response $|E(x,y)|$ is already available from the above discussion (Fig. 1c, d). The Seebeck coefficient $S(x,y)$ is obtained from Mott's formula[48]. We require the carrier mobility, residual carrier concentration and Fermi level as a function of gate voltage. From the gate curve (Fig. 3b), we extract a carrier mobility of $\mu_0 \sim 900\,cm^2/Vs$ and a residual carrier concentration $n_0 \sim 5*10^{11}\,cm^{-2}$ (see Supplementary Fig. 2). Next, we simulate the electrostatic potential in the graphene channel in a region that is unperturbed by the resonators (see Supplementary Note 1). Figure 3e shows the simulated resulting Fermi level as a function of gate voltage (top panel). With the extracted

parameters, we now use Mott's formula to calculate the Seebeck coefficient as a function of gate voltage (Fig. 3d, bottom panel).

To link the force (Eq. (1)) to the measurement data of Fig. 3c, we simulate the potential along the graphene channel for three gate voltages $\overline{V_G} = V_G - V_D = [0.6, -0.2, -0.8]V$. Assuming a contact doping of $-0.05$ eV for Ti on graphene[40], we arrive at the three profiles depicted in Fig. 3f. The center panel explains the zero response at $\overline{V_G} = -0.2V$. The Seebeck coefficient is almost flat along the whole channel so that there is no Seebeck coefficient gradient present minimizing the carrier driving force term. Comparing the top ($\overline{V_G} = 0.6V$) and bottom panel ($\overline{V_G} = -0.8V$), we find a sign switch in the Seebeck coefficient $S$ which explains the photoresponse sign switch at the two gate voltages for the same polarization. We also find the reason for the non-equal amplitude in the photoresponse for the positive and negative gate voltages, as the Seebeck coefficient reaches a saturation value and drops off, limiting the gradient due to the fixed Ti contact doping level (gray dashed line Fig. 3e) and thereby lowering the resulting force.

Next, we plot the simulated absolute electric field squared as a function of channel position in Fig. 3g. An illumination with 45° polarized light (top panel) leads to an asymmetric electric field profile. On the other hand, illumination with 135° polarized light (center panel) creates a mirror image field response. Lastly, the two cases of 90° and 0° polarized light (bottom panel) induces symmetrically either no strong electric field or a strong one. Even for the strong electric field response, there is no induced PTE-DC due to the symmetry.

The thermoelectric directional response as a function of polarization and the applied gate voltage from Fig. 3c can now be understood. For this the Seebeck gradients from plot 3 f and the intensities for the respective gate and polarizations need to be multiplied as per the formula above. It can now be understood that the PTE-DC switches its sign when the polarization rotates.

Lastly, we test the efficiency of the metamaterial absorber enhancement on the photoresponse. For this we first measure the absorption of the MPA, Fig. 3h. The measured absorption for unpolarized light (i.e., corresponding to the 45° linear polarization) reaches above 60% at 1460 nm with a full-width half maximum of 200 nm. As a reference we measured a monolayer graphene with the same layer stack but without any resonators (labeled dashed line). An almost flat weak absorption of ~5% is observed, as one would expect from theory and previous reports[4]. Next, we verify that the resonant enhancement indeed leads to a higher photocurrent. Towards this goal we measure the normalized photoresponse from 1480 to 1630 nm (blue line, right axis). It can be seen that the photocurrent follows the same trend as the absorption, i.e., if the absorption is halved, so is the photoresponse. This indicates that the measured total absorption is linked to the absorption in graphene and that the PTE-DC can indeed be enhanced with a MPA stack. The responsivity is found to be -1.5 V/W and photovoltage signals >18 mV are extractable with the high optical power tolerance, see Supplementary Note 8 for more details.

## Photoresponse as function of gate and bias voltage

To further understand how the polarization-dependent photoresponse from a polarization-independent absorption arises, we discuss the crossbar-oriented device in Fig. 4. The resonators are rotated by 90° with respect to the ones of Fig. 3, i.e., they are now in the crossbar orientation. The measured normalized photoresponse as function of polarization is shown in the polar plot in Fig. 4a. Two particular things can be noted here. First, rotating the resonators by 90° is not equivalent to a polarization rotation of 90°. This is evident from the lobe orientation (0°, 90°) in comparison to Fig. 3d (45°, 135°). Second, if one follows the cross-sectional force model introduced in the previous section (see Supplementary Note 2), the model will predict zero response for an incident polarization of 90°. However, a clear

response at 90° is also present. The intuitive model introduced above thus needs to be extended.

To understand and capture this PTE-DC response, we expand the above force model by using a hydrodynamic flow model[43]. Here, the vectorial force given in Eq. (1) is used as a source term and the 2D potential landscape is also accounted for, which allows to capture the directional carrier flow phenomena (see Supplementary Note 3 for more details). We then assume operation at the ideal gating point (as identified above, see teal line in Fig. 3c) and pick three distinct polarization points indicated by (1), (2) and (3) in Fig. 4a to evaluate the model. The points have been picked to be (1) close to the maximum positive photoresponse, (2) close to zero photoresponse and (3) close to the maximum negative photoresponse. Figure 4b–d shows simulations of the optical E-field resonantly enhanced at the metamaterial resonators for the three polarizations (heatmap). From these fields, the hydrodynamic simulations reveal the induced electron currents. The electron flow trajectories have then been overlaid in the same plots (black lines). In case (1), the electrons are predominantly flowing towards the source contact, whereas in case (3) the electrons have an overall movement towards the drain contact. In case (2), a current is induced but the resulting current is directed perpendicularly to the channel where no collection electrodes can capture the charges. Thereby, the charges are lost. The resulting direction of the PTE-DC $I_{PTE-DC}$ (blue and red arrows) for a given polarization orientation (labeled black double arrows) are schematically depicted in the polar plots below the electron flow simulations. The measured current is ultimately a projected current $I_{Projected}$ along the source-drain channel direction, as schematically depicted below the schematic polar plots. Thus, the hydrodynamic flow model indeed provides net currents that could not be explained by the simpler model outlined in the previous section.

Beyond the control of polarization and gate voltage, we are further able to apply a bias voltage. By doing so, we get access to additional photodetection mechanisms, namely the bolometric (BOL) and photoconductive (PC) effects. Both could offer beneficial performance if they're combinable with the PTE-DC. The measured photoresponses for the three polarization points (1)–(3) as a function of gate and bias voltage are given in Fig. 4e–g. In these plots, we also labeled the dominant physical effects within an operation area. If operated with the polarization of point (2) one finds the well-known graphene response pattern[39] (Fig. 4f). The device response is dominated by the BOL effect at high doping levels (large negative gate voltages). The BOL effect provides an inverse photoresponse with respect to the source-drain bias voltage. When approaching the Dirac point (which for this device is at $V_G \approx 0.2V$) the sign of the photoresponse switches as the response is dominated by the PC effect. Another switch to the BOL operation is observed for large positive gate voltages with some perturbation from the non-zero contribution of the PTE-DC in the gate voltage range where the response is strongest (~1–2 V, see also Fig. 3c). If we now select the polarization of point (1), we find a strong photoresponse in said operation range (Fig. 4e, blue color). The strong PTE-DC response adds to the BOL response which leads to a suppression of the BOL effect for positive bias voltages (top right corner), whereas an additive effect is observed for negative bias voltages (top left corner). By simultaneously controlling bias and gate voltage, we can therefore improve the response of the device by a factor 2 when choosing an operation point where the PTE-DC and BOL response add up. A factor 1.4 is obtained when combining the PTE-DC with the PC response. Operation with a polarization chosen as in point (3) changes the sign of the PTE-DC, adding a negative photoresponse in the previous range (Fig. 4g, red region). Now the negative response can again be combined with the PC and BOL response effectively leading to a mirrored behavior of the interesting operation points in comparison to case (1). These results indicate that the different detection mechanisms in graphene are additive. The possibility to control the polarization in a

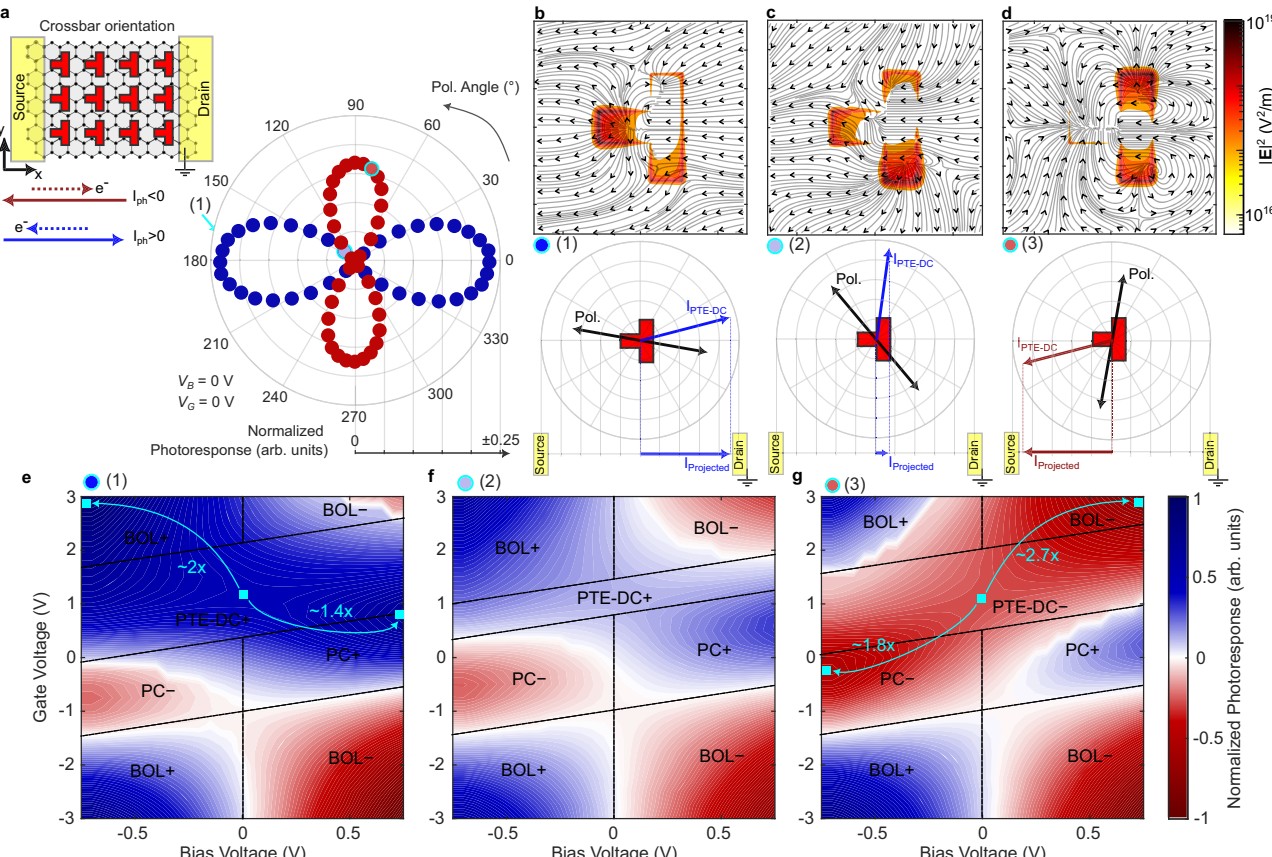

**Fig. 4 | Influence of bias and gate voltage on photoresponse. a** Normalized photoresponse as a function of polarization angle. The inset shows the resonator orientation in the device. Three distinct points are picked: (1), (2), (3) for the following discussion. **b–d** Simulated electric field response and resulting electron flow within a unit cell that correspond to the three polarization points selected in (**a**). **b–d** Correspond to points (1), (2), (3). The resulting direction of the electrons are given by simulated streamlines. A schematic depiction of the resonator orientation, polarization and induced current as well as projected current is given in the corresponding panels below the simulation results. **e–g** Measured normalized photoresponse as a function of gate and bias voltages. Depending on the bias and gate voltages different effects dominate. The dominant effects are labeled. **e** Shows operation in pol. point (1). A large positive photothermoelectric directional current

(PTE-DC + ) response for gate voltages between 0 to 2 V is found. The zero-bias PTE-DC response can be improved by a factor of 2 when negatively biasing the device and combining it with the positive bolometeric (BOL + ) effect and a factor of 1.4 if positively biased and combined with the positive photoconductive (PC + ) effect, indicated by the cyan curved arrows. **f** Operation with polarization state of point (2), where the PTE-DC is almost 0. Here, typical graphene photodetector (PD) effects such as the BOL + , BOL− or PC + , PC− dominate depending on the applied voltages. **g** Operation in point (3) showing a large negative photothermoelectric directional currents (PTE-DC − ) region. By biasing the graphene PD, one can again improve the response by adding the sign-flipped BOL and PC effects, namely the negative bolometric (BOL − ) or negative photoconductive (PC − ) response to the PTE-DC − .

---

manner that does not change the overall absorption in the graphene but only where the absorption spatially is happening allows to turn on and off mechanisms making them clearly distinguishable. By choosing the contact metal[49] of the resonators appropriately and control of the gate voltage, one could further optimize the combination of the detection mechanisms and the peak operation.

### High-speed frequency response of detection mechanisms

Quite a few graphene detection mechanisms have already been demonstrated to reach high-speed operation. No experimental verification showing high speed exists though for the PTE-DC effect. First reports of the PTE-DC frequency response were limited to the low kHz regime, but frequency responses exceeding 100 GHz have been predicted[42]. Later on, pulsed measurements showed time constants in the order of several hundred picoseconds[43,44], clearly falling short of the high bandwidths demonstrated by using other detection effects in graphene.

Here, we trim a device for high speed. We have used RF simulations to find an architecture that does not limit the bandwidth of the device but ideally reveals potential speed limitations of the effect (see also Supplementary Note 9). To test the device for the high-speed

frequency response, we use a laser beating scheme (see Fig. 5b and "Methods"). This allows us to measure our graphene PD in the range of 3–500 GHz. The results of the frequency response measurements in a fully passive operation is given in Fig. 5a. The colors of the dots indicate measurements with different setups. So, for instance, for frequencies above 110 GHz different subharmonic mixers were used as indicated in the setup schematic (Fig. 5b). A flat frequency response is observed for the whole range with a roll-off beginning at 400 GHz, leading to a 3 dB cutoff frequency of 420 GHz. The roll-off approximately follows a 60 dB/dec slope.

To understand the high roll-off frequency, we look at the time constants. Typically, the intrinsic response of a PD is described with $\tau_{tot} = \left(\tau_{rel}^{-1} + \tau_{tr}^{-1}\right)^{-1}$ with $\tau_{rel}$ the relaxation time and $\tau_{tr}$ the transit time[50]. We first look into the transit time. With a device area of $10 \times 10 \, \mu m^2$—and a resulting channel length of $10 \, \mu m$—a frequency of 400 GHz would correspond to a maximum transit time of 2.5 ps. This would result in carrier velocities of $4 \times 10^8$ cm/s, clearly exceeding possible saturation velocities for graphene on an oxide[51]. The results therefore point towards a different current collection mechanism. Due to the semi-metallic nature of graphene, it is possible to describe the graphene PD response with the Shockley−Ramo theorem[52−54].

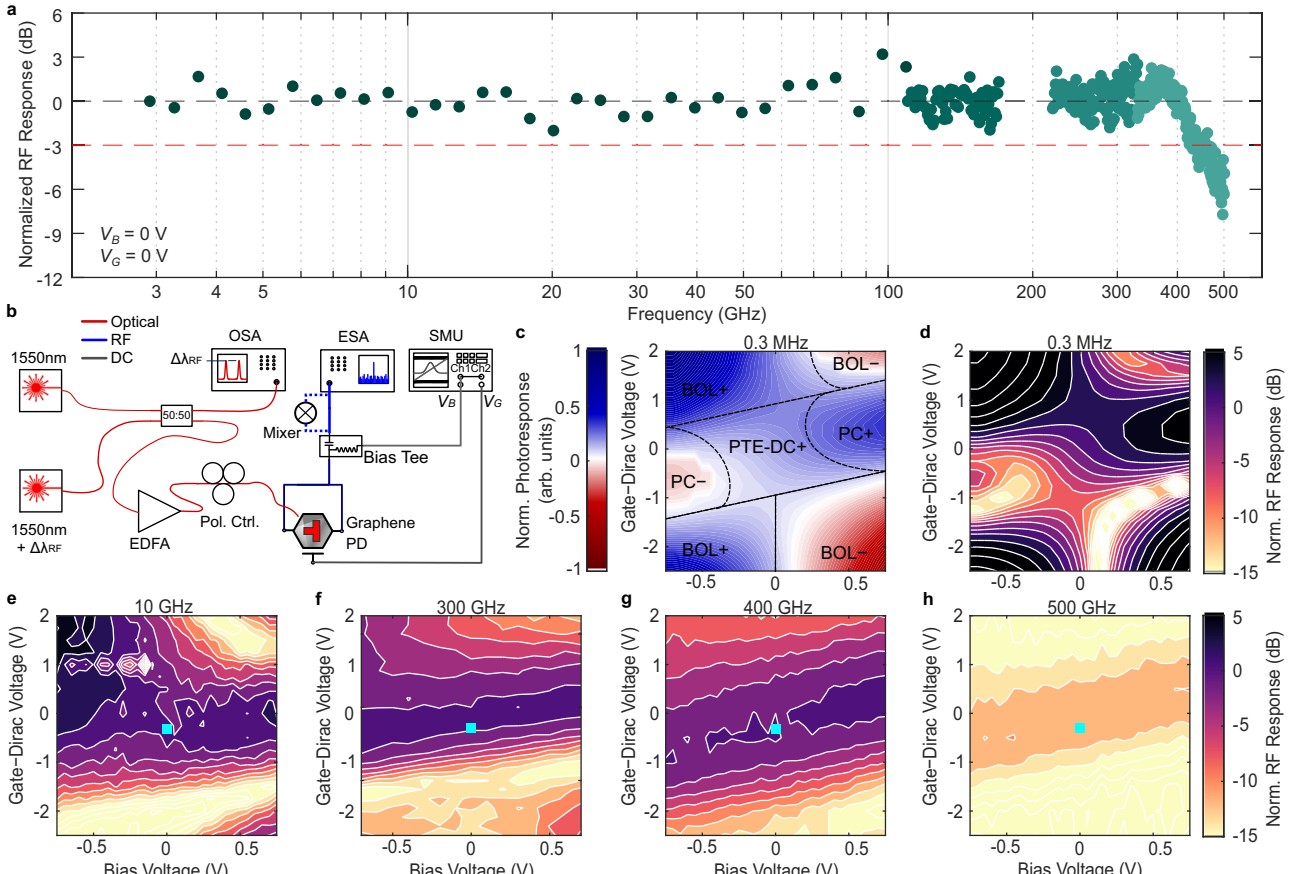

**Fig. 5 | High-speed characterization. a** Normalized radio frequency (RF) response of the device from 3 to 500 GHz. A flat frequency response up to 400 GHz is measured, with a 3-dB cutoff at 420 GHz. Each dot color represents a different frequency setup window, where above 110 GHz subharmonic mixers are employed. The measurement density is increased to 1 GHz steps. **b** Schematic of the employed laser beating scheme. Two 1550 nm lasers with the corresponding frequency detuning are combined to create the RF beating tone which is measured by connecting the device to an electrical spectrum analyzer (ESA). The optical beating is

verified in an optical spectrum analyzer (OSA). A source measure unit (SMU) is used to bias and gate the device. **c** Measured normalized photoresponse on a linear scale at low frequency (320 kHz) and **d** the same data transformed to a normalized RF response. **e–h** High-frequency characterization of bias- and gate voltage-dependent normalized RF response at **e** 10 GHz, **f** 300 GHz, **g** 400 GHz and **h** 500 GHz. The teal square indicates the operation point where the frequency sweep in **a** was recorded.

It describes the quasi-instantaneous current induced by a charge moving within the graphene channel at a distance from the contacts. The current induced at the electrodes is not dependent on the number of charges collected at the contacts but described by the quasi-instantaneous induced electromotive forces created by the locally moving photoinduced carriers. This process is therefore dependent on the relaxation dynamics of the induced electromotive forces, and thereby the relaxation time of the photoinduced hot carriers.

Previous reports on hot carrier dynamics in graphene report a double exponential relaxation dynamic with time constant of ( < 0.1 ps, 2.1 ps)[50] and (0.15 ps, 3.05 ps)[55]. One should note that these studies were carried out with high-quality exfoliated graphene. A recent study, however, found that the maximum bandwidth of a graphene PD is also related to the carrier mobility of graphene[56]. Lower mobilities also have faster decay times which was partially explained by disorder-assisted supercollision cooling. Extrapolating from this assumption to our mobility values, one would arrive at decay rates in the order of ~200 fs, which are also not able to explain the measured 420 GHz cutoff. Still, their observed roll-off also appears to follow a 60 dB/dec slope, which could indicate the same physical limitation. This steep roll-off also points away from an RC-limitation.

We can further test the bandwidth of the different detection mechanisms as we have the possibility to induce different detection mechanisms by controlling the gate and bias voltages. We expand the

frequency response analysis by recording the bias- and gate voltage-dependent photoresponse at four different RF beating tones, 10, 300, 400, and 500 GHz. Firstly, we again look at the low-frequency response (0.32 MHz) where the phase information is available with lock-in detection. Figure 5c depicts the distribution of the dominant detection mechanisms that are labeled accordingly. Subsequently, we transition to logarithmic frequency response plots. This will allow us to better visualize the frequency drop-offs. We start by transforming the linear RF response at 0.32 MHz from Fig. 5c into a logarithmic plot in Fig. 5d. White contour lines are separated by 2 dB each, dark purple colors correspond to strong responses while bright yellow colors indicate low responses. When moving from an almost DC signal to 10 GHz (Fig. 5e), the regions previously associated with the PC effect are disappearing. Moving to 300 GHz (Fig. 5f) shows that the BOL effect (corners of the RF response map) is also weakening. At 400 GHz (Fig. 5g) the corner show almost no more features. We estimate a 3 dB cutoff frequency of the BOL effect in the range of 250–330 GHz (see Supplementary Note 9). Finally, at 500 GHz (Fig. 5h), where we are measuring in the roll-off of the response, the device performance is essentially bias-independent except for the slope corresponding to the shift of the reference ground. These measurement results indicate a quasi-instantaneous Shockley–Ramo response for this device architecture of the PTE-DC, whereas the PC and BOL effects are limited to lower frequencies. The measurements thereby provide speed limitations of

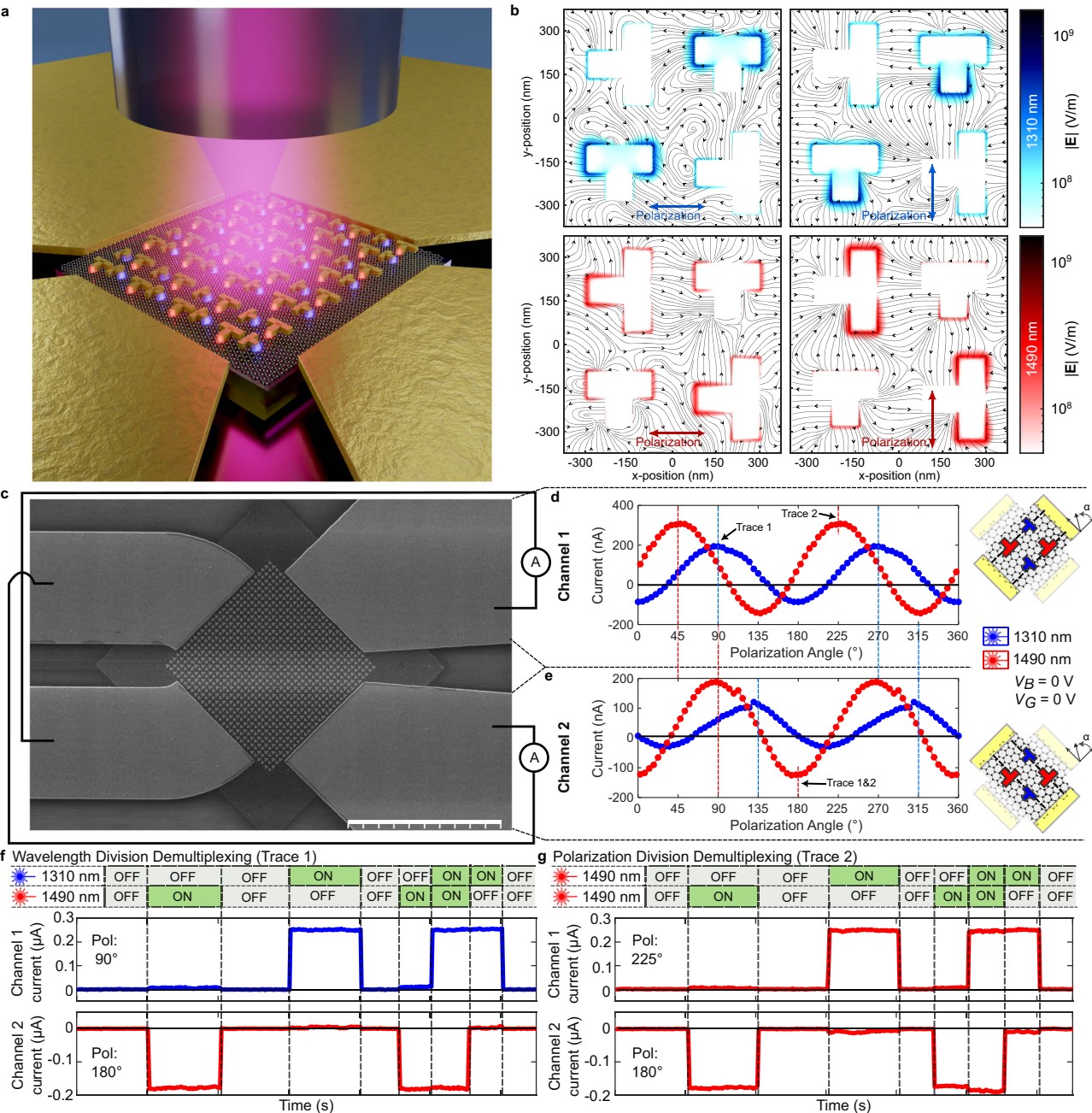

**Fig. 6 | Supercell metamaterial perfect absorber and multiplexing. a** Artistic visualization of a four-contact, multi-resonant device architecture. The metamaterial perfect absorber (MPA) constitutes of a larger unit cell combining four resonators, allowing to target two resonances simultaneously. The four contacts form two perpendicular channels. **b** Simulated absolute electric field response of the unit cell for four different excitations varying the wavelength (1310 and 1490) and polarization (s and p). Strong hot spots are observed for the targeted resonators: small resonators are resonant for shorter wavelengths (blue) and the larger resonators are resonant for longer wavelengths (red). **c** Scanning electron micrograph of the tested four port device with the two perpendicular channels as schematically connected. Scalebar 10 μm. **d**, **e** Response of the two channels as a function of polarization for illumination under 1310 and 1490 nm. **f** Time sequence of both channels for alternating and combined illumination under 1310 and 1490 nm from a single-mode fiber visualizing the wavelength multiplexing capabilities. **g** Time sequence of both channels for alternating and combined illumination for two differently polarized 1490 nm sources visualizing the polarization multiplexing capabilities.

detection mechanisms within graphene and are not hindered by RC-limits or setup limitations. Thus far only one graphene PD architecture consisting of interdigitated electrodes exploiting the photovoltaic effect was able to exceed the frequency response presented here[36].

### Directionality and demultiplexing

We now want to advance the use of the PTE-DC further by showing a proof-of-concept on how two independent detectors can coexist within the same active area. This can be used to resolve the linear polarization of a signal or detect two different wavelengths in the two respective detectors. For this purpose, we realized a four-contact device with a supercell MPA design. We provide a visualization in Fig. 6a. We simultaneously integrate different sized T-resonators optimized for 1310 nm (smaller) and 1490 nm (larger) in a single, larger unit cell. The square graphene sheet is contacted by four electrodes, which form two perpendicular current extraction channels.

The simulated electric field response and induced electron streams to both wavelengths (1310 and 1490 nm) and horizontal and vertical polarization excitations is given in Fig. 6b. The smaller resonators show hot spots if they're excited with 1310 nm. The larger resonators are resonant for incident light at 1490 nm. As for the single resonator case, the baseline or crossbar resonator arm oriented to the incident polarization excites hot spots.

We first verify the PTE-DC operation of the new supercell metamaterial structure. We connect the two channels (1 and 2) of a fabricated device (Fig. 6c) to ammeters and test both perpendicular channels as a function of polarization for both 1310 nm and 1490 nm. Figure 6d, e shows the measured photocurrents of the two channels as function of polarization for 1310 nm and 1490 nm, respectively. The results resolve the PTE-DC character induced by the supercell metamaterial. Minima along one channel correspond to maxima in the perpendicular channel. The maxima further follow the same polarization orientation as the measurements presented in Fig. 3 and Fig. 4 for the two types of resonator orientation with respect to the source-drain lines. The larger resonators have a baseline orientation in channel 1 and a crossbar orientation in channel 2. In channel 1, a positive maximum corresponds to a 45° polarization and a negative maximum to a 135° polarization. In channel 2 a positive maximum corresponds to a 0° polarization and a negative maximum to a 90° polarization. This way, linear polarization of signal can be measured. Here, the smaller resonators have the opposite baseline and crossbar orientations with respect to the larger resonators.

Having two perpendicular channels now opens an opportunity for demultiplexing of signals. We test the device for its potential for two concepts, namely wavelength- and polarization demultiplexing, and show that this can be achieved with low crosstalk. We first combine 1490 and 1310 nm lasers in a single SMF and optimize the individual polarizations to the maximum response in channels 1 and 2 for 1490 and 1310 nm, respectively. The operation points are marked in Fig. 6d, e with "Trace 1". Figure 6f shows a sequence over time (20 s and 10 s blocks) of the readout of both channels where the two lasers are turned on and off in time. Both channels show a response to the corresponding laser and only a small crosstalk to the other laser. On the other hand, Fig. 6g shows the sequence with two 1490 nm lasers combined in a SMF, where the two polarizations are individually controlled to result in a maximum in either one or the other channel (Fig. 6f "Trace 2"). We observe on average ≈3% crosstalk between the two channels, which can be further seen in Supplementary Note 6 where we provide the data also on a logarithmic scale. Separating the two resonance wavelengths further apart or actively controlling the polarization could improve crosstalk further.

The directional character can also be used to detect the linear polarization state of the incoming light. We provide in Supplementary Note 5 design and measurements on the PTE-DC results of a previously reported three terminal device[42] that we placed in our MPA stack. We experimentally verify the architecture's absorption to be larger than >80% for various targeted dimensions with peaks ranging from 1150 nm to beyond 1600 nm.

## Discussion

To conclude, we presented an expanded view on the photothermoelectric-induced directional photocurrents in graphene and how they could be exploited to build a new generation of high-speed photodetectors. The device offers a self-powered 3 dB bandwidth of 420 GHz—thereby being one of the world's fastest photodetectors. The architecture combining graphene in a metamaterial perfect absorber stack successfully tackles the small light-graphene interaction, while simultaneously providing electrostatic doping to induce currents. In combination with the passivation, stable operation of the device with low, CMOS compatible gate voltages is achieved. This allowed us to study the device physics. The different effects such as the bolometric, the photoconductive or the PTE induced directional current strongly dependent on polarization, gate and bias voltages. With a clever choice of gate and bias voltages one can switch between and even add different detection mechanisms. This can be exploited to find the individual frequency responses of the different detection mechanisms revealing that different high-speed photodetection mechanisms within the same graphene device have different bandwidths.

We further leveraged the directionality of the induced currents by introducing a four-contact pad design with a supercell metamaterial. This design highlights the flexibility of the PTE-DC. Direct access to demultiplexing schemes is enabled due to the two perpendicular channels. Furthermore, the flexibility in the metamaterial design and graphene's absorption characteristics enable tunability over a large spectral range. The scheme offers thereby the potential of operation in the MIR spectral range where high-speed detectors do not exist. In addition, they offer the flexibility to provide built-in functionalities such as polarization state detection, dual-wavelength detection and two signal demultiplexing within the same active area removing the need to split-up signals or for bulky constructions. Further, the results clearly highlight that future endeavors on graphene photodetectors should not simply focus on the optimization of the absorption but must rely on a multidisciplinary modeling and design approach. Therefore, photothermoelectric-induced directional photocurrents in graphene are an excellent mechanism to further the understanding in the material, physics, and device design, with the promise of strong performing highest-speed photodetectors.

## Methods

### Photodetector fabrication

The photodetectors were fabricated on a generic intrinsic silicon substrate with a specified resistivity of 10000–1,000,000 Ohm cm on which 1.5 μm of thermal silicon dioxide were grown. Gate pads and back reflectors were created by patterning the structures using electron beam lithography (EBL) followed by etching recesses into the $SiO_2$ using reactive ion etching (RIE). The recesses were filled by electron beam evaporation of gold and subsequent lift-off process. The aluminum oxide spacer layer was grown using plasma-enhanced atomic layer deposition (ALD). Next, commercially available monolayer graphene grown on copper foil via chemical vapor deposition (CVD) was transferred using conventional wet transfer with a PMMA support layer. Using EBL and RIE, the graphene was patterned. Next, contact pads and metamaterial resonators were added again by using EBL, electron beam evaporation and lift-off. Lastly, all structures were encapsulated by an aluminum oxide layer grown by ALD and contact pads are made accessible by standard photolithography and wet chemical etching.

### Low-frequency characterization

Zero-bias photoresponse measurements were performed using a simplified version of the setup presented in Fig. 5b: The detector is connected to the source measure unit (SMU) and illuminated by a non-modulated tunable laser source (TLS). The TLS is polarized by using a polarization synthesizer. The generated photocurrents as function of polarization and gate voltage are then read out at the SMU. If several channels are measured simultaneously, multiple SMUs were employed.

For biased photoresponse measurement, the TLS was amplitude modulated with an acousto-optic modulator at a frequency of 320 kHz. The electrical read-out was expanded by including a bias tee where the AC path was connected to a lock-in amplifier.

### High-frequency characterization

To characterize the RF frequency response of the device, we used a laser beating setup as depicted in Fig. 5b. Two lasers are combined in a

single fiber where one laser is fixed at 1550 nm and the other is detuned by a difference frequency $\Delta\lambda_{RF}$. The optical power at the fiber output was set to 12.5 dBm. The beat-note at $\Delta\lambda_{RF}$ was measured by capturing the device response with an electrical spectrum analyzer (ESA). The ESA (Keysight UXA N9041B) offers a frequency range up to 110 GHz. The device was connected to the ESA with high-frequency GSG-probes, a bias tee, and an RF cable. The probe losses are compensated with the loss calibration provided by the vendor. The bias tee and RF cable losses have been calibrated by performing reference measurements using a commercial high-speed photodetector. Frequencies above 110 GHz were accessed by employing subharmonic mixers (Virginia Diodes, SAX modules) connected to the ESA. The high-frequency probe losses are again compensated with the loss calibration provided by the vendor. The loss or respective conversion efficiencies of the subharmonic mixers are compensated with the calibration data provided by the vendor.

### Optical simulations and simulated current flows

Simulations of the optical electric field response and the absorption of the metamaterial structure were carried out in a commercial 3D finite element method solver (CST Microwave Studio 2022). The metamaterial was modeled as a single unit cell with periodic boundary conditions excited by Floquet modes. The simulated electric field responses were used to calculate the driving forces for the current flow simulations performed as 2D simulations in a steady-state turbulent flow environment (Comsol Multiphysics 6.0). Further details are provided in the Supplementary Note 3.

### Spectroscopic measurements of optical properties

Reflection measurements were carried out in a custom-built reflection microscopy setup. A broadband light source (halogen lamp) was coupled to the microscope to illuminate the sample by using a ×10 magnification objective (NA = 0.26). Test metamaterials for optical characterization have a size of ~200 × 200 μm$^2$. The reflected light was collected and coupled to an optical spectrum analyzer. To calibrate the path and lamp, a gold mirror was used as a reference. As the transmission through the gold back reflector was zero, we calculated the absorption of the metamaterial by dividing the measured reflection by the reflection of the gold mirror.

## Data availability

Data from this work is provided in the manuscript, supplementary information figures and from the corresponding author upon request.

## Code availability

Code from this work is available upon request by contacting the corresponding authors.

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

## Acknowledgements
This work was supported by the European Union's Horizon 2020 Research and Innovation Program through the project aCryComm, FET Open Grant Agreement no. 899558 and the EMPIR programme project SuperQuant (20FUN07), which has received funding from the EMPIR program co-financed by the Participating States and from the European Union's Horizon 2020 research and innovation program. The Swiss National Science Foundation (SNF) is acknowledged for support through the REquip program (206021_198113). The authors thank the Cleanroom and Operations team of the Binning and Rohrer Nanotechnology Center (BRNC) for their support. We thank Virginia Diodes, Inc. (VDI) for lending additional equipment.

## Author contributions
S.K. and R.G. developed the device structure, where establishing the concept and structure was supported by J.L. S.K. developed the fabrication process and simulation routine. S.K., R.G., and M.B. performed the measurements, while Y.K. and M.D. helped in developing the measurement routines. R.G. and Y.K. supported the simulation and model development. D.R. and A.G. performed the current flow simulations. Y.K., S.N., K.K., and Y.F. supported the fabrication development. M.D. supported in the generation of the figures. The manuscript was prepared by S.K., which was revised by all co-authors.

## Funding

## Competing interests
The authors declare no competing interests.
