## [Peer Review File · Nature Communications]

Controlling Photothermoelectric Directional Photocurrents in Graphene with over 400 GHz BandwidthREVIEWER COMMENTS

Reviewer #1 (Remarks to the Author):

Review of “Highest-Speed Photothermoelectric Directional Currents in Graphene”

The article “Highest-Speed Photothermoelectric Directional Currents in Graphene” by Koepfli et al. shows a high-speed graphene detector based on a metamaterial perfect absorber suitable for NIR and MIR detection regime. The manuscript is very clear, instructive, and well written. Having said that, I do not see any relevant novelty (neither fundamental nor applications-related) justifying a publication in Nature Communications. In the following I’ll explain my point of view and my major concerns.

Fundamental aspects

1. Concerning the fundamental aspects covered in the paper, the authors discuss known concepts and designs, as PTE DCs have already been analyzed in some papers also cited by the authors. Indeed 10.1038/s41467-020-20115-1 proposes the same geometry and shows measurements very similar to the one shown in this manuscript. It is already known that PTE is responsible for DC (10.1038/s41566-021-00819-6, 10.1038/s41566-022-01115-7, 10.1038/s41467-023-39071-7). The dynamic flow model has been already applied to study directional currents as a function of light polarization, and as a function of the metal contact disposition with respect to the electric field resonance produced by the nanostructures composing the metamaterial (10.1038/s41566-021-00819-6).

2. Authors say that the “the interaction with other effects is not yet understood”. The authors calculated the PTE DC using known formulas. Then, they experimentally observed that by biasing the graphene channel, PC and PB effects add up. This is known: at zero bias, the only mechanisms occurring are PV and PTE. Instead, photocurrent generation in biased graphene is at the charge of PC or PB effects, depending on the graphene chemical potential (see, as an example, 10.1038/nphoton.2012.314). Of course, these two effects add up constructively or destructively with the photocurrents produced by PV or PTE effects, which are mainly bias independent (or even degrade with bias). The competition between many photocurrent mechanisms has also been experimentally shown in many papers. Mainly all the papers (except very few) dealing with PTE detectors show PB operations under source-drain bias. This is done to measure, e.g., eye diagrams, since the PB effect is, as a matter of fact, stronger than PTE for samples with mobilities $< \sim 5000 \text{ cm}^2\text{V}^{-1}\text{s}^{-1}$.

Applications-related aspects

3. The title of the work starts with the words “Highest-Speed”, from which we can derive that the authors consider these as a major achievement of their work. I agree, nevertheless, photo

thermoelectric and bolometric detectors based on graphene reaching set-up limited bandwidths have been demonstrated in a plethora of papers in the last years. The ultrahigh speed of these detectors is a fact, supported by theory and by fundamental experiments revealing the ultrafast dynamics of hot electrons in graphene. For these reasons, while showing the record speeds of graphene detectors is very useful for the community, these achievements cannot be ascribed as breakthroughs or novelties anymore, but rather as useful information confirming known facts, as no real scientific/ technical originalities rise from these measurements.

4. As the authors surely know, while frequency response is not an issue affecting graphene detectors, responsivity instead is. Specifically, addressing responsivity *and* linearity at the same time is the only real missing key performance parameter required to compete with SOTA. I completely agree with authors when they assert that “most of these approaches suffer from early optical power saturation onsets due to a high confinement onto a small active area”. Indeed, the big issue of graphene detectors is that when responsivity enhancement is achieved, saturation takes place at very low optical input powers. Instead, linear detectors present very low responsivity. This issue comes from the 2-dimensional nature of graphene and is still not solved. Concerning this, responsivity is not mentioned in the main text, but just in the supplementary information, and is very low compared to graphene SOTA detectors. It should be increased by x1000 to compete with SOTA graphene detectors or UTC-PDs exhibiting even higher bandwidths. At the same time, responsivity should be kept linear, with optical power levels > 0 dBm before saturation, to compete with SOTA technology. Since addressing just one of these two KPIs is completely useless for applications, it would be interesting to give information about device response saturation, and how this is related to responsivity. Would a responsivity increase affect the response saturation against optical power, as happens for all reported devices so far? If they claim that this is not the case, why? Are there theoretical reasons? Do they have experimental evidences?

5. Authors propose in the supplementary information some improvements to increase device responsivity. First, they claim that passing from low quality polycrystalline graphene to exfoliated graphene would increase the device responsivity. I agree, and this can be achieved also using single crystal graphene grown by CVD and encapsulated with hBN, since exfoliated graphene has nothing to do with real applications. Then, they compare their device with the one in 10.1038/s41467-020-20115-1. The comparison ends up with a multiplication which is in my opinion not realistic: mobility, number of layers and resistivity are decoupled, and considered separately, which does not make much sense: mobility and resistivity are actually directly connected, number of layers determine resistance and mobility as well... in other words, these parameters should be put together into a simulation and cannot be considered separately. Again, even reaching the (already demonstrated) value of 36 mA/W, what is the maximum optical power that can be coupled before saturation?

6. Having said that application relevant performances are in my opinion not considered in the main text and are revealed to be low in the supplementary information (even taking into account the list of improvements in the supplementary file), application relevant concepts and functionalities enabled by this graphene detector could be sufficient to justify a novelty. Nevertheless, the proposed concepts and functionalities are not peculiar to this device and the arguments of the

authors about this aspect are quite weak.

-Authors claim that their device supports operations in a broadband optical range. They mention many times the MIR window. Moreover, they assert that other solutions like, e.g., waveguide-integrated devices would not afford such “broadband” operation. First, the paper does not show any MIR operation, apart from a supplementary section showing a simulation. Second, the operation of the proposed device is wavelength dependent as for integrated photonic devices based on, e.g., silicon nitride or Si, which design can be optimized to operate at a central wavelength, with optical bandwidths chosen to trade-off efficiency. Integrated photonic waveguides can operate in ranges even larger than those experimentally considered by the authors, in many cases with very low design efforts. Moreover, the wavelength tuneability of the proposed device requires lithographic accuracies in the order of tens of nm, which is not exactly of easy implementation using standard photonic technology. Indeed, the real device central wavelength operation is different from the designed one, as also observed by the authors. Therefore, I really do not see the author’s point in this claim. At least, I don’t see why integrated solutions are weaker than the proposed one.

-The discussion of SOTA on polarization/ wavelength demultiplexing is not present in the text. How does their polarization /wavelength demultiplexing compare with already existing systems in terms of efficiency and compactness (also considering the space occupied by the active polarization control)? Polarization/wavelength demultiplexing is a function which can be easily performed not only for two wavelengths. It is implemented for many wavelengths in both CWDM a DWDM systems, also using integrated photonics. In addition, the scheme proposed by the authors is sensitive to polarization when “demultiplexing” two wavelengths, and sensitive to wavelength when detecting polarization. Moreover, active polarization control is needed. What is the application for such a detector? The “small” crosstalk of 3% is 15 dB, which is in my opinion not interesting for applications, where $\gg 20$ dB isolation between the channels are required in systems in which many wavelengths are managed. How can this be solved, also accounting that real applications require \gg two wavelengths? How the responsivity and wavelength selectivity scale when considering more than two wavelengths? What is the problem that this scheme could solve and that is not already solved with existing systems?

In conclusion, my general opinion is that this manuscript shows a work based on known fundamental phenomena. This fact by itself would not be a point for negatively judging the paper, if new application-relevant novelties or ideas would have been shown. But from this standpoint, it presents just a well performed engineering exercise (which certainly could be suggestive for further developments), with no sufficiently relevant application perspectives.

Other observations:

7. Authors should put the real image of the RF device with the dimensions.

8. As far as I understood, the device length is 10 μm . Apart from transit time and quasi-instantaneous response of PTE, one needs to consider the pads capacitance which is most probably responsible for the frequency cut-off. Considering that the resistance of the device is 6000 ohms, I get $\sim 5\text{fF}$ parasitic pads capacitance to get ~ 450 GHz, which is impressively good!

Maybe the authors could add information about that, and about the RF pads design. Also, information about the substrate resistivity should be included to show that no RF ohmic losses occur due to substrate.

9. Given the low power level of the detector, authors should add more information about the noise of the equipment they use, and the calibration of the same. How did the authors calibrate the measurement? Did they use reference signals generated by, e.g., commercial UTCs with known frequency response? The information about the measurement is, in general, not exhaustive, given the large, spanned bandwidth and the different components used in the different frequency ranges.

10. Apart from the “multiplication” approach discussed in point 5, that is in my opinion not appropriate, I have some concerns on the observations done in the calculation:

- Metal on top of graphene pins the chemical potential but can also degrade graphene quality and so its mobility. Maybe low mobility graphene would not be strongly affected from metal decoration, but high quality could, since high quality graphene is more sensitive to defects. Considering this, are authors sure that higher quality graphene would give 3x mobility if metal is put on top?

- Concerning resistance by itself, why does decreasing resistance by 5 means x5 responsivity? Putting low impedance detectors in parallel does not necessarily mean increasing responsivity, this can be easily checked using simple equivalent circuits. Maybe authors refer to the increased electrical power transfer from the detector to the load, which would rise because of impedance matching? Indeed, the electrical power transfer would rise by > 30 dB while passing from 6k Ω to 50 Ω . Anyway, this is not related to (DC) responsivity in my opinion but only in power transfer efficiency from the detector.

Reviewer #2 (Remarks to the Author):

Koepfli et al report zero-bias graphene photodetectors that feature centrosymmetry-broken nanostructures and vectorial photoresponse. While the design concept of “T”-shaped structures has been reported by another group [Nat Commun 11, 6404 (2020)] as acknowledged by the authors, this work further studies mechanisms by examining the frequency response of the devices up to 500 GHz. They revealed that the photoresponse based on bolometric and photoconductive effects would vanish above about 10 GHz, and the hot carriers assisted photo-thermoelectric effect allows an ultrahigh 3dB cutoff frequency of 420 GHz, which is far faster than the state-of-the-art photodetectors (usually below 100 GHz, even after integrating with waveguides. See Nat. Nanotechnol. 15, 118–124, 2020). Besides, the authors use CVD-grown graphene rather than exfoliated graphene, representing a big step towards the practical applications of this new type of device, although the responsivity is still two orders smaller (200-300 $\mu\text{A/W}$) than exfoliated ones (> 30 mA/W). Overall, I think this work reports exciting and promising results for ultrahigh-speed photodetectors and also deepens our understanding of the underlying mechanisms, so I would highly recommend the publication on Nature Communications if the authors can address my

following concerns:

1. Title. An expression like “highest-speed” sounds unprofessional. I would recommend using an explicit number.
2. Comparison with other works. The authors have just reported a similar work based on asymmetric contact materials which shows a response speed of 500 GHz [Koepfli, S. M. et al. Metamaterial graphene photodetector with bandwidth exceeding 500 gigahertz. *Science* 380, 1169–1174 (2023)]. So, I don’t think this work shows the “highest” speed.
3. Frequency-dependent contribution of different mechanisms. In Figure 5a, the authors provide the measured frequency response at $V_b = V_g = 0$ V, where PTE dominates. By looking at Figure 5c-h, it seems PC and Bol will be largely suppressed > 10 GHz. My question is, is possible to extract the respective cut-off frequencies for the PC, Bol effects also? For example, in low-frequency or even DC range? What are the percentages for each mechanism? Besides, the authors report that gate voltage also affects the speed. I would recommend the authors provide more details about the microscopic mechanisms.
4. Another question to consider: what is the special design in their characterization setup? While the response speed of graphene has been extensively measured in numerous studies, the majority of these measurements have been limited to frequencies below 40 GHz. How did the authors ensure the accuracy and reliability of their measurements?
5. What limits the 3dB frequency? As the authors have demonstrated in a recent *Science* paper, similar architecture can reach 500 GHz. What is the factor to limit this work to 420 GHz then?
6. I suggest that the authors include time response measurements, such as eye patterns, as complementary data to their frequency response analysis. This additional data would strengthen their assertion of a > 400 GHz 3dB cut-off frequency.
7. Figure 3b caption: What do the authors mean by “The Dirac point is found close to 0 V at 0.6 V”?
8. Figure 3g bottom panel: 180o should be a typo? It should be (0o & 90o).
9. Figure 3h: The unit of absorption is missing.

Reviewer #3 (Remarks to the Author):

The authors Stefan et al. report the highest photothermoelectric response of metamaterial integrated graphene photodetector. The ultrahigh frequency response over 400 GHz is impressive, which is supported by the quasi-instantaneous response mechanism. On the other hand, the photothermoelectric response mechanism, as well as other response mechanisms were also studied systematically. There are some concerns and questions should be addressed before considering publish in *Nature Communications*. Please see following comments.

1. Could the author simulate the typical absorption in graphene, but not the total absorption?
2. Although the authors report the ultrafast photoresponse speed and discussed the possible limiting factors, but the cut-off frequency of 420 GHz still not be well explained. Could the author

further study this point, as reveal the limitation of cut-off frequency is meaningful for the design of other high-speed PDs.

3. Besides highest response speed, the responsivity and detectivity are also important. How the thickness of graphene influences the responsivity and detectivity?

4. For the application demonstration, the authors show the wavelength and polarization division multiplexing. However, the time scale is in second level as shown in Figure 6, which is incommensurate to its ultrafast response speed.

5. The English of this manuscript should be further polished for easy reading.

Reviewer 1

Below are detailed responses to each of the reviewer's comments (*italic gray*), highlighted in **blue** for clarity. Additionally, alterations made to the initial manuscript are clearly marked in **red** which are directly added below each comment.

"The article "Highest-Speed Photothermoelectric Directional Currents in Graphene" by Koepfli et al. shows a high-speed graphene detector based on a metamaterial perfect absorber suitable for NIR and MIR detection regime. The manuscript is very clear, instructive, and well written. Having said that, I do not see any relevant novelty (neither fundamental nor applications-related) justifying a publication in Nature Communications. In the following I'll explain my point of view and my major concerns."

1.	Concerning the fundamental aspects covered in the paper, the authors discuss known concepts and designs, as PTE DCs have already been analyzed in some papers also cited by the authors. Indeed 10.1038/s41467-020-20115-1 proposes the same geometry and shows measurements very similar to the one shown in this manuscript. It is already known that PTE is responsible for DC (10.1038/s41566-021-00819-6, 10.1038/s41566-022-01115-7, 10.1038/s41467-023-39071-7). The dynamic flow model has been already applied to study directional currents as a function of light polarization, and as a function of the metal contact disposition with respect to the electric field resonance produced by the nanostructures composing the metamaterial (10.1038/s41566-021-00819-6).
	We are sorry to hear that the many new aspects introduced by this paper were not clearly communicated. Also, as pointed out by the reviewer, we cite all the works correctly and we do not claim novelty on any of the above mentioned aspects. Of course, we had to introduce the interesting geometry that was introduced by others as we performed research on this geometry. It is not possible to discuss the device without introducing it first. This also includes making use of the already developed modelling framework. We would like to point out the many messages and novelties of this manuscript based on the figures: - Figure 1 & 2: Introduction of the device architecture. Novel here is that we incorporate the resonators in a metamaterial perfect absorber layer stack. This boosts the absorption and additionally makes gating possible with low CMOS compatible voltages – we demonstrate operation voltages ~1 V as opposed to ~100 V from the previous demonstrations.- Figure 4: Here we describe again the physics of the device and test the interaction of the PTE-DC with other effects; something that has not yet been shown either in simulation or experimentally. These new types of measurements show that it is possible to clearly distinguish between different detection mechanisms in graphene which is rather difficult. By having – in addition to bias and gate voltage – the possibility to control the polarization in a manner that does not change the overall absorption in the graphene but only where the absorption is happening there is the possibility to turn on and off mechanisms making them distinguishable.- Figure 5: The high-speed characterization of the device where we show a 420 GHz 3 dB frequency cut-off. We note that we do not show a setup limited performance but find a frequency cut-off of the detection mechanism. Furthermore, we find that different detection mechanisms within the same device show different frequency cut-offs. All these findings are not only remarkable (as of the high cut-off) but have never been reported before.- Figure 6: We introduce the demultiplexing scheme showing the flexibility of the metamaterial design, i.e., that directional currents can be induced by different metamaterial sub cells allowing to combine different resonances (wavelength and/or polarisation) within the same architecture. Therefore, in our view, only Figure 3 is a re-iteration of already published work. However, in our view Figure 3 is helpful to understand the full discussion in the manuscript. We answer the more detailed questions below. Wherever we make changes to the manuscript we consider the comment on the novelty and make changes accordingly to communicate the novelty more clearly.

2.	Authors say that the “the interaction with other effects is not yet understood”. The authors calculated the PTE DC using known formulas. Then, they experimentally observed that by biasing the graphene channel, PC and PB effects add up. This is known: at zero bias, the only mechanisms occurring are PV and PTE. Instead, photocurrent generation in biased graphene is at the charge of PC or PB effects, depending on the graphene chemical potential (see, as an example, 10.1038/nphoton.2012.314). Of course, these two effects add up constructively or destructively with the photocurrents produced by PV or PTE effects, which are mainly bias independent (or even degrade with bias). The competition between many photocurrent mechanisms has also been experimentally shown in many papers. Mainly all the papers (except very few) dealing with PTE detectors show PB operations under source-drain bias. This is done to measure, e.g., eye diagrams, since the PB effect is, as a matter of fact, stronger than PTE for samples with mobilities $< \sim 5000 \text{ cm}^2\text{V}^{-1}\text{s}^{-1}$.
	We thank the reviewer for his comment on the interaction of the graphene detection mechanisms. We agree with the above statements, but we want to highlight that our measurements go beyond the above stated. While intuitively and from previous reports these interactions of the effects are clear, there is no experimental demonstration of the interaction of the PTE induced directional currents with the PB and PC effects. Furthermore, while one would be able to estimate regions of detection mechanisms, the magnitude of each effect is very difficult to estimate. We are not aware of a complete framework that is able to model PV, PC, PB and PTE accurately. The polarization dependence of the PTE induced directional currents combined with the polarization independent absorption characteristics of the devices’ metamaterial allow for an ideal method to also compare magnitude and interaction of the effects which is summarized in Fig. 4. The reviewer comments that mainly all the papers dealing with PTE detectors show PB operations under source-drain bias. As one can see from Fig. 4 (e-g) there is a strong gate dependence and the PTE induced directional currents can co-exist both with the BOL and PC effect. Therefore, our measurements reveal that the biased operation of the PTE device does not automatically mean a BOL operation, but that also a PC operation comes into play. This insight can become important when additional factors modifying the PC and BOL response come into play, e.g., temperature. As the interaction regions of PC, BOL and PTE are depending on the contact doping given by the resonator material this interaction can also be set. Furthermore, the reviewer comments that the PB effect is stronger than the PTE effect and is therefore used to measure eye diagrams. We find in our devices that the PTE induced directional currents reach $\sim 3/4$ of the PB response. While the response is indeed weaker, the PB effect comes inherently with large dark currents due to the conductivity of graphene. We found in our previous work (10.1126/science.adg8017) on the PV effect that the zero-bias operation outperforms the biased operation in data measurements due to the much lower noise. Furthermore, zero-bias operation clearly grants advantages in circuit complexity, no required voltage source and furthermore also does not have an electrical power consumption. As we show for the first time in the frequency response measurements the BOL effect is also slower than the PTE-DC. Therefore, in our experience there is no advantage in operating such a device as a bolometer as often done in literature. We added the following sentences to the manuscript to highlight more the implications of the measurements presented in Fig. 4: p8, l43: “These results indicate that the different detection mechanisms in graphene are additive. The possibility to control the polarization in a manner that does not change the overall absorption in the graphene but only where the absorption spatially is happening allows to turn on and off mechanisms making them clearly distinguishable. By choosing the contact metal⁴⁹ of the resonators appropriately and control of the gate voltage, one could further optimize the combination of the detection mechanisms and the peak operation.

3.	The title of the work starts with the words “Highest-Speed”, from which we can derive that the authors consider these as a major achievement of their work. I agree, nevertheless, photo thermoelectric and bolometric detectors based on graphene reaching set-up limited bandwidths have been demonstrated in a plethora of papers in the last years. The ultrahigh speed of these detectors is a fact, supported by theory and by fundamental experiments revealing the ultrafast dynamics of hot electrons in graphene. For these reasons, while showing the record speeds of graphene detectors is very useful for the community, these achievements cannot be ascribed as breakthroughs or novelties anymore, but rather as useful information confirming known facts, as no real scientific/ technical originalities rise from these measurements.
	We thank the reviewer for his comment on the high-speed characteristics. We indeed believe strongly that the high-frequency response – and characterization – are some of the key results of this work. To the best of our knowledge, there is no other device technology able to achieve 420 GHz bandwidth and there is only one other demonstration showing >500 GHz with graphene. Nevertheless, we see that the title might not age well. We therefore changed the title to:
	Controlling Photothermoelectric Directional Photocurrents in Graphene with over 400 GHz Bandwidth
	The reviewer points out that there has been a plethora of setup limited graphene PD papers over the years. While this is true, one has to consider that a setup limit of 110, 70, 40 or even 10 GHz does hardly compare to 500 GHz. Furthermore, we do not experience a setup limitation, but find a device limited bandwidth of 420 GHz. We measure the bandwidth of the detection mechanism and not an RC limited cut-off. We therefore cannot fully agree with the last part of the reviewer’s statement. The high speed of graphene PDs is not a fact and requires considerable engineering effort to achieve. We agree that the fast dynamics of hot electrons in graphene are a fact – but even here it is not well understood what the influence of substrate, passivation and material quality is. For example, one can look at a more recent study on graphene carrier dynamics (10.1038/s41566-022-01058-z). Here not only the gate dependence on the relaxation dynamics is tested but also the decay rates across four different samples with strongly varying carrier mobilities. The different graphene qualities lead to strongly varying carrier dynamics. Depending on the graphene quality, the carrier dynamics seem further influenced. The topic is therefore far from being entirely understood and additional measurements are required and should be reported. To highlight the novelty in our frequency characterization more strongly, we expand the discussion on the frequency response in the manuscript as follows:
	p10, l26: “When moving from an almost DC signal to 10 GHz (Fig. 5e), the regions previously associated with the PC effect are disappearing. Moving to 300 GHz (Fig. 5f) shows that the BOL effect (corners of the RF response map) is also weakening. At 400 GHz (Fig. 5g) the corner show almost no more features. We estimate a 3 dB cut-off frequency of the BOL effect in the range of 250 to 330 GHz (see Supplementary Information). Finally, at 500 GHz (Fig. 5h), where we are measuring in the roll-off of the response, the device performance is essentially bias independent except for the slope corresponding to the shift of the reference ground. These measurement results indicate a quasi-instantaneous Shockley-Ramo response for this device architecture of the PTE-DC, whereas the PC and BOL effects are limited to lower frequencies. The measurements thereby provide speed limitations of detection mechanisms within graphene and are not hindered by RC-limits or setup limitations. Thus far only one graphene PD architecture consisting of interdigitated electrodes exploiting the photovoltaic effect was able to exceed the frequency response presented here³⁶.”

	p12, 138: This can be exploited to find the individual frequency responses of the different detection mechanisms revealing that different high-speed photodetection mechanisms within the same graphene device have different bandwidths.
4.	As the authors surely know, while frequency response is not an issue affecting graphene detectors, responsivity instead is. Specifically, addressing responsivity and linearity at the same time is the only real missing key performance parameter required to compete with SOTA. I completely agree with authors when they assert that “most of these approaches suffer from early optical power saturation onsets due to a high confinement onto a small active area”. Indeed, the big issue of graphene detectors is that when responsivity enhancement is achieved, saturation takes place at very low optical input powers. Instead, linear detectors present very low responsivity. This issue comes from the 2-dimensional nature of graphene and is still not solved. Concerning this, responsivity is not mentioned in the main text, but just in the supplementary information, and is very low compared to graphene SOTA detectors. It should be increased by x1000 to compete with SOTA graphene detectors or UTC-PDs exhibiting even higher bandwidths. At the same time, responsivity should be kept linear, with optical power levels > 0dBm before saturation, to compete with SOTA technology. Since addressing just one of these two KPIs is completely useless for applications, it would be interesting to give information about device response saturation, and how this is related to responsivity. Would a responsivity increase affect the response saturation against optical power, as happens for all reported devices so far? If they claim that this is not the case, why? Are there theoretical reasons? Do they have experimental evidences?
	We thank the reviewer for his comment on the responsivity. We agree that increasing the responsivity is one of the major challenges for graphene photodetectors. Here, we would like to stress that our paper offers a solution to the very challenge. What we show is that for high-speed applications what matter is the maximum achievable output signal which is the product of responsivity and saturation power or its linear range. Here we are able to compensate in part for the relatively low responsivity as we can operate the device at higher optical input powers. We can use higher optical input powers as we are spreading the power across a larger area in comparison to e.g., waveguide based devices. While the atomical thickness remains the same, the power per area on the photodetector is much lower in such a top illuminated device. So, we believe that even when we increase the responsivity of the device, we will not run into the same saturation issues as many other devices experience. We conducted an additional measurement on the linearity of the device which we added to the supplementary information. We swept the optical input power over more than 3 orders of magnitude. Here we now used a lensed fiber that we procured in the meantime. The smaller illumination spot (~5 μm) ensures that the optical power is coupled to the device’s active area. The lensed fiber creates a better overlap compared to a standard single mode fiber, but also creates under-illuminated areas within the device. A slow loss of linearity in the 5-10 mW regime is observed. We comment on this further below on the reviewer’s question on saturation. The changes are summarized in the adapted Supplementary Note 8 including an additional figure. We also added the responsivity to the main text:
	p7, 15: “This indicates that the measured total absorption is linked to the absorption in graphene and that the PTE-DC can indeed be enhanced with a MPA stack. The responsivity is found to be ~1.5 V/W and photovoltage signals >18 mV are extractable with the high optical power tolerance, see Supplementary Note 8 for more details.”

Supplementary Note 8: Discussion on responsivity

Fig. S15: Voltage and current responsivity and linearity

a Device resistance and normalized photocurrent response as function of gate voltage. The maximum current responsivity is found at a gate voltage of 0.5 V where the device resistance is 2.75 kΩ. **b** Resonator orientation of the specified device and the polarization orientation convention used (blue – 0°, red – 90°). **c,d** Photoresponse as function of optical input power for the 0° polarization orientation and a wavelength of 1480 nm. The corresponding responsivities are extracted from the slope as **c** $\mathcal{R}_I = -0.462 \text{ mA/W}$ and **d** $\mathcal{R}_V = 1.4 \text{ V/W}$. **e,f** Photoresponse as function of optical input power for the 90° polarization orientation and a wavelength of 1550 nm. The current and voltage responsivities are **e** $\mathcal{R}_I = 0.424 \text{ mA/W}$ and **d** $\mathcal{R}_V = -1.6 \text{ V/W}$. The devices have been illuminated with a lensed fiber (focus spot $\sim 5 \mu\text{m}$) to ensure that all light is focused to the active area of the device. A loss of linearity in the 5-10 mW regime is observed.

At this point, what is still open is the performance in terms of responsivity. The presented devices in this work have responsivities of $\sim 1.5 \text{ V/W}$ or $\sim 400\text{-}500 \mu\text{A/W}$, see also Fig. S15. These responsivity values are considerably lower than the 27 V/W or 36.3 mA/W of the first demonstration⁷. However, the responsivity of the device in this paper can be increased by quite a few measures. ...

We believe that the photodetectors have the potential to reach the 0.1 A/W mark (as outlined in the supplementary information). With these values, they would match or outperform other graphene devices and also other technologies such as UTC-PDs. We also highlight that we provide an external responsivity unlike most WG based devices that deduct coupling losses and only provide an internal responsivity. We further want to note that, to the best of our knowledge, there is no UTC-PD with a bandwidth covering the low GHz up to 420 GHz, especially not without any external bias (SOTA zero-bias: 135 GHz 10.1063/5.0119244). Furthermore, the comparison to other technologies should also take into account other factors than responsivity and bandwidth. A graphene based device has clear advantages in manufacturability, integration capabilities and cost. Also, the first UTC-PD demonstration happened ~ 15 years before the first high-speed graphene PD demonstration and benefitted from decades of PIN-PD developments. We believe, if given time, graphene will be able to compete with the old and established technology which took many years to be developed.

5.	Authors propose in the supplementary information some improvements to increase device responsivity. First, they claim that passing from low quality polycrystalline graphene to exfoliated graphene would increase the device responsivity. I agree, and this can be achieved also using single crystal graphene grown by CVD and encapsulated with hBN, since exfoliated graphene has nothing to do with real applications. Then, they compare their device with the one in 10.1038/s41467-020-20115-1. The comparison ends up with a multiplication which is in my opinion not realistic: mobility, number of layers and resistivity are decoupled, and considered separately, which does not make much sense: mobility and resistivity are actually directly connected, number of layers determine resistance and mobility as well... in other words, these parameters should be put together into a simulation and cannot be considered separately. Again, even reaching the (already demonstrated) value of 36 mA/W, what is the maximum optical power that can be coupled before saturation?
	We thank the reviewer for his comment on the responsivity improvement discussion. Indeed, the low quality polycrystalline graphene does not need to be replaced with exfoliated graphene but could also be replaced with monocrystalline graphene grown by CVD. To clarify that we refer to the quality and not the fabrication method, we change the sentence accordingly: SI p13, l6: "Moving from high quality mechanically exfoliated few layer graphene to CVD grown polycrystalline monolayer graphene can explain a large fraction of the deficit." Regarding the multiplication of the improvement factors. We agree that mobility, number of layers and resistivity are interconnected. We state that mobility influences the Seebeck coefficient and the resistance. We further say that the multilayer graphene influences absorption and resistance. In the end, we multiply: (Seebeck coeff.) × (Absorption enhancement) × (wavelength unit cell change) × (resistance) These coefficients are independent of each other, but as the reviewer pointed out correctly, linked to the graphene properties such as mobility and thickness. We stress the meaning of the multiplication by adding the meaning of each factor to the text: SI p13, l24: "Multiplying all estimated contributions leads to a total of (Seebeck coeff.) × (Absorption enhancement) × (wavelength unit cell change) × (resistance factor) $3 \times 4 \times 4 \times 5 \sim 10^2$ explaining the two order of magnitude lower current responsivity in our devices." We note again, as mentioned in the manuscript, that this is only a very rough estimate to explain the two orders of magnitude lower responsivity values. To estimate the saturation power when the responsivity increases is extremely difficult. Some literature reports that the responsivity for the PTE does scale with P_{in}^0 for low powers and later on scales with $P_{in}^{-1/3}$ (10.1038/s41467-021-23436-x). However, power sweeps and maximum output signals are very underreported in graphene based devices. To shed some light on the topic we summarized some of the PTE literature in the new table which we add to the Supplementary Information 8:

An overview of the state of the art of different graphene based PTE photodetectors is provided in the Table S1. The table lists the graphene fabrication technology (exfoliated vs. photonic integrated circuit), the operation wavelength, the voltage responsivity, the maximum reported photovoltage and the bandwidth.

Table S1: Comparison of graphene based PTE photodetectors sorted by publication year. “Fab” stands for the fabrication method of the graphene sheet (Ex. - mechanical exfoliation, CVD - chemical vapor deposition). “Int.” stands for integration scheme (PIC – photonic integrated circuit, FS – free space illuminated). λ is the illumination wavelength in nanometer. \mathcal{R}_V is the reported voltage responsivity. V_{max} is the maximum reported photovoltage signal. The last column summarizes the reported bandwidths (BW) of the devices. The larger than “>” symbol denotes setup limited bandwidths.

Ref.	Year	Fab	Int.	λ (nm)	\mathcal{R}_V	V_{max}	BW
/10.1038/nnano.2014.182	2014	Ex.	FS	100'000	715 V/W	0.0027 mV	3.2 GHz
/10.1021/acs.nanolett.6b03374	2016	Ex.	PIC	1550	3.5 V/W	---	65 GHz
/10.1021/acsp Photonics.8b01128	2018	Ex.	PIC	1550	4.7 V/W	1.5 mV no power sweep	>18 GHz
/10.1021/acs.nanolett.9b02238	2019	CVD	PIC	1550	12.2 V/W	2.9 mV	42 GHz
/10.1038/s41467-020-20115-1	2020	Ex.	FS	4000	27 V/W	0.35 mV	>4 kHz
/10.1021/acsnano.0c02738	2020	CVD, monocryst.	PIC	1550	6 V/W	4 mV no power sweep	>67 GHz
/10.1038/s41467-021-21137-z	2021	CVD, monocryst.	PIC	1550	3.5 V/W	15.1 mV loss of linearity	70 GHz
/10.1038/s41467-021-23436-x	2021	Ex.	PIC	1550	90 V/W	38.6 mV loss of linearity	12 GHz
/10.1038/s41566-021-00819-6	2021	Ex.	FS	4000	15.6 V/W	6.8 mV	0.52 MHz
/10.1038/s41467-022-31607-7	2022	CVD	PIC	5200	1.5 V/W	0.015 mV no power sweep	> 1 MHz
/10.1038/s41566-022-01115-7	2023	Ex.	FS	4000	392 V/W	3 mV	> 0.39 MHz
This work	2024	CVD	FS	1550	1.6 V/W	18.5 mV loss of linearity	420 GHz

With this table and the above new measurements which we provided with respect to reviewer's comment #4, there is evidence that optical inputs beyond several mW of optical power are possible. Evidence that higher saturation powers are possible even with higher responsivity can be seen by the maximum measured signals. Here we measured $> 5 \mu A$ limited by the low current responsivity, but $> 18 mV$ in photovoltage output – the 2nd highest reported value. In 10.1126/science.adg8017 we've demonstrated $\sim 50 \mu A$ output signal with a free-space illuminated graphene PD. Recently we measured generated photocurrents $> 100 \mu A$ in free-space illuminated graphene devices. However, these results are not yet published.

As the PD can be seen as a distribution of smaller PDs over an area, there is also the possibility to increase the active area of the device by adding more of these elements. This in turn would again allow higher optical input powers as the power is spread across a larger area. However, how the responsivity and RF response is influenced by the scaling is not yet clear. Depending on the application there might ultimately be trade-offs between RF bandwidth, responsivity, sensitivity and saturation. Further research on the topic is required.

6.1	Having said that application relevant performances are in my opinion not considered in the main text and are revealed to be low in the supplementary information (even taking into account the list of improvements in the supplementary file), application relevant concepts and functionalities enabled by this graphene detector could be sufficient to justify a novelty. Nevertheless, the proposed concepts and functionalities are not peculiar to this device and the arguments of the authors about this aspect are quite weak. -Authors claim that their device supports operations in a broadband optical range. They mention many times the MIR window. Moreover, they assert that other solutions like, e.g., waveguide-integrated devices would not afford such “broadband” operation. First, the paper does not show any MIR operation, apart from a supplementary section showing a simulation. Second, the operation of the proposed device is wavelength dependent as for integrated photonic devices based on, e.g., silicon nitride or Si, which design can be optimized to operate at a central wavelength, with optical bandwidths chosen to trade-off efficiency. Integrated photonic waveguides can operate in ranges even larger than those experimentally considered by the authors, in many cases with very low design efforts. Moreover, the wavelength tuneability of the proposed device requires lithographic accuracies in the order of tens of nm, which is not exactly of easy implementation using standard photonic technology. Indeed, the real device central wavelength operation is different from the designed one, as also observed by the authors. Therefore, I really do not see the author’s point in this claim. At least, I don’t see why integrated solutions are weaker than the proposed one.
	We thank the reviewer for his comments on the applications and wavelength operation range. The MIR operation is indeed only showed in simulation in our work. At the time of writing, we had no access to a MIR light source and did therefore not fabricate any active metamaterial devices for the MIR range. However, in the introduction we refer to many other works that showed detection in the MIR as well as our own previous work, which is based on the same metamaterial perfect absorber material stack. We rewrote a sentence in the introduction that might give the impression that we show experimental MIR operation of the photodetector:
	p2, l32: “Here in this work, we investigate the photothermoelectric induced directional currents (PTE-DC) within a metamaterial perfect absorber (MPA) architecture for its viability to serve as infrared photodetector-both the needs of the NIR and MIR detection regime and thereby uncover its quasi-instantaneous frequency response.”
	The MIR operation that we do discuss is two-fold; (1) the Dirac cone band structure of graphene which allows broadband operation and (2) the top illuminated metamaterial architecture. The reviewer argues that there is no real advantage in aspect (2) as photonic integrated circuit (PIC) approaches could also offer broadband operation. From our view, the design of a PIC platform allowing for single mode operation over a broad wavelength range is not trivial and does not come with low design efforts. First, we note that the discussion does not include all platforms:  - InP: the SOTA platform for high-speed photodetector technology as it allows for the high quality growth of III-V materials. Here, the wavelength range is limited to a window of ~1000 to 2000 nm. Furthermore, the substrate itself is expensive and scalability of fabrication processes is difficult. They are therefore excluded from the further discussion. - SiN and Si. These platforms can indeed also operate within the MIR regime. To the point of the design effort: Si and SiN PICs have been shown to operate in the MIR range, but the design efforts are considerable. As an example, typical SOI platforms used for NIR applications have Si thicknesses of 220 nm. Such thicknesses are inefficient for MIR operation as optical modes

	extend largely into the buried oxide in these types of stacks. The oxides induce large losses. One solution is to change the material thicknesses which in turn again limits the spectral range achievable on a single substrate. A good perspective on the topic can be found in 10.1515/nanoph-2017-0085. Other approaches without changing the material layer stack include photonic crystal structures (10.1364/OE.448284). Again, here the design effort is considerable, and the fabrication is lithographically heavy. This discussion further only covers the waveguide itself; additional passives such as couplers, bends, splitters, etc. also require an appropriate design for each target wavelength that they would be used. Each element also introduces losses until the light is guided to the active photodetector region. We want to highlight that we do not make any changes to our layer stack, not in the material composition nor the material thicknesses. The only element that is changed is the resonator shape. We therefore strongly believe that our approach is far easier to tune to a target operation wavelength than a PIC-based photodetector. Regarding the reviewer's comment on the lithographic accuracy: This aspect is true that we did not hit the targeted resonance wavelength in this first iteration. All our active devices have only been designed with one resonator shape to allow us to perform more comparative measurements on properties such as the resonator orientation. We note however that we are only 70 nm off from the target (1480 nm instead of 1550 nm) and that the resonance offers a broad peak with a FWHM of ~200 nm, still allowing us to operate the device with only little deficit at 1550 nm. Furthermore, adapting the simulation framework to include more realistic shapes (e.g., rounded corners) that we can now extract post-fabrication, we strongly believe that we can improve the accuracy of the metamaterial design even further in future works. We highlight that Fig. 2 illustrates the successful tunability of the structure and Fig. S10 shows that a size change of 14 nm already yields a very good overlap between measurement and simulation. It has been further shown that nanometre scale metamaterial structures can be efficiently and accurately fabricated with nanoimprint lithography if mass production is a target, e.g., discussed also in this recent review and perspective 10.1021/acsphotonics.3c00457. However, this is clearly beyond the scope of this paper.
6.2	-The discussion of SOTA on polarization/ wavelength demultiplexing is not present in the text. How does their polarization /wavelength demultiplexing compare with already existing systems in terms of efficiency and compactness (also considering the space occupied by the active polarization control)? Polarization/wavelength demultiplexing is a function which can be easily performed not only for two wavelengths. It is implemented for many wavelengths in both CWDM a DWDM systems, also using integrated photonics. ...
	We thank the reviewer on his comment of the demultiplexing application. We first want to point out that the demonstrated device combines the two functionalities (wavelength division demultiplexing and polarization demultiplexing) simply as a proof-of-concept device. The goal of the device is foremost to show that more complex metamaterial unit cells can be employed and do not perturb the overall effect, and in addition provide access to advanced functionalities. We do not claim – or even target – that we would outperform any other wavelength or polarization demultiplexing schemes at this time. We made the following change to the manuscript:
	p10, l47: "We now want to advance the use of the PTE-DC further by showing a proof-of-concept on how two independent detectors can coexist within the same active area."

6.3	... In addition, the scheme proposed by the authors is sensitive to polarization when “demultiplexing” two wavelengths, and sensitive to wavelength when detecting polarization. Moreover, active polarization control is needed. What is the application for such a detector? ...
	We further agree with the reviewer that the combination of both mechanisms in the same device does not lead to any direct advantage. However, we note that there is the possibility to also build devices that you can do either of the two things.  - (1) Devices that work for a single wavelength and can detect two signals with different polarizations or - (2) Devices that work for two different wavelengths but are otherwise polarization independent. Number (1) is trivial as one can simply make all resonators identical. Number (2) requires a completely different metamaterial resonator geometry to achieve the polarization independent, wavelength directed photocurrents. We do have theoretical and simulation work that shows that it is possible. However, these preliminary results are beyond the scope of this paper. The applications of these two types of detectors are for example linear polarization state detection or sensing of two different wavelengths at the same time with a single device which can for example be beneficial in environmental sensing applications such as gas detection. Additionally, the high speed of the devices could be used in telecommunication applications. With illumination from a single fiber, it is possible to have essentially two detectors occupying the same space. As such, the bandwidth per area can effectively be doubled. Here either wavelength division demultiplexing or polarization demultiplexing are possible schemes. We note here further that the possibility of the wavelength tunability of the resonators gives access to wider spectral bands, e.g., around 2000 nm which are foreseen to be used in future telecommunication applications.
6.4	... The “small” crosstalk of 3% is 15 dB, which is in my opinion not interesting for applications, where >> 20 dB isolation between the channels are required in systems in which many wavelengths are managed. How can this be solved, also accounting that real applications require >> two wavelengths? How the responsivity and wavelength selectivity scale when considering more than two wavelengths?...
	We thank the reviewer for his comment on the crosstalk and the WDM application. We agree with the reviewer that the crosstalk would ideally be -20 dB or lower. We believe parts of the crosstalk are the relatively broad resonances of the metamaterial structures and the contacting scheme that we employed. With the peaks showing FWHM values of ~200 nm, there is an overlap in the absorption spectra of the two resonators. Ideally one would choose the peaks further apart to show the true crosstalk of the device. However, we did not have any other laser sources available apart from the O- and C-band lasers which is the reason why we chose these specific wavelengths. Secondly, the contacts were designed to be only one period away from the metamaterial structures. Therefore, we have large sections of contact doped graphene at the contact edges which will also contribute to the overall photoresponse which leads to an additional crosstalk. On the question of additional wavelengths: as the device encodes wavelength on a spatial direction in a 2D plane, there is only the possibility to truly support two wavelengths. However, due to the small

	size and direct top illumination it is of course possible to have space division multiplexing (SDM) of the devices, effectively halving the amount of space required as two photodetectors are built into a single device. If one is interested in simply differentiating between a set of different wavelengths, it is possible to design a polarization independent variant that has four distinct resonances. Then, each λ is encoded in the four spatial directions $+x, -x, +y, -y$. In such a scenario a mixture of wavelengths will however lead to currents that cancel each other out.
6.5	...What is the problem that this scheme could solve and that is not already solved with existing systems?
	As discussed above, there is a variety of applications that could benefit from such a scheme. To achieve the same functionality, one would always require several photodetectors and additional elements. E.g., in free-space illumination the optical signal will need to be split or dispersed to illuminate each PD with the specific functionality. In a PIC approach the same is true with passive elements such as polarization beam splitters or wavelength division demultiplexing schemes. We add the following to the conclusion in the main text to put a stronger focus on potential applications:
	p12, l40: "We further leveraged the directionality of the induced currents by introducing a four contact pad design with a supercell metamaterial. This design highlights the flexibility of the PTE-DC. Direct access to demultiplexing schemes is enabled due to the two perpendicular channels. Furthermore, the flexibility in the metamaterial design and graphene's absorption characteristics enable tunability over a large spectral range. The scheme offers thereby the potential of operation in the MIR spectral range where high-speed detectors do not exist. Additionally, they offer the flexibility to provide built-in functionalities such as polarization state detection, dual wavelength detection and two signal demultiplexing within the same active area removing the need to split-up signals or for bulky constructions. Further, the results clearly highlight that future endeavors on graphene photodetectors should not simply focus on the optimization of the absorption but have to rely on a multidisciplinary modelling and design approach. Therefore, photothermoelectric induced directional photocurrents in graphene are an excellent mechanism to further the understanding in the material, physics, and device design, with the promise of strong performing highest-speed photodetectors."
6.6.	In conclusion, my general opinion is that this manuscript shows a work based on known fundamental phenomena. This fact by itself would not be a point for negatively judging the paper, if new application-relevant novelties or ideas would have been shown. But from this standpoint, it presents just a well performed engineering exercise (which certainly could be suggestive for further developments), with no sufficiently relevant application perspectives.
	We thank the reviewer for his detailed comments above and his concluding remarks. We appreciate the positive assessment describing the manuscript as very clear, instructive, and well written. We also value that our engineering work is being complimented. We strongly believe that bringing known fundamental phenomena into a functioning device structure should not be underestimated and can be considered as novelty on its own. A first demonstration of a PTE photodetector with more than 400 GHz bandwidth is worth reporting on as no other technology platform was able to achieve such numbers. Following the above discussions, we also added the relevant applications that could benefit from such a device. While at the moment the device responsivity is still lacking, we hope the reviewer can appreciate that this is in essence a first demonstration of a PTE photodetector with 100s of GHz bandwidth and that there is room to enhance the responsivity. We note that we did not only improve upon the speed of previous demonstrations, but we also introduced a design guideline to include the asymmetric

	resonators in a passivated metamaterial perfect absorber structure. This allows for operation with low gate voltages (CMOS compatible sub 5V compared to 80-120V of the previous reports) and also stable operation in ambient condition. With the promise of additional functionalities, we strongly believe that this type of high-speed device architecture will be useful. We answer below the additional points the reviewer added:
7.	Authors should put the real image of the RF device with the dimensions.
	We thank the reviewer for pointing out the lack of an image of the RF devices. We're of course happy to add an image of a device. We assume this comment is related to the RC-cut off in the next comment and will add the image in the discussion below.
8.	As far as I understood, the device length is 10 μm. Apart from transit time and quasi-instantaneous response of PTE, one needs to consider the pads capacitance which is most probably responsible for the frequency cut-off. Considering that the resistance of the device is 6000 ohms, I get ~5fF parasitic pads capacitance to get ~450 GHz, which is impressively good! Maybe the authors could add information about that, and about the RF pads design. Also, information about the substrate resistivity should be included to show that no RF ohmic losses occur due to substrate.
	We thank the reviewer for his excited comment about the RF pad design and RC cut-off. First, we comment on the reviewer's calculated capacitance. It is important to consider the full electrical circuit when operating the device in a high-speed application.  The photodetector resistance is essentially in parallel to the load resistance. In a high frequency circuit this corresponds to 50 Ω. The relevant resistance for an RC cut-off of the system is thereby $\left(\frac{1}{6000 \Omega} + \frac{1}{50 \Omega}\right)^{-1} \approx 50 \Omega$. Further, the RF pads are a travelling-wave design which makes the lumped capacitance of the pads not directly relevant. To discuss the RF pad design, we added a new section in the supplementary information that discusses this in more detail. It reads as follows:
	Main text, p9, I3: "Here, we trim a device for high-speed. We have used RF simulations to find an architecture that does not limit the bandwidth of the device but ideally reveals potential speed-limitations of the photodetection effect (see also Supplementary Note 9)."

Supplementary Note 9: Discussion on Frequency Response and RC cut-off

Fig. S17: Radio frequency pad design.

a Top view of the full device geometry implemented in the simulation environment (CST Studio). The contact pads (gray squares) are connected to an optimized co-planar waveguide section. The active area of the device is represented in red, which is implemented as a lumped element. **b** Corresponding optical microscope image of a fabricated device. **c** Side view of the layer stack within the simulation environment. The zoom-in shows the active region layer stack. The substrate is silicon (intrinsic) with a $1\mu\text{m}$ thick thermally grown silicon oxide layer. Within the silicon oxide the gold gate is buried. A layer of aluminum oxide is on top followed by the active area of the device, and the ground and signal lines. A 50 nm aluminum oxide passivation layer is added on top. The structure is surrounded by air. **d** Simulated RF response from 0 to 500 GHz for the above depicted geometry. No roll-off behaviour is found.

To ensure ideal operation at high frequencies we optimized the contact pad design. Fig. S17 illustrates the design. We implement the structure in CST Studio and perform time domain simulations. The RF pad design is illustrated in Fig. S17a. Three contact pads (gray squares) modelled as perfect electric conductors are connected to the co-planar waveguides (CPW) forming the Ground-Signal-Ground (GSG) structure. The CPW was optimized in a first step to match the wave impedance of the contact structure to the $50\ \Omega$ of the RF circuit. For this, only the CPW structure connected terminated with two ports have been simulated separately. The active area of the device was fixed at $10\ \mu\text{m}$ as well as the pitch of the pads which was set to $50\ \mu\text{m}$. Therefore, the parameters to optimize were the widths of the metal lines, the gaps between the lines and the length of the CPW. These parameters were optimized until the line impedance was close to $50\ \Omega$. We note that the material layer stack illustrated in Fig. S17c is determined by the metamaterial design as illustrated in Fig. S8.

Using the optimized CPW geometry we simulated the full device as illustrated in Fig. S17a. The active area of the device is here simply approximated as a lumped element with a resistance of $6000\ \Omega$. At the same spot a port for the excitation signal is implemented mimicking the photoresponse of the PD. The transmitted and received signal is then detected at the gray GSG pads which are terminated with $50\ \Omega$. The resulting simulated RF response is shown in Fig. S17d. No roll-off behaviour is found in the range from 0 to 500 GHz. Oscillation in the spectrum stem from small impedance mismatches between the active area and the probing. Strong drops in the frequency range have been identified

	as resonant coupling to the gate pad. In the measurements we did not experience these sharp drops which we assume stems from damping in the gate due to resistive losses.
9.	Given the low power level of the detector, authors should add more information about the noise of the equipment they use, and the calibration of the same. How did the authors calibrate the measurement? Did they use reference signals generated by, e.g., commercial UTCs with known frequency response? The information about the measurement is, in general, not exhaustive, given the large, spanned bandwidth and the different components used in the different frequency ranges.
	We thank the reviewer for his remark on the high frequency characterization methods. As discussed in the paper, we essentially use two different setups to characterize the frequency response of the device. (1) In the range of 3-110 GHz directly with an electrical spectrum analyzer (ESA) and (2) with additional subharmonic mixers in the bands from 110-170, 220-330 and 330-500 GHz. For the calibration of the system:  - (1) The path here consists of a GSG-probe, a bias tee and a RF cable. We calibrate the probe losses with the calibration curve provided by the vendor. The bias tee and RF cable losses are compensated by measuring a high-speed reference PD once directly attached to the ESA and once with the additional components in the RF path. - (2) The path here consists of a GSG-probe directly attached to a subharmonic mixer. The probe losses are again compensated with the loss calibration provided by the vendor. The loss or respective conversion efficiencies of the mixers are compensated with the calibration data provided by the vendor. The noise floor strongly depends on the frequency range of interest but is with our equipment and settings typically around or below -90 dBm. As an example, the noise floor at 400 GHz is -94 dBm and the measured RF power of the device is -74 dBm, leaving us a 20 dB span above the noise floor. We expanded the discussion on the high frequency characterization in the method section to provide more details: High frequency characterization: To characterize the RF frequency response of the device we used a laser beating setup as depicted in Fig. 5b. Two lasers are combined in a single fiber where one laser is fixed at 1550 nm and the other is detuned by a difference frequency $\Delta\lambda_{RF}$. The beat-note at $\Delta\lambda_{RF}$ was measured by capturing the device response with an electrical spectrum analyzer (ESA). The ESA (Keysight UXA N9041B) offers a frequency range up to 110 GHz. The device was connected to the ESA with high-frequency GSG-probes, a bias tee and an RF cable. The probe losses are compensated with the loss calibration provided by the vendor. The bias tee and RF cable losses have been calibrated by performing reference measurements using a commercial high-speed photodetector. Frequencies above 110 GHz were accessed by employing sub-harmonic mixers (Virginia Diodes, SAX modules) connected to the ESA. The high-frequency probe losses are again compensated with the loss calibration provided by the vendor. The loss or respective conversion efficiencies of the sub-harmonic mixers are compensated with the calibration data provided by the vendor.
10.1	Apart from the “multiplication” approach discussed in point 5, that is in my opinion not appropriate, I have some concerns on the observations done in the calculation: -Metal on top of graphene pins the chemical potential but can also degrade graphene quality and so its mobility. Maybe low mobility graphene would not be strongly affected from metal decoration, but high quality could, since high quality graphene is more sensitive to defects. Considering this, are authors sure that higher quality graphene would give 3x mobility if metal is put on top?

	We thank the reviewer for his comment on the metal pinning. This is indeed a very interesting point. We first want to note that we did not make any comments on what mobility our device could reach. We merely argued that with our relative low mobility of $< 1000 \text{ cm}^2/\text{Vs}$ a moderate improvement in the mobility of factor 5x would lead to an increase in the Seebeck coefficient of a factor 3x. The work we compare ourselves to in this estimation does not report any mobility values and therefore we chose a relatively low mobility value for high quality graphene. We are aware of the negative effect of metal contacts on the carrier mobility in graphene. However, there are studies that demonstrate that even with high impurity concentrations it is possible to achieve several $1000 \text{ cm}^2/\text{Vs}$ in mobility, e.g., (10.1038/s42005-021-00518-2). Therefore, we strongly believe that this improvement in Seebeck coefficient is achievable.
10.2	- Concerning resistance by itself, why does decreasing resistance by 5 means x5 responsivity? Putting low impedance detectors in parallel does not necessarily mean increasing responsivity, this can be easily checked using simple equivalent circuits. Maybe authors refer to the increased electrical power transfer from the detector to the load, which would rise because of impedance matching? Indeed, the electrical power transfer would rise by $> 30 \text{ dB}$ while passing from 6 kohm to 50 ohm. Anyway, this is not related to (DC) responsivity in my opinion but only in power transfer efficiency from the detector.
	We thank the reviewer for his question on the link between resistance and responsivity. As PTE based devices are typically seen as generating a photovoltage and not a photocurrent the device resistance directly also influences the responsivity in terms of photocurrent. Photovoltages generated by the Seebeck effect are linked to thermoelectric currents across a PD by Ohms law: $I_{ph} = V_{ph}/R$ or $\mathcal{R}_A = \mathcal{R}_V/R$. This can be seen in various examples in literature:  - $\mathcal{R}_V = 27 \text{ V/W}$ or $\mathcal{R}_A = 36.3 \text{ mA/W}$ with a device resistance of $500 - 1000 \Omega$ (10.1038/s41467-020-20115-1) - $\mathcal{R}_V = 3.5 \text{ V/W}$ or $\mathcal{R}_A = 35 \text{ mA/W}$ with a device resistance of $\sim 110 \Omega$ (10.1021/acs.nanolett.6b03374) - Split-gate PTE configuration where ideal voltage responsivity (high resistance) and ideal current responsivity (low resistance) is identified as a function of two gate voltages. (10.1038/s41467-022-31607-7) To clarify this point, we expanded Supplementary Note 8 and added the new Figure S15 showing the above mentioned new recorded power sweep where we measure once the photovoltage and once the photocurrent:
	SI, p13, I23: Lastly, the resistance difference between our device and the previous report is a factor 5x. As photovoltages generated by the Seebeck effect are linked to thermoelectric currents across a PD by Ohms law the resistance has a direct influence on the current responsivity. This effect is seen also in Fig. S15; The voltage responsivity divided by the current responsivity leads to $\sim 3000 \Omega$ matching the device resistance in the operation point.

We want to thank the reviewer again for his very detailed comments. By addressing all the reviewer's comments and making the according changes to the manuscript including additional measurements the manuscript is definitely improved. The novelty aspect is now also highlighted stronger.

Reviewer 2

Below are detailed responses to each of the reviewer's comments (*italic gray*), highlighted in **blue** for clarity. Additionally, alterations made to the initial manuscript are clearly marked in **red** which are directly added below each comment.

“Koepfli et al report zero-bias graphene photodetectors that feature centrosymmetry-broken nanostructures and vectorial photoresponse. While the design concept of “T”-shaped structures has been reported by another group [Nat Commun 11, 6404 (2020)] as acknowledged by the authors, this work further studies mechanisms by examining the frequency response of the devices up to 500 GHz. They revealed that the photoresponse based on bolometric and photoconductive effects would vanish above about 10 GHz, and the hot carriers assisted photothermoelectric effect allows an ultrahigh 3dB cutoff frequency of 420 GHz, which is far faster than the state-of-the-art photodetectors (usually below 100 GHz, even after integrating with waveguides. See Nat. Nanotechnol. 15, 118–124, 2020). Besides, the authors use CVD-grown graphene rather than exfoliated graphene, representing a big step towards the practical applications of this new type of device, although the responsivity is still two orders smaller (200-300 $\mu\text{A/W}$) than exfoliated ones ($> 30 \text{ mA/W}$). Overall, I think this work reports exciting and promising results for ultrahigh-speed photodetectors and also deepens our understanding of the underlying mechanisms, so I would highly recommend the publication on Nature Communications if the authors can address my following concerns:”

We thank the reviewer for this extensive assessment of our work. We provide below the answer to the detailed comments.

1.	“Title. An expression like “highest-speed” sounds unprofessional. I would recommend using an explicit number.”
	“Comparison with other works. The authors have just reported a similar work based on asymmetric contact materials which shows a response speed of 500 GHz [Koepfli, S. M. et al. Metamaterial graphene photodetector with bandwidth exceeding 500 gigahertz. Science 380, 1169–1174 (2023)]. So, I don't think this work shows the “highest” speed.”
	We thank the reviewer for his comment on the paper title. Indeed, the word “highest” might not have been the best choice. It was intended to reference to the photothermoelectric part of the title, in the sense of “highest-speed PTE” and not in the sense of “highest-speed photodetector”. We also agree that it might age poorly. We therefore change the paper title to:
	Controlling Photothermoelectric Directional Photocurrents in Graphene with over 400 GHz Bandwidth
2.	“Frequency-dependent contribution of different mechanisms. In Figure 5a, the authors provide the measured frequency response at $V_b = V_g = 0 \text{ V}$, where PTE dominates. By looking at Figure 5c-h, it seems PC and Bol will be largely suppressed $> 10 \text{ GHz}$. My question is, is possible to extract the respective cut-off frequencies for the PC, Bol effects also? For example, in low-frequency or even DC range? What are the percentages for each mechanism? Besides, the authors report that gate voltage also affects the speed. I would recommend the authors provide more details about the microscopic mechanisms.”
	We thank the reviewer for the interesting questions on the frequency bandwidth of the different detection mechanism. To estimate the cut-off frequency from the BOL effect we extract the top left corner points from Figures 5c-h. At this point we have the strongest BOL response and are furthest away from the

other effects. We plot these data points in a new supplementary figure Fig. S18 (see below). Assuming the BOL effect has the same roll-off behaviour as the PTE-DC (i.e., 60 dB/dec), we estimate from these data points a 3 dB cut-off frequency between 250 and 330 GHz.

From the Figure 5c-h it is difficult to make a clear assessment of the cut-off frequency of the PC mechanism as there are not sufficient data points and the PTE-DC effect is strongly overlapping with the PC effect. Therefore, we are not able to make an estimate on the cut-off frequency of the PC effect.

p10, l26: “When moving from an almost DC signal to 10 GHz (Fig. 5e), the regions previously associated with the PC effect are disappearing. Moving to 300 GHz (Fig. 5f) shows that the BOL effect (corners of the RF response map) is also weakening. At 400 GHz (Fig. 5g) the corner show almost no more features. We estimate a 3 dB cut-off frequency of the BOL effect in the range of 250 to 330 GHz (see Supplementary Information). Finally, at 500 GHz (Fig. 5h), where we are measuring in the roll-off of the response, the device performance is essentially bias independent except for the slope corresponding to the shift of the reference ground. These measurement results indicate a quasi-instantaneous Shockley-Ramo response for this device architecture of the PTE-DC, whereas the PC and BOL effects are limited to lower frequencies. The measurements thereby provide speed limitations of detection mechanisms within graphene and are not hindered by RC-limits or setup limitations. Thus far only one graphene PD architecture consisting of interdigitated electrodes exploiting the photovoltaic effect was able to exceed the frequency response presented here³⁶.”

Supplementary Note 9: Discussion on Frequency Response and RC cut-off

To estimate the cut-off frequency of the bolometric (BOL) effect we extract the top left corner points from Figures 5c-h of the main text. At this operation point we have the strongest BOL response and are furthest away from the other effects. These data points are shown in Fig. S18. Assuming the BOL effect has the same roll-off behaviour as the PTE-DC (i.e., 60 dB/dec), we estimate from these data points a 3 dB cut-off frequency between 250 and 330 GHz.

Fig. S18: 3 dB cut-off frequency estimation of the bolometric effect. Green point correspond to photothermoelectric induced directional currents (PTE-DC) with 0 V bias and -0.6 V gate voltage with respect to the Dirac point position (corresponding to the measurement shown in Fig. 5). A roll-off with -60 dB/dec is found. Orange points correspond to bolometric (BOL) operation with -0.75 V bias and 2 V gate voltage with respect to the Dirac point position. Assuming the same roll-off slope a 3 dB cut-off frequency between 250 and 330 GHz is estimated.

4. “Another question to consider: what is the special design in their characterization setup? While the response speed of graphene has been extensively measured in numerous studies, the majority of these measurements have been limited to frequencies below 40 GHz. How did the authors ensure the accuracy and reliability of their measurements?”

	We thank the reviewer for the interest in our measurement setup. Our research group has over a decade of experience in high-frequency characterization of electro-optic components. With this, we accumulated an extensive arsenal of equipment targeted specifically for this purpose. Furthermore, we have also developed over the years the corresponding calibration routines to ensure proper measurement results. Additionally, we believe that our effective passivation of the graphene is an additional key property that allows us to have such detailed measurements where we can test a device with multiple operation conditions at different frequencies without having an altered device performance. So far, we experienced no device degradation in air during measurements. Furthermore, setting the working point of the graphene with the gate voltage is also a mechanism that does show very little to no drift. This allows us to take detailed measurements that require long measurements time without any issues in loss of device performance. We expanded the discussion on the high frequency characterization in the method section to provide more details: High frequency characterization: To characterize the RF frequency response of the device we used a laser beating setup as depicted in Fig. 5b. Two lasers are combined in a single fiber where one laser is fixed at 1550 nm and the other is detuned by a difference frequency $\Delta\lambda_{RF}$. The beat-note at $\Delta\lambda_{RF}$ was measured by capturing the device response with an electrical spectrum analyzer (ESA). The ESA (Keysight UXA N9041B) offers a frequency range up to 110 GHz. The device was connected to the ESA with high-frequency GSG-probes, a bias tee and a RF cable. The probe losses are compensated with the loss calibration provided by the vendor. The bias tee and RF cable losses have been calibrated by performing reference measurements using a commercial high-speed photodetector. Frequencies above 110 GHz were accessed by employing sub-harmonic mixers (Virginia Diodes, SAX modules) connected to the ESA. The high-frequency probe losses are again compensated with the loss calibration provided by the vendor. The loss or respective conversion efficiencies of the sub-harmonic mixers are compensated with the calibration data provided by the vendor.
5.	“What limits the 3dB frequency? As the authors have demonstrated in a recent Science paper, similar architecture can reach 500 GHz. What is the factor to limit this work to 420 GHz then?”
	We thank the reviewer for their interesting question on the frequency response limit. As discussed within the paper, the cut-off can either stem from a (1) carrier dynamic related time constant or a (2) resistance-capacitance (RC) limit. We believe that the 420 GHz cut-off stems from the carrier dynamic aspect. (1) Regarding the carrier dynamics: We discuss the carrier dynamic relaxation by citing references 50, 55 and 56 that are able to measure high speed carrier dynamics within graphene with pump-probe measurements. Regrettably, we do not have the capabilities to perform these types of measurements for our specific samples that would offer additional insights. However, we see from reference 56 that the graphene quality and properties are crucial to reach high speeds. It’s important to note that in comparison to the Science paper, the device here is operating with a different detection mechanism. We argue that the Science paper reaches such high speeds because of the optimized carrier transit time, whereas here we make use of the fast relaxation time dynamics. (2) Regarding the RC limit: This is a crucial part to design high-speed PDs in general. Classical pin photodiodes suffered a clear trade-off between responsivity and frequency response, e.g., see 10.1017/CBO9780511635595. This classical view is here however not applicable. Therefore, we simulated the RC cut-off of the device by full 3D simulation of the layer stack and optimized the

overall gate and contact pad geometry. We added the following section in the SI that discusses this in more detail:

Main text, p9, l3: “Here, we trim a device for high-speed. We have used RF simulations to find an architecture that does not limit the bandwidth of the device but ideally reveals potential speed-limitations of the effect (see also Supplementary Note 9).”

Supplementary Note 9: Discussion on Frequency Response and RC cut-off

Fig. S17: Radio frequency pad design.

a Top view of the full device geometry implemented in the simulation environment (CST Studio). The contact pads (gray squares) are connected to an optimized co-planar waveguide section. The active area of the device is represented in red, which is implemented as a lumped element. **b** Corresponding optical microscope image of a fabricated device. **c** Side view of the layer stack within the simulation environment. The zoom-in shows the active region layer stack. The substrate is silicon (intrinsic) with a 1 μ m thick thermally grown silicon oxide layer. Within the silicon oxide the gold gate is buried. A layer of aluminum oxide is on top followed by the active area of the device, and the ground and signal lines. A 50 nm aluminum oxide passivation layer is added on top. The structure is surrounded by air. **d** Simulated RF response from 0 to 500 GHz for the above depicted geometry. No roll-off behaviour is found.

To ensure ideal operation at high frequencies we optimized the contact pad design. Fig. S17 illustrates the design. We implement the structure in CST Studio and perform time domain simulations. The RF pad design is illustrated in Fig. S17a. Three contact pads (gray squares) modelled as perfect electric conductors are connected to the co-planar waveguides (CPW) forming the Ground-Signal-Ground (GSG) structure. The CPW was optimized in a first step to match the wave impedance of the contact structure to the 50 Ω of the RF circuit. For this, only the CPW structure connected terminated with two ports have been simulated separately. The active area of the device was fixed at 10 μ m as well as the pitch of the pads which was set to 50 μ m. Therefore, the parameters to optimize were the widths of the metal lines, the gaps between the lines and the length of the CPW. These parameters were optimized until the line impedance was close to 50 Ω . We note that the material layer stack illustrated in Fig. S17c is determined by the metamaterial design as illustrated in Fig. S8.

	Using the optimized CPW geometry we simulated the full device as illustrated in Fig. S17a. The active area of the device is here simply approximated as a lumped element with a resistance of 6000 Ω. At the same spot a port for the excitation signal is implemented mimicking the photoresponse of the PD. The transmitted and received signal is then detected at the gray GSG pads which are terminated with 50 Ω. The resulting simulated RF response is shown in Fig. S17d. No roll-off behaviour is found in the range from 0 to 500 GHz. Oscillation in the spectrum stem from non-well matched impedances between the active area and the probing. Strong drops in the frequency range have been identified as resonant coupling to the gate pad. In the measurements we did not experience these sharp drops which we assume stems from damping in the gate due to resistive losses.
6.	"I suggest that the authors include time response measurements, such as eye patterns, as complementary data to their frequency response analysis. This additional data would strengthen their assertion of a > 400 GHz 3dB cut-off frequency."
	We thank the reviewer for their comment on additional frequency response measurement data. Due to the low responsivity of the device, we're at the moment not able to record any meaningful time traces with a high-speed scope. We are however continuing our work to demonstrate that PTE-DC devices can also be used in data transmission experiments with high data rates which we would report in future iterations. We further note that the fastest available scope offers an analog 3 dB bandwidth of 113 GHz and the fastest waveform generators have 3dB analog bandwidths of 80 GHz. So even with the best available equipment there is no possibility to measure time traces close to the speed limit of the photodetectors. With the current responsivity, the signal quality measured with such a high-speed scope will not lead to any meaningful results. The measurement technique with the ESA is far more accurate to characterize the true frequency response of the device.
7.	"Figure 3b caption: What do the authors mean by "The Dirac point is found close to 0 V at 0.6 V"?"
	We thank the reviewer for pointing out that this sentence in the figure caption is unclear. We wanted to say that the measured Dirac point of the device is at 0.6 V, which is in our opinion a low set voltage to reach the Dirac point compared to many other works. Hence, the wording "close to 0 V". We changed it accordingly to make it clearer:
	"b Resistance of the device as a function of gate voltage (back- and forth sweep). The Dirac point is found at close to 0 V 0.6 V, showing low doping and making the Dirac point accessible with sub-Volt gating."
8.	"Figure 3g bottom panel: 180o should be a typo? It should be (0o & 90o)."
	We thank the reviewer for pointing out that the figure caption and figure have a mistake. We fixed the labels and caption accordingly.
9.	"Figure 3h: The unit of absorption is missing."
	We thank the reviewer for pointing out that the units are unclear. Absorption in this case is total absorption, i.e., without a unit. To clarify this point, we change the units to (%) and changed the y-ticks accordingly.

Reviewer 3

Below are detailed responses to each of the reviewer’s comments (*italic gray*), highlighted in blue for clarity. Additionally, alterations made to the initial manuscript are clearly marked in red which are directly added below each comment.

“The authors Stefan et al. report the highest photothermoelectric response of metamaterial integrated graphene photodetector. The ultrahigh frequency response over 400 GHz is impressive, which is supported by the quasi-instantaneous response mechanism. On the other hand, the photothermoelectric response mechanism, as well as other response mechanisms were also studied systematically. There are some concerns and questions should be addressed before considering publish in Nature Communications. Please see following comments.”

We thank the reviewer for their overall assessment of our work. Our answers to the following comments below:

1.	“Could the author simulate the typical absorption in graphene, but not the total absorption?”
	We thank the reviewer for pointing out the lack of this property in the manuscript. While we cannot measure the absorption within graphene, we do have simulation data on the absorption distribution. We provide the following new Figure in Supplementary Note 8 on the responsivity:   Fig. S16: Absorption per material of the architecture Simulated absorption per material (rows) of the metamaterial absorber structure for different graphene thicknesses (columns). The first row shows the total absorption of the layer stack. The second row shows the absorption in the resonators consisting of gold and titanium. The third row shows the absorption in graphene which is strongly increasing with additional layers. Lastly, the fourth row shows the losses in the adhesion layer between the gold backplane and the alumina spacer layer.
2.	“Although the authors report the ultrafast photoresponse speed and discussed the possible limiting factors, but the cut-off frequency of 420 GHz still not be well explained. Could the author further study

	this point, as reveal the limitation of cut-off frequency is meaningful for the design of other high-speed PDs.”
	We thank the reviewer for this interesting point. Indeed, the 3 dB-frequency cut-offs at 420 GHz is a critical part that is worth discussing further – especially on the design aspect. As discussed within the paper, the cut-off can either stem from a (1) carrier dynamic related time constant or a (2) resistance-capacitance (RC) limit. We believe that the 420 GHz cut-off stems from the carrier dynamic aspect. (1) Regarding the carrier dynamics: We discuss the carrier dynamic relaxation by citing references 50, 55 and 56 that are able to measure high speed carrier dynamics within graphene with pump-probe measurements. Regrettably, we do not have the capabilities to perform these types of measurements for our specific samples that would offer additional insights. However, we see from reference 56 that the graphene quality and properties are crucial to reach high speeds. Comparing the frequency response of graphene PDs with the same design and built with specifically chosen graphene could in the future gain some insights into the material requirements for high-speed graphene PDs. However, this is clearly beyond the scope of this paper. (2) Regarding the RC limit: This is a crucial part to design high-speed PDs in general. Classical pin photodiodes suffered a clear trade-off between responsivity and frequency response, e.g., see 10.1017/CBO9780511635595. This classical view is here however not applicable. Therefore, we simulated the RC cut-off of the device by full 3D simulation of the layer stack and optimized the overall gate and contact pad geometry. We added the following section in the SI that discusses this in more detail: Main text, p9, l3: “Here, we trim a device for high-speed. We have used RF simulations to find an architecture that does not limit the bandwidth of the device but ideally reveals potential speed-limitations of the effect (see also Supplementary Note 9).”

Supplementary Note 9: Discussion on Frequency Response and RC cut-off

Fig. S17: Radio frequency pad design.

a Top view of the full device geometry implemented in the simulation environment (CST Studio). The contact pads (gray squares) are connected to an optimized co-planar waveguide section. The active area of the device is represented in red, which is implemented as a lumped element. **b** Corresponding optical microscope image of a fabricated device. **c** Side view of the layer stack within the simulation environment. The zoom-in shows the active region layer stack. The substrate is silicon (intrinsic) with a $1\mu\text{m}$ thick thermally grown silicon oxide layer. Within the silicon oxide the gold gate is buried. A layer of aluminum oxide is on top followed by the active area of the device, and the ground and signal lines. A 50 nm aluminum oxide passivation layer is added on top. The structure is surrounded by air. **d** Simulated RF response from 0 to 500 GHz for the above depicted geometry. No roll-off behaviour is found.

To ensure ideal operation at high frequencies we optimized the contact pad design. Fig. S17 illustrates the design. We implement the structure in CST Studio and perform time domain simulations. The RF pad design is illustrated in Fig. S17a. Three contact pads (gray squares) modelled as perfect electric conductors are connected to the co-planar waveguides (CPW) forming the Ground-Signal-Ground (GSG) structure. The CPW was optimized in a first step to match the wave impedance of the contact structure to the $50\ \Omega$ of the RF circuit. For this, only the CPW structure connected terminated with two ports have been simulated separately. The active area of the device was fixed at $10\ \mu\text{m}$ as well as the pitch of the pads which was set to $50\ \mu\text{m}$. Therefore, the parameters to optimize were the widths of the metal lines, the gaps between the lines and the length of the CPW. These parameters were optimized until the line impedance was close to $50\ \Omega$. We note that the material layer stack illustrated in Fig. S17c is determined by the metamaterial design as illustrated in Fig. S8.

Using the optimized CPW geometry we simulated the full device as illustrated in Fig. S17a. The active area of the device is here simply approximated as a lumped element with a resistance of $6000\ \Omega$. At the same spot a port for the excitation signal is implemented mimicking the photoresponse of the PD. The transmitted and received signal is then detected at the gray GSG pads which are terminated with $50\ \Omega$. The resulting simulated RF response is shown in Fig. S17d. No roll-off behaviour is found in the range from 0 to 500 GHz. Oscillation in the spectrum stem from non-well matched impedances between the active area and the probing. Strong

		drops in the frequency range have been identified as resonant coupling to the gate pad. In the measurements we did not experience these sharp drops which we assume stems from damping in the gate due to resistive losses.
3.	“Besides highest response speed, the responsivity and detectivity are also important. How the thickness of graphene influences the responsivity and detectivity?”	
		We thank the reviewer for this question on responsivity and detectivity. In addition to the answer to question 1, we added the simulated absorption in graphene as a function of different graphene thicknesses. As one can see, the absorption in graphene is increasing with numbers of layers without destroying the resonant behaviour of the structure. This should directly result in an increase in responsivity. Regarding the detectivity, one has to consider in addition to the responsivity the noise. As the device is operated without external bias, the main noise source in low light conditions is thermal noise. The thermal noise scales proportionally to the resistance.  - Higher number of graphene layers will reduce the resistance almost by a factor of number of layers:  - leading to an increase in thermal noise - but simultaneously to an increase in current responsivity. - The higher number of graphene layers will lead to an absorption enhancement which increases the responsivity. - The carrier mobility in multilayer graphene drops compared to single layer graphene, leading to a slight reduction in the Seebeck coefficient and therefore lower responsivity. Therefore, we expect overall a slight gain in detectivity by using multilayer graphene over single layer graphene. However, there will be an optimum layer thickness that balances the responsivity gain and the noise increase to achieve the best detectivity. We added the following sentences at the end of Supplementary Note 8 discussing these points:
		[...] Furthermore, the responsivity can be increased by etching the graphene channel. This would also increase the resistance and as a result could further reduce thermal noise and improve the NEP. However, for high-speed operation, it would cause a larger mismatch between the typically 50 Ω terminated RF-circuits and would lead to worse power transmission. Employing multilayer graphene would improve the current responsivity due to higher absorption and lower resulting resistance. Furthermore, saturation could potentially be increased. However, multilayer graphene will also have a lower mobility which in turn leads to a reduced Seebeck coefficient and therefore lower responsivity. The responsivity and detectivity will therefore improve up to a certain number of layers.
4.	“For the application demonstration, the authors show the wavelength and polarization division multiplexing. However, the time scale is in second level as shown in Figure 6, which is incommensurate to its ultrafast response speed.”	
		We thank the reviewer for pointing out this seeming discrepancy. The explanation here is due to setup limitations. We use source-measure units with an integration time set to 100 ms to measure the two perpendicular channels. While high-speed time traces would be interesting, we lack the necessary equipment to generate and measure such signals. This would require at least a real-time scope, two high-speed modulators operating in the O- and C-band, corresponding lasers and two high-speed waveform generators. Further, the fastest available scope offers an analog 3 dB bandwidth of 113 GHz and the fastest waveform generators have 3 dB analog bandwidths of 80 GHz. So even with

		the best available equipment there is no possibility to measure time traces close to the speed limit of the photodetectors. With the current responsivity the signal quality measured with such a high-speed scope will also be worse than with the precision source-measure units.
5.		"The English of this manuscript should be further polished for easy reading."
		We thank the reviewer for pointing out this flaw. We regret that the English did not meet the expectation. We re-checked the manuscript for typos and grammatical errors and fixed them throughout the manuscript.

REVIEWER COMMENTS

Reviewer #1 (Remarks to the Author):

see attached document

[Editorial Note: this document is displayed over the next ten pages]

Reviewer #2 (Remarks to the Author):

The authors have addressed all my concerns, and now I recommend its publication on Nature Communications. Congratulations.

Reviewer #3 (Remarks to the Author):

The authors have addressed all my concerns in the last round review.

My opinion on the paper is essentially unchanged. I truly appreciate the work that has been done, nevertheless I still believe that this work deserves a publication to a different journal and I cannot suggest the publication of the paper in Nature Communications. Moreover, given the new information provided by the authors, I sincerely suggest to revise data and methods.

Below I point out my concerns on almost the totality of authors points:

We would like to point out the many messages and novelties of this manuscript based on the figures: - Figure 1 & 2: Introduction of the device architecture. Novel here is that we incorporate the resonators in a metamaterial perfect absorber layer stack. This boosts the absorption

This boosts absorption compared to what? Integrated photonics detectors can easily achieve almost complete absorption.

and additionally makes gating possible with low CMOS compatible voltages – we demonstrate operation voltages ~ 1 V as opposed to ~ 100 V from the previous demonstrations.

I don't see any correspondence of these claims with literature. Here is a list of graphene gated devices with ~ 1 V gate operation exploiting PTE.

10.1021/acs.nanolett.9b02238

10.1021/acs.nanolett.6b03374

10.1038/s41467-021-21137-z

- Figure 4: Here we describe again the physics of the device and test the interaction of the PTE-DC with other effects; something that has not yet been shown either in simulation or experimentally.

These new types of measurements show that it is possible to clearly distinguish between different detection mechanisms in graphene which is rather difficult.

Well, a good experimental paper on how to distinguish the various effects is 10.1038/nphoton.2012.314. Concerning theory, the works of Prof. Song give the theoretical instruments to distinguish from PTE to PV. For PB, this can be calculated from the Drude conductivity model (or Kubo formula) coupled to the heat equation. These calculations are present in many papers dealing with graphene detectors.

By having – in addition to bias and gate voltage – the possibility to control the polarization in a manner that does not change the overall absorption in the graphene but only where the absorption is happening there is the possibility to turn on and off mechanisms making them distinguishable.

As already discussed in my previous review, this has already been shown

- Figure 5: The high-speed characterization of the device where we show a 420 GHz 3 dB frequency cut-off. We note that we do not show a setup limited performance but find a frequency cut-off of the detection mechanism.

There is no experimental evidence that this cut-off isn't due to parasitic elements. On the contrary this can be very probable, as discussed in the next comments.

- Furthermore, we find that different detection mechanisms within the same device show different frequency cut-offs. All these findings are not only remarkable (as of the high cut-off) but have never been reported before.

The frequency response of a photoconductor is textbook. The dynamic response of graphene photobolometers and photoconductors can be therefore calculated using the formulas present, e.g., in “Fundamentals of Photonics” by Saleh. Long photoconductors (in the charge carriers’ transport direction) lead to low BW, this is what authors observe.

Photovoltaic is textbook, same book for instance.

Concerning PTE, it is known that the relevant dynamics is carriers cooling, that is dependent on many factors. The papers list is long here, a good recent review is present in doi.org/10.1039/D0NR09166A.

Importantly, following simple rules that can be extracted from the mentioned references, one cannot a-priori exclude that the contrary can be obtained: slow PTE effect and a fast PB effect. The fact that authors have made a device that is fast using PTE and slower using PB is of course interesting, but follows simple geometrical rules and shows expected results. That cannot be ascribed as novelty, but rather as a good experimental information confirming what is already known.

Figure 6: We introduce the demultiplexing scheme showing the flexibility of the metamaterial design, i.e., that directional currents can be induced by different metamaterial sub cells allowing to combine different resonances (wavelength and/or polarisation) within the same architecture.

As I have previously remarked, the authors do not show any potential application. See comments below

While intuitively and from previous reports these interactions of the effects are clear, there is no experimental demonstration of the interaction of the PTE induced directional currents with the PB and PC effects. Furthermore, while one would be able to estimate regions of detection mechanisms, the magnitude of each effect is very difficult to estimate. We are not aware of a complete framework that is able to model PV, PC, PB and PTE accurately.

In my opinion there is a framework. It a set of the three coupled equations:

1 Kubo formula for conductivity with properly modeled scattering mechanisms, which ultimately depend on the dielectric environment and graphene mobility as a function of the chemical potential. Alternatively, a simple Drude model can be used.

2 Heat equation which includes Peltier, Seebeck effect and other cooling terms which depend on the graphene quality. In the sample presented in this, manuscript the graphene quality is low ($900 \text{ cm}^2/\text{Vs}$), therefore super-collision cooling terms must be considered, while in high quality samples encapsulated with hBN the cooling term due to coupling with hyperbolic phonons must be included.

3 charge conservation formula relating the charge density to chemical potential

For the complete theoretical framework, authors can consult Prof. Song PhD thesis “Hot Carriers in Graphene”. This only excludes hyperbolic phonons in hBN, that was still not known, but this is just a supplementary cooling term that can be easily found in literature.

The reviewer comments that mainly all the papers dealing with PTE detectors show PB operations under source-drain bias. As one can see from Fig. 4 (e-g) there is a strong gate dependence and the PTE induced directional currents can co-exist both with the BOL and PC effect. Therefore, our measurements reveal that the biased operation of the PTE device does not automatically mean a BOL operation, but that also a PC operation comes into play.

Yes, I agree. this depends on the chemical potential value, see 10.1038/nphoton.2012.314. Nevertheless, PC effect is weaker than PB

This insight can become important when additional factors modifying the PC and BOL response come into play, e.g., temperature. As the interaction regions of PC, BOL and PTE are depending on the contact doping given by the resonator material this interaction can also be set.

Yes, I agree. The photocurrent depends on doping, regardless of what is on top: resonator material or metal contact or electrostatic gating or whatever. Check 10.1038/nphoton.2012.314.

Furthermore, the reviewer comments that the PB effect is stronger than the PTE effect and is therefore used to measure eye diagrams. We find in our devices that the PTE induced directional currents reach $\sim 3/4$ of the PB response. While the response is indeed weaker, the PB effect comes inherently with large dark currents due to the conductivity of graphene. We found in our previous work (10.1126/science.adg8017) on the PV effect that the zero-bias operation outperforms the biased operation in data measurements due to the much lower noise.

Since the authors cite 10.1126/science.adg8017, I checked it. In the cited paper, even the 32 Gbit/s eye diagram gives a BER of 5.5×10^{-4} after “offline *digital signal processing (DSP)*” [...] “used with *Volterra nonlinear equalization, data aided decision feedback equalization for PAM-2*”. I guess this result is due to very poor SNR rising from the reported low responsivity. This BER is far from outperforming biased operation (see, as an example, a paper from the same group 10.1021/acsp Photonics.8b01234). Why authors say that they found that PV outperforms the biased operation in data measurements? Anyway, the current manuscript lacks data measurement, I guess because of the same ultra-low SNR issue.

Furthermore, zero-bias operation clearly grants advantages in circuit complexity, no required voltage source and furthermore also does not have an electrical power consumption. As we show for the first time in the frequency response measurements the BOL effect is also slower than the PTE-DC. Therefore, in our experience there is no advantage in operating such a device as a bolometer as often done in literature.

I agree, but decent responsivity should be achieved. This is the problem of graphene detectors: achieving high responsivity without the need of bias. This has been shown by very few groups in the world.

We measure the bandwidth of the detection mechanism and not an RC limited cut-off.

As an example of broadband detector, see 10.1049/el.2018.0932. In this paper the bandwidth is > 300 GHz for the photodiode module (so already assembled) and the device has been assembled to operate in the range 400 – 900 GHz with flat response over 340 GHz, and delivering a power in the range 400 – 900 GHz that is more than one hundred thousand times higher than the power declared by the authors in this manuscript during characterization (~ -74 dBm as declared in this rebuttal), and obtained without the need of a big amount of optical input power, simply because it has a decent responsivity. High speed devices can exhibit lower BW when assembled. In the paper that I take as an example, the module response is almost flat in the range 400-900 for more than 340-GHz BW. Below 400 GHz there is no reason for which the detector itself should not work. Simply, the capacitive coupling required in the assembly is such that this range is outside the operating region.

For many devices, it is just difficult to perform such large bandwidth measurement. This because large BW measurement is something that should be considered with extreme care. Authors claim a measurement revealing dynamics not due to parasitics. I respectfully provide my point of view about the definition of bandwidth measurement, considering that authors want to exclude parasitic effects. This must involve:

- Using a VNA
- Calibrate the measurement, including cables, RF probes etc... with calibration standards
- Fabricate and measure ad-hoc structures for de-embedding the device under test, therefore evaluating the parasitics due to pads, substrate etc...

After this, one can conclude that the measured bandwidth is due to the intrinsic limitation of the detector or to an RC limit.

Next, why there is a hole between 170 GHz and 220 GHz in the “bandwidth” measurement? Why the measurement starts at 3GHz? What’s going on before 3 GHz? The authors can surely access this frequency in some ways.

I asked to the authors information about the setup. The answer, below in this rebuttal, I must say, is very poor. Authors should:

- Share information about the noise of the used mixers, in terms of measured and declared NEP. I already asked for it, I just got the noise level of the used ESA.
- Share more details. For example, what is the exact model of the 110-170 mixer? And the model of the 220-330 GHz mixer? And for 330-500 GHz? What is the NEP of these devices? A complete set of information would comprise measured and declared performances by the vendor. What is the downconverted frequency range for each subharmonic mixer? Is it always the same? And the sensitivity? Is the LO frequency always the same?

Authors say that this is the second measurement performed revealing such huge BW, and the first was done by them as well. Well, since there are no other examples by other groups on this measurement, authors should:

- Share RAW original data of each RF probe response that has been used
- Share RAW original data of each RF cable response that has been used
- Share RAW original data of each subharmonic mixer response provided by the vendor.
- Share the measured absolute power response at the ESA
- Give a convincing explanation on why a hole of 50 GHz around 200 GHz should be acceptable, and also why a device bandwidth up to 430 GHz should be accepted as a record, provided that a flat response between 400 – 900 GHz has been already observed on a packaged device which of course works even at lower frequencies, and which exhibits at least one hundred thousand times more output electrical power using ~ mW optical input power.

Even providing these information, which would of course support this complicated measurement, this would just confirm that authors are measuring a device with 430 GHz bandwidth. There is no way that an ESA measurement without VNA and de-embedding devices can exclude parasitic elements limiting the bandwidth.

- We therefore cannot fully agree with the last part of the reviewer’s statement. The high speed of graphene PDs is not a fact and requires considerable engineering effort to achieve. We agree that the fast dynamics of hot electrons in graphene are a fact – but even here it is not well understood what the influence

of substrate, passivation and material quality is. For example, one can look at a more recent study on graphene carrier dynamics (10.1038/s41566-022-01058-z). Here not only the gate dependence on the relaxation dynamics is tested but also the decay rates across four different samples with strongly varying carrier mobilities. The different graphene qualities lead to strongly varying carrier dynamics. Depending on the graphene quality, the carrier dynamics seem further influenced. The topic is therefore far from being entirely understood and additional measurements are required and should be reported.

I agree, this involves the right experimental procedures. There is no way, in my opinion, to probe such fast carriers dynamics using an ESA and subharmonic mixers contacting a device. These measurements are instead performed using other methods involving, e.g. pump-probe measurements. Using a VNA and proper procedures would be interesting, but in my opinion rather complicated.

We thank the reviewer for his comment on the responsivity. We agree that increasing the responsivity is one of the major challenges for graphene photodetectors. Here, we would like to stress that our paper offers a solution to the very challenge.

I do not agree, responsivity is very low.

If one takes a commercial fast detector, and adds 30 dB attenuation, one gets the same linearity and responsivity as the one reported here.

By just adding a low pass filter at the output of a commercial 100 GHz detector, one gets a 500 GHz detector with highest responsivity compared to the reported one. Also linearity would be comparable, maybe better.

So, what is the “offered solution” as authors claim?

What we show is that for high-speed applications what matter is the maximum achievable output signal which is the product of responsivity and saturation power or its linear range.

I agree. The reported product of responsivity and saturation power is very low due to ultra-low responsivity. Moreover the onset of saturation at 10 mW (observed in this detector) is completely normal in detectors with $\times 1000$ responsivity.

Here we are able to compensate in part for the relatively low responsivity as we can operate the device at higher optical input powers. We can use higher optical input powers as we are spreading the power across a larger area in comparison to e.g., waveguide based devices. While the atomical thickness remains the same, the power per area on the photodetector is much lower in such a top illuminated device. So, we believe that even when we increase the responsivity of the device, we will not run into the same saturation issues as many other devices experience. We conducted an additional measurement on the linearity of the device which we added to the supplementary information. We swept the optical input power over more than 3 orders of magnitude. Here we now used a lensed fiber that we procured in the meantime. The smaller illumination spot ($\sim 5 \mu\text{m}$) ensures that the optical power is coupled to the device's active area. The lensed fiber creates a better overlap compared to a standard single mode fiber, but also creates under-illuminated areas within the device. A slow loss of linearity in the 5-10 mW regime is observed. We comment on this further below on the reviewer's question on saturation.

How exactly authors quantify this “low loss” of linearity? Usually, the 1 dB compression point is taken.

We believe that the photodetectors have the potential to reach the 0.1 A/W mark (as outlined in the supplementary information).

I don't see any rigorous argument supporting this belief

With these values, they would match or outperform other graphene devices and also other technologies such as UTC-PDs. We also highlight that we provide an external responsivity unlike most WG based devices that deduct coupling losses and only provide an internal responsivity. We further want to note that, to the best of our knowledge, there is no UTC-PD with a bandwidth covering the low GHz up to 420 GHz, especially not without any external bias (SOTA zerobias: 135 GHz 10.1063/5.0119244). Furthermore, the comparison to other technologies should also take into account other factors than responsivity and bandwidth. A graphene based device has clear advantages in manufacturability, integration capabilities and cost. Also, the first UTC-PD demonstration happened ~15 years before the first high-speed graphene PD demonstration and benefitted from decades of PIN-PD developments. We believe, if given time, graphene will be able to compete with the old and established technology which took many years to be developed.

I agree on this, but it is not reported here

Regarding the multiplication of the improvement factors. We agree that mobility, number of layers and resistivity are interconnected. We state that mobility influences the Seebeck coefficient and the resistance. We further say that the multilayer graphene influences absorption and resistance. In the end, we multiply:

(Seebeck coeff.) × (Absorption enhancement) × (wavelength unit cell change) × (resistance)

These coefficients are independent of each other, but as the reviewer pointed out correctly, linked to the graphene properties such as mobility and thickness. We stress the meaning of the multiplication by adding the meaning of each factor to the text:

SI p13, l24: "Multiplying all estimated contributions leads to a total of

(Seebeck coeff.) × (Absorption enhancement) × (wavelength unit cell change) × (resistance factor)

$$3 \times 4 \times 4 \times 5 \sim 10^2$$

explaining the two order of magnitude lower current responsivity in our devices."

I still say that this multiplication is not realistic at all, for the reasons that I have already given. My statement has not been considered by the authors. The resistance depends on the mobility, the Seebeck coefficient as well. scaling the resistance by a factor of 2 does not imply multiplying the responsivity by a factor of two. Also, the seebeck coefficient depends on the resistance (see its formula). Concerning the absorption term, authors are multiplying it by 4. The device as it is now absorbs the 60 % of the incoming radiation. How is it possible to absorb more than 100%?

As the PD can be seen as a distribution of smaller PDs over an area, there is also the possibility to increase the active area of the device by adding more of these elements. This in turn would again allow higher

optical input powers as the power is spread across a larger area. However, how the responsivity and RF response is influenced by the scaling is not yet clear.

Yes it is, it can be checked in the supplementary information of 10.1038/s41467-021-21137-z, and it is a textbook exercise.

Depending on the application there might ultimately be trade-offs between RF bandwidth, responsivity, sensitivity and saturation. Further research on the topic is required.

Completely agree on this

We thank the reviewer on his comment of the demultiplexing application.

We first want to point out that the demonstrated device combines the two functionalities (wavelength division demultiplexing and polarization demultiplexing) simply as a proof-of-concept device. The goal of the device is foremost to show that more complex metamaterial unit cells can be employed and do not perturb the overall effect, and in addition provide access to advanced functionalities.

Authors do not discuss what are these advanced functionalities

We do not claim – or even target – that we would outperform any other wavelength or polarization demultiplexing schemes at this time.

Therefore, I respectfully ask to the authors: what is the point?

We thank the reviewer for his excited comment about the RF pad design and RC cut-off. First, we comment on the reviewer's calculated capacitance. It is important to consider the full electrical circuit when operating the device in a high-speed application.

The photodetector resistance is essentially in parallel to the load resistance. In a high frequency circuit this corresponds to 50Ω . The relevant resistance for an RC cut-off of the system is thereby

$$\left(\frac{1}{6000 \Omega} + \frac{1}{50 \Omega} \right)^{-1} \approx 50 \Omega.$$

I assume that C_{PD} is the pad capacitance. This circuit has ~ 450 GHz bandwidth if C_{PD} is 7fF (I wrote ~ 5 fF in my previous comment, but the correct number is 7fF).

By looking at the image of the device that the authors put in this new version, I get even 9fF pad capacitance (<https://www.emissoftware.com/calculator/coplanar-capacitance/>) considering just silicon oxide as the substrate, $\epsilon_{ps}=3.9$, therefore effective epsilon around 2.45 (calculated from a modal simulation).

Considering the silicon substrate, the effective epsilon is higher, thus things get even worse. I consider 2 parallel pads with dimension of 72x72 with 14 um gap. I took these values from the scale on the image that the authors have attached. Capacitance could be even higher in reality which means, lower frequency cutoff.

From this considerations, my point is: there is no experimental evidence that the cutoff that authors show is intrinsic to the device, it can be due to the parasitics (and anyway is outstanding), therefore no conclusions can be done on it. This is the reason why carrier dynamics are measured with more reliable methods.

Further, the RF pads are a travelling-wave design which makes the lumped capacitance of the pads not directly relevant. To discuss the RF pad design, we added a new section in the supplementary information that discusses this in more detail. It reads as follows:

How can an RF pad be “travelling wave”? They are lumped by definition. Also the electrodes that connect the pads to the graphene active area do not represent a transmission line (or a “travelling wave” electrode, adopting the authors vocabulary). Considering around 2.5 effective epsilon, one wavelength at 500 GHz is around 400 um, which is longer than the entire electrode.

To ensure ideal operation at high frequencies we optimized the contact pad design. Fig. S17 illustrates the design. We implement the structure in CST Studio and perform time domain simulations. The RF pad design is illustrated in Fig. S17a. Three contact pads (gray squares) modelled as perfect electric conductors are connected to the co-planar waveguides (CPW) forming the Ground-Signal-Ground (GSG) structure. The CPW was optimized in a first step to match the wave impedance of the contact structure to the 50 Ω of the RF circuit. For this, only the CPW structure connected terminated with two ports have been simulated separately. The active area of the device was fixed at 10 μm as well as the pitch of the pads which was set to 50 μm . Therefore, the parameters to optimize were the widths of the metal lines, the gaps between the lines and the length of the CPW. These parameters were optimized until the line impedance was close to 50 Ω . We note that the material layer stack illustrated in Fig. S17c is determined by the metamaterial design as illustrated in Fig. S8

Using the optimized CPW geometry we simulated the full device as illustrated in Fig. S17a. The active area of the device is here simply approximated as a lumped element with a resistance of 6000 Ω . At the same spot a port for the excitation signal is implemented mimicking the photoresponse of the PD. The transmitted and received signal is then detected at the gray GSG pads which are terminated with 50 Ω . The resulting simulated RF response is shown in Fig. S17d. No roll-off behaviour is found in the range from 0 to 500 GHz. Oscillation in the spectrum stem from small impedance mismatches between the active area and the probing. Strong drops in the frequency range have been identified

Some remarks:

- according to the figures and to the device picture, the pitch is not at all 50 um.
- What is the meaning of an RF simulation in a simulation area that is not even one wavelength long?
- How can the RF response of a 6000 Ohm resistor connected to a 50 Ohm load be 0dB?

as resonant coupling to the gate pad. In the measurements we did not experience these sharp drops which we assume stems from damping in the gate due to resistive losses.

What are the resistive losses the authors are referring on? Thin metal? This would be present also at the device level and would contribute to frequency cutoff, since losses contribute to a cutoff. These resistive losses are not even modeled, for example the substrate is considered as intrinsic. I asked for the resistivity of the substrate but I received no answers on this.

The noise floor strongly depends on the frequency range of interest but is with our equipment and settings typically around or below -90 dBm. As an example, the noise floor at 400 GHz is -94 dBm and the measured RF power of the device is -74 dBm, leaving us a 20 dB span above the noise floor.

There is something that I really don't understand. Let's consider $500\mu\text{A/W}$ responsivity, as declared. Let's consider 1 mW optical input. The generated current is 500 nA. Let's consider the device model, without parasitics:

Authors can easily check that this corresponds to a power of ~ -90 dBm at the subharmonic mixer. Let's consider the best Virginia diode SAX model in terms of SSB conversion efficiency. This has 10 dB loss per sideband, this means 13 dB loss. Let's consider 10, for simplicity. This means that the signal, without any kind of loss (which is physically impossible) should be ~ -100 dBm at the ESA. Well, authors can state that they used 10 mW optical input, which means ~ -80 dBm at the ESA with no loss.

This means that, to get -74 dBm with such extremely low responsivity, even without any kind of loss (and negligible loss are physically impossible at such frequencies), authors used > 10 mW optical input. I would rather say 30-50 mW, even more, realistically speaking. If we then consider SSB losses of the Virginia diodes SAX modules at higher frequencies, well, things become even more complicated.

I think that authors must clarify this, in whatever journal they publish this work. The exact optical power coupled to the device has to be declared, and this must correspond to an electrical signal level extracted from the already declared DC responsivity. This signal level must be compared to the noise level of the used equipment chain, also considering losses.

Moreover if huge optical power has been used to extract this ultra-low electrical signal, authors must revise the RF measurement which does not correspond anymore to a small-signal regime. At this regard, authors should also revise their experimental conclusions since they are working in a saturation regime according to their linearity measurement

We thank the reviewer for his comment on the metal pinning. This is indeed a very interesting point.

We first want to note that we did not make any comments on what mobility our device could reach.

Well, that should be done in my opinion

We merely argued that with our relative low mobility of $< 1000 \text{ cm}^2/\text{Vs}$ a moderate improvement in the mobility of factor 5x would lead to an increase in the Seebeck coefficient of a factor 3x. The work we compare ourselves to in this estimation does not report any mobility values and therefore we chose a relatively low mobility value for high quality graphene. We are aware of the negative effect of metal contacts on the carrier mobility in graphene. However, there are studies that demonstrate that even with high impurity concentrations it is possible to achieve several $1000 \text{ cm}^2/\text{Vs}$ in mobility, e.g., (10.1038/s42005-021-00518-2). Therefore, we strongly believe that this improvement in Seebeck coefficient is achievable.

I did not speak about impurities concerning metal. I

'm not aware on papers showing $> 5000 \text{ cm}^2/\text{Vs}$ mobility in Coulomb-dominated scattering regime on SiO₂ substrates.

We thank the reviewer for his question on the link between resistance and responsivity. As PTE based devices are typically seen as generating a photovoltage and not a photocurrent the device resistance directly also influences the responsivity in terms of photocurrent. Photovoltages generated by the Seebeck effect are linked to thermoelectric currents across a PD by Ohms law: $I_{ph} = V_{ph}/R$ or $\mathcal{R}_A = \mathcal{R}_V/R$.

This can be seen in various examples in literature:

- $\mathcal{R}_V = 27 \text{ V/W}$ or $\mathcal{R}_A = 36.3 \text{ mA/W}$ with a device resistance of 500 – 1000 Ω (10.1038/s41467-020-20115-1)
- $\mathcal{R}_V = 3.5 \text{ V/W}$ or $\mathcal{R}_A = 35 \text{ mA/W}$ with a device resistance of $\sim 110\Omega$ (10.1021/acs.nanolett.6b03374)
- Split-gate PTE configuration where ideal voltage responsivity (high resistance) and ideal current responsivity (low resistance) is identified as a function of two gate voltages. (10.1038/s41467-022-31607-7)

To clarify this point, we expanded Supplementary Note 8 and added the new Figure S15 showing the above mentioned new recorded power sweep where we measure once the photovoltage and once the photocurrent:

There is a clear theory beyond this, relating mobility, resistance, seebeck coefficient and cooling time, ultimately predicting responsivity. This theory can be of course trimmed, but is the only path that triggers designs. I honestly do not approve this method of multiplying things based on a couple of papers.

Reviewer 1

Summarized below are the reviewer's comments and replies for both review rounds. The formatting is as follows:

- First Round (gray shaded background):
 - o Reviewer's comments (*italic light gray*),
 - o Author's answers (**blue**),
 - o Alterations to the initial manuscript (**red**)

- Second round:
 - o Reviewer's comments (*italic gray*),
 - o Author's answers (**light blue**)
 - o Alteration to the initial manuscript (**light red**)

"The article "Highest-Speed Photothermoelectric Directional Currents in Graphene" by Koepfli et al. shows a high-speed graphene detector based on a metamaterial perfect absorber suitable for NIR and MIR detection regime. The manuscript is very clear, instructive, and well written. Having said that, I do not see any relevant novelty (neither fundamental nor applications-related) justifying a publication in Nature Communications. In the following I'll explain my point of view and my major concerns."

1.	Concerning the fundamental aspects covered in the paper, the authors discuss known concepts and designs, as PTE DCs have already been analyzed in some papers also cited by the authors. Indeed 10.1038/s41467-020-20115-1 proposes the same geometry and shows measurements very similar to the one shown in this manuscript. It is already known that PTE is responsible for DC (10.1038/s41566-021-00819-6, 10.1038/s41566-022-01115-7, 10.1038/s41467-023-39071-7). The dynamic flow model has been already applied to study directional currents as a function of light polarization, and as a function of the metal contact disposition with respect to the electric field resonance produced by the nanostructures composing the metamaterial (10.1038/s41566-021-00819-6).
	We are sorry to hear that the many new aspects introduced by this paper were not clearly communicated. Also, as pointed out by the reviewer, we cite all the works correctly and we do not claim novelty on any of the above mentioned aspects. Of course, we had to introduce the interesting geometry that was introduced by others as we performed research on this geometry. It is not possible to discuss the device without introducing it first. This also includes making use of the already developed modelling framework. We would like to point out the many messages and novelties of this manuscript based on the figures: - Figure 1 & 2: Introduction of the device architecture. Novel here is that we incorporate the resonators in a metamaterial perfect absorber layer stack. This boosts the absorption and additionally makes gating possible with low CMOS compatible voltages – we demonstrate operation voltages ~1 V as opposed to ~100 V from the previous demonstrations.

		- This boosts absorption compared to what? Integrated photonics detectors can easily achieve almost complete absorption. - I don't see any correspondence of these claims with literature. Here is a list of graphene gated devices with ~ 1V gate operation exploiting PTE. 10.1021/acs.nanolett.9b02238 10.1021/acs.nanolett.6b03374 10.1038/s41467-021-21137-z
		The paper the reviewer now cites are based on a completely different architecture and use a split gate configuration to introduce a PTE effect. Also, from the cited papers only one achieves sub-5V gate operation. Our comparative statements are with respect to the directional photocurrent papers that the reviewer cited in his initial comment to which this answer was directed. The same is true for the remark on the gate voltage. In general, our manuscript discusses the properties of the photothermoelectric induced directional photocurrents.
		- Figure 4: Here we describe again the physics of the device and test the interaction of the PTE-DC with other effects; something that has not yet been shown either in simulation or experimentally. These new types of measurements show that it is possible to clearly distinguish between different detection mechanisms in graphene which is rather difficult. ...
		- Well, a good experimental paper on how to distinguish the various effects is 10.1038/nphoton.2012.314. - Concerning theory, the works of Prof. Song give the theoretical instruments to distinguish from PTE to PV. For PB, this can be calculated from the Drude conductivity model (or Kubo formula) coupled to the heat equation. These calculations are present in many papers dealing with graphene detectors.
		- We fully agree with the comment on the 2012 Nature Photonics paper being an excellent reference that experimentally shows the occurrence of different detection mechanisms in graphene. Our measurements shown in Fig. 4f correspond to the measurements demonstrated in this paper. But again, we clearly expand this analysis by Fig. 4e and Fig. 4g where we add the directional photocurrents on top of the PC and PB effects. Furthermore, we also investigate the frequency dependence on these effects. - We thank the reviewer for this reference. We're aware of many of the papers that resulted from said thesis. We give a more detailed answer on the later comment on the modelling framework.
		... By having – in addition to bias and gate voltage – the possibility to control the polarization in a manner that does not change the overall absorption in the graphene but only where the absorption is happening there is the possibility to turn on and off mechanisms making them distinguishable.
		As already discussed in my previous review, this has already been shown.
		Only Fig. 4(f) is known. The interesting finding seen from Fig. 4(e) and (g) are the experimental demonstrations that PTE-DC effect is additive. There is no demonstration or experimental verification on this finding. In addition, there is the frequency response analysis which is also something that has not been done to the here reported findings.

	- Figure 5: The high-speed characterization of the device where we show a 420 GHz 3 dB frequency cut-off. We note that we do not show a setup limited performance but find a frequency cut-off of the detection mechanism. ...
	There is no experimental evidence that this cut-off isn't due to parasitic elements. On the contrary this can be very probable, as discussed in the next comments.
	So what? Even if it were parasitic limited, we can show that the experimentally achievable frequency response is at least 420 GHz – a finding that outperforms the current state-of-the art by 350 GHz! Anyhow, the experiments show that the PC, the BOL effects drop off below the PTE-DC effects. These are clear indications that at least the former are not parasitic limited. We give an in-depth answer on page 10.
	... Furthermore, we find that different detection mechanisms within the same device show different frequency cut-offs. All these findings are not only remarkable (as of the high cut-off) but have never been reported before.
	The frequency response of a photoconductor is textbook. The dynamic response of graphene photobolometers and photoconductors can be therefore calculated using the formulas present, e.g., in "Fundamentals of Photonics" by Saleh. Long photoconductors (in the charge carriers' transport direction) lead to low BW, this is what authors observe. Photovoltaic is textbook, same book for instance. Concerning PTE, it is known that the relevant dynamics is carriers cooling, that is dependent on many factors. The papers list is long here, a good recent review is present in doi.org/10.1039/D0NR09166A. Importantly, following simple rules that can be extracted from the mentioned references, one cannot a-priori exclude that the contrary can be obtained: slow PTE effect and a fast PB effect. The fact that authors have made a device that is fast using PTE and slower using PB is of course interesting, but follows simple geometrical rules and shows expected results. That cannot be ascribed as novelty, but rather as a good experimental information confirming what is already known.
	A fast frequency response is never textbook. There are simply too many effects that may prevent a real device to follow textbook. On the contrary, textbook statements have to be substantiated by experimental findings. We provide such experimental work and demonstrate the frequency response of an actual device and show that it is possible to switch between mechanisms. There is no prior art on this topic. We're aware of the Fundamentals by Saleh and Teich and the equations mentioned by the reviewer, but we don't simply report on theoretical calculations here. We're also aware that PTE based photodetection is based on carrier cooling and as the reviewer says is a complex topic. Therefore, it is difficult to make a priori statements which is why experimental work has to be done and should be reported. To quote the extensive review cited by the reviewer: "These devices typically display a bandwidth larger than 40 GHz (ref. 286, 289, 290 and 292) and the highest to date surpasses 67 GHz (setup-limited, see Fig. 20b). (ref. 287) [...] Ultimately, improving the large-scale fabrication (ref. 298) and design (ref. 13) of graphene-based

		photodetectors could open the door to faster and more power-efficient receivers in tele- and datacom modules." Again, highlighting that the experimental demonstration so far ends at 67 GHz. So, our 420 GHz seems to be quite a novelty.
		- Figure 6: We introduce the demultiplexing scheme showing the flexibility of the metamaterial design, i.e., that directional currents can be induced by different metamaterial sub cells allowing to combine different resonances (wavelength and/or polarisation) within the same architecture. Therefore, in our view, only Figure 3 is a re-iteration of already published work. However, in our view Figure 3 is helpful to understand the full discussion in the manuscript. We answer the more detailed questions below. Wherever we make changes to the manuscript we consider the comment on the novelty and make changes accordingly to communicate the novelty more clearly.
		As I have previously remarked, the authors do not show any potential application. See comments below
		We show a polarization multiplexing and for the first time a new wavelength multiplexing scheme. These are very relevant applications that can now be obtained within a single photodetector. Many new applications might be derived from this work. Given the length of the paper, we cannot further elaborate more applications. And beyond, this would be beyond the scope of this paper.
2.		Authors say that the "the interaction with other effects is not yet understood". The authors calculated the PTE DC using known formulas. Then, they experimentally observed that by biasing the graphene channel, PC and PB effects add up. This is known: at zero bias, the only mechanisms occurring are PV and PTE. Instead, photocurrent generation in biased graphene is at the charge of PC or PB effects, depending on the graphene chemical potential (see, as an example, 10.1038/nphoton.2012.314). Of course, these two effects add up constructively or destructively with the photocurrents produced by PV or PTE effects, which are mainly bias independent (or even degrade with bias). The competition between many photocurrent mechanisms has also been experimentally shown in many papers. Mainly all the papers (except very few) dealing with PTE detectors show PB operations under source-drain bias. This is done to measure, e.g., eye diagrams, since the PB effect is, as a matter of fact, stronger than PTE for samples with mobilities $< \sim 5000 \text{ cm}^2\text{V}^{-1}\text{s}^{-1}$.
		We thank the reviewer for his comment on the interaction of the graphene detection mechanisms. We agree with the above statements, but we want to highlight that our measurements go beyond the above stated. While intuitively and from previous reports these interactions of the effects are clear, there is no experimental demonstration of the interaction of the PTE induced directional currents with the PB and PC effects. Furthermore, while one would be able to estimate regions of detection mechanisms, the magnitude of each effect is very difficult to estimate. We are not aware of a complete framework that is able to model PV, PC, PB and PTE accurately. The polarization dependence of the PTE induced directional currents combined with the polarization independent absorption characteristics of the devices' metamaterial allow for an ideal method to also compare magnitude and interaction of the effects which is summarized in Fig. 4.
		In my opinion there is a framework. It a set of the three coupled equations: 1 Kubo formula for conductivity with properly modeled scattering mechanisms, which ultimately depend on the dielectric environment and graphene mobility as a function of the chemical potential. Alternatively, a simple Drude model can be used.

		2 Heat equation which includes Peltier, Seebeck effect and other cooling terms which depend on the graphene quality. In the sample presented in this, manuscript the graphene quality is low (900 cm²/Vs), therefore super-collision cooling terms must be considered, while in high quality samples encapsulated with hBN the cooling term due to coupling with hyperbolic phonons must be included. 3 charge conservation formula relating the charge density to chemical potential For the complete theoretical framework, authors can consult Prof. Song PhD thesis “Hot Carriers in Graphene”. This only excludes hyperbolic phonons in hBN, that was still not known, but this is just a supplementary cooling term that can be easily found in literature.
		Such a simulation is for sure something of high interest and worth pursuing, but clearly beyond the scope of this paper. Anyhow, we thank the reviewer for outlining framework by Prof. Song to model the graphene photodetection mechanisms.  - 1) We already use the Kubo formula to calculate the optical sheet conductivity of the graphene to simulate the optical absorption. - 2) We are aware of these equations including the supercollision cooling and hyperbolic phonon coupling – as we also cited in the manuscript. We also use several of the heat equations. However, we did not include all of these into our modelling framework yet. - 3) We use the charge conservation to model the phenomenon of the directional current flow. The optical absorption is used as an input for these simulations. We further looked through the PhD thesis the reviewer cited. While covering a lot of aspects, we still do not see a 1:1 framework with coupled equations that model optical, thermal, electrical and frequency dependent behaviour of a graphene based PD that has been tested on a corresponding device. Fully coupling all of these equations on a device architecture in a multiphysics simulation is a huge endeavour. Also, such simulations require extensive experiments to substantiate the unknown parameters. In this work we provide experimental data to support future simulation efforts.
		The reviewer comments that mainly all the papers dealing with PTE detectors show PB operations under source-drain bias. As one can see from Fig. 4 (e-g) there is a strong gate dependence and the PTE induced directional currents can co-exist both with the BOL and PC effect. Therefore, our measurements reveal that the biased operation of the PTE device does not automatically mean a BOL operation, but that also a PC operation comes into play. [...]
		Yes, I agree. this depends on the chemical potential value, see 10.1038/nphoton.2012.314. Nevertheless, PC effect is weaker than PB.
		We're aware of the dependence of the chemical potential values, that is why we answered accordingly and we have cited 10.1038/nphoton.2012.314 by Freitag, Low and Xia. Indeed, the PC effect is weaker than the PB and that is what we report. Yet, depending on operation regions the PC effect will occur. This contrary to the reviewers' statement. Obviously, this seems to be new.
		[...]This insight can become important when additional factors modifying the PC and BOL response come into play, e.g., temperature. As the interaction regions of PC, BOL and PTE are depending on the contact doping given by the resonator material this interaction can also be set.
		Yes, I agree. The photocurrent depends on doping, regardless of what is on top: resonator material or metal contact or electrostatic gating or whatever. Check 10.1038/nphoton.2012.314.

		We're not sure how to interpret this comment. The reviewer states that the photocurrent depends on doping – and this may be by a metal or gating. Fine, in our paper we even show how the two doping mechanism interplay. See e.g. Fig. 3(f).
		Furthermore, the reviewer comments that the PB effect is stronger than the PTE effect and is therefore used to measure eye diagrams. We find in our devices that the PTE induced directional currents reach ~3/4 of the PB response. While the response is indeed weaker, the PB effect comes inherently with large dark currents due to the conductivity of graphene. We found in our previous work (10.1126/science.adg8017) on the PV effect that the zero-bias operation outperforms the biased operation in data measurements due to the much lower noise.
		Since the authors cite 10.1126/science.adg8017, I checked it. In the cited paper, even the 32 Gbit/s eye diagram gives a BER of 5.5×10^{-4} after “offline digital signal processing (DSP)” [...] “used with Volterra nonlinear equalization, data aided decision feedback equalization for PAM-2”. I guess this result is due to very poor SNR rising from the reported low responsivity. This BER is far from outperforming biased operation (see, as an example, a paper from the same group 10.1021/acsp Photonics.8b01234). Why authors say that they found that PV outperforms the biased operation in data measurements? Anyway, the current manuscript lacks data measurement, I guess because of the same ultra-low SNR issue.
		The “ultra-low SNR issues” that the reviewer refers to are still the highest data rates (132 Gbit/s) yet achieved for a graphene-based photodetector. In said paper we did not report the biased operation because it was worse. So our statement that PV outperforms the biased operation is based on our experimental finding. In this work however, there was no point in doing data experiments as we report on a photodetector, its operation principles and mechanisms. This detector can be used for anything such as RF sensing, data detection,... . A data measurement will for sure be a worthwhile task in the future to demonstrate one specific use case out of many.
		Furthermore, zero-bias operation clearly grants advantages in circuit complexity, no required voltage source and furthermore also does not have an electrical power consumption. As we show for the first time in the frequency response measurements the BOL effect is also slower than the PTE-DC. Therefore, in our experience there is no advantage in operating such a device as a bolometer as often done in literature. We added the following sentences to the manuscript to highlight more the implications of the measurements presented in Fig. 4: p8, l43: “These results indicate that the different detection mechanisms in graphene are additive. The possibility to control the polarization in a manner that does not change the overall absorption in the graphene but only where the absorption spatially is happening allows to turn on and off mechanisms making them clearly distinguishable. By choosing the contact metal⁴⁹ of the resonators appropriately and control of the gate voltage, one could further optimize the combination of the detection mechanisms and the peak operation.
		I agree, but decent responsivity should be achieved. This is the problem of graphene detectors: achieving high responsivity without the need of bias. This has been shown by very few groups in the world.

		We highlight again that achievable maximum output signal is a key metric for high-speed photodetectors and that we report on the 2nd highest output photovoltage of a high-speed graphene PTE device while achieving an order of magnitude higher bandwidth (see Table S1). High responsivity has to be achieved in tandem with high optical saturation power and high bandwidth.
		As an example of broadband detector, see 10.1049/el.2018.0932. In this paper the bandwidth is > 300 GHz for the photodiode module (so already assembled) and the device has been assembled to operate in the range 400 – 900 GHz with flat response over 340 GHz, and delivering a power in the range 400 – 900 GHz that is more than one hundred thousand times higher than the power declared by the authors in this manuscript during characterization (~-74 dBm as declared in this rebuttal), and obtained without the need of a big amount of optical input power, simply because it has a decent responsivity . High speed devices can exhibit lower BW when assembled. In the paper that I take as an example, the module response is almost flat in the range 400-900 for more than 340-GHz BW. Below 400 GHz there is no reason for which the detector itself should not work. Simply, the capacitive coupling required in the assembly is such that this range is outside the operating region.
		We're well aware of the UTC literature and the paper the reviewer cites. However, we do not understand the point the reviewer is trying to make. The reviewer says very few groups in the world are able to achieve zero-bias graphene PDs with high responsivity and continues the argument by referring to a biased UTC PD paper. We never make any claims that we outperform a biased UTC PD in terms of RF output power. In this paper we explore the PTE-DC effect and communicate on the potentials with this technology.
		For many devices, it is just difficult to perform such large bandwidth measurement. This because large BW measurement is something that should be considered with extreme care. Authors claim a measurement revealing dynamics not due to parasitics. I respectfully provide my point of view about the definition of bandwidth measurement, considering that authors want to exclude parasitic effects. This must involve:  - Using a VNA - Calibrate the measurement, including cables, RF probes etc... with calibration standards - Fabricate and measure ad-hoc structures for de-embedding the device under test, therefore evaluating the parasitics due to pads, substrate etc... After this, one can conclude that the measured bandwidth is due to the intrinsic limitation of the detector or to an RC limit.
		We appreciate the reviewer's insights on how to perform high-speed measurements and perform calibrations. The head of the institute has been working on high-speed optoelectronics for the past 30 years and the group belongs to the leading institutes in the field. We are indeed aware of all the calibration issues... . The reviewer is correct, that limitations may be due to setup but also due to devices. The setup limitations are known to us. The device limitations are to be explored. And there are not just RC limitations.
		Next, why there is a hole between 170 GHz and 220 GHz in the "bandwidth" measurement? Why the measurement starts at 3GHz? What's going on before 3 GHz? The authors can surely access this frequency in some ways.
		This is a new question that is not based on the reviewer's question from round 1. The hole between 170 GHz and 220 GHz stems from missing measurement equipment. We choose to invest our finances into more interesting higher frequency bands. Thus, we do not have access to the electrical spectrum analyser extender module in said frequency range. The lower limit of 3 GHz originates from the measurement routine chosen. With the laser beating the lowest frequencies we can reliably measure are around 1 GHz. However, we chose to perform the whole sweep with the same ESA settings (span, resolution bandwidth, integration time, etc)

		for which the 3 GHz starting point is the lowest that makes sense to stay consistent within the bandwidth measurement. We do have access to the lower frequency regime by means of electro-optic modulators and RF sources. We have not seen any low frequency behaviour different from the GHz-regime and did therefore not perform the measurements for this specific device with a different setup.
		I asked to the authors information about the setup. The answer, below in this rebuttal, I must say, is very poor. Authors should:  - Share information about the noise of the used mixers, in terms of measured and declared NEP. I already asked for it, I just got the noise level of the used ESA. - Share more details. For example, what is the exact model of the 110-170 mixer? And the model of the 220-330 GHz mixer? And for 330-500 GHz? What is the NEP of these devices? A complete set of information would comprise measured and declared performances by the vendor. What is the down converted frequency range for each subharmonic mixer? Is it always the same? And the sensitivity? Is the LO frequency always the same?
		We regret that the reviewer perceives our answer as very poor. We do not see how the requested information to this extensive detail would improve upon the quality of the paper. We report our findings as it is typically done in the scientific literature.  - The noise level we shared was the combination of the mixer and the ESA. The extension modules work in unison with the ESA. Conversion losses are directly taken into account by the instrument itself. The calibration files are directly read-out by the ESA and incorporated. - We provided the down mixer information in terms of product type (SAX), vendor name (Virginia Diodes) and frequency bands (110-170, 220-330, 330-500). From these information it is clear which modules we're referring to, see VDI's webpage: https://vadiodes.com/en/products/spectrum-analyzer The specifications and operation principles of each mixer can be found in the data sheets and manuals. We do not plot vendor information and manuals in a scientific paper.
		Authors say that this is the second measurement performed revealing such huge BW, and the first was done by them as well. Well, since there are no other examples by other groups on this measurement, authors should:  -Share RAW original data of each RF probe response that has been used -Share RAW original data of each RF cable response that has been used -Share RAW original data of each subharmonic mixer response provided by the vendor. -Share the measured absolute power response at the ESA
		The reviewer now asks for excessive data. Data that are not relevant as calibration is done through calibrated reference measurements of the setup and the RF prober only. It is not necessary to measure every single element. Providing the reference measurements will not improve the quality of the paper and will not help anybody to repeat the experiment. We therefore append the measured RF power and the loss calibration only here to this response but will not extend the paper further.

Again, we state that the recorded RF power plotted above does already take into account any conversion losses by the mixers. Therefore, only the probe losses are deducted from the measurement region. Only for <110 GHz measurement there are additional cable, connector and bias tee losses which are calibrated by reference measurements and fitted accordingly.

However, the reviewers question rather reveals a lack of experience in the field. This is rather annoying as the only thing this question implies is that quote, “this complicated measurement” is not done correctly by us, essentially implying we are not competent enough to perform RF measurements, or even worse implying we fabricated our data. We do not understand how the data we provided leads the reviewer to such a conclusion. If the reviewer thinks that we have done erroneous measurements, then he should be clearer about what has been done wrong.

- Give a convincing explanation on why a hole of 50 GHz around 200 GHz should be acceptable, [...]

We lack the corresponding measurement equipment to perform the measurement for this 50 GHz gap. Yet, it is physically very unlikely that within this 50 GHz region a strong drop occurs where the values continue again with a flat response beyond 220 GHz. Further, we performed all other measurements above 110 GHz with a 1 GHz spacing between data points to ensure that we resolve all features.

[...] and also why a device bandwidth up to 430 GHz should be accepted as a record, provided that a flat response between 400 – 900 GHz has been already observed on a packaged device which of course works even at lower frequencies, and which exhibits at least one hundred thousand times more output electrical power using ~ mW optical input power.

The reviewer again compares an UTC diode based on III/V technology with a graphene detector. It does not seem to be proper to compare a new technology against an established technology. The idea of science is to explore new materials and technologies for their potential.

Anyhow, the UTC paper the reviewer is referring to reports a 3dB bandwidth of 340 GHz bandwidth. Even when ignoring the outliers in their measurement there is a 10 dB drop in the 400-900 GHz range. Therefore, it is certainly not flat and the 3dB 430 GHz bandwidth we report is higher.

Further, from the paper it is not clear that the frequency response would be flat down to DC. Many packaged PIN and UTC diodes use resonant RF circuits or inductive peaking to achieve better responses in certain frequency regimes at the cost of steeper cuts or blocking out of these bands.

		Lastly, said paper also uses a laser beating scheme to generate the RF tone. The detection of the RF signal is done with a power meter, which is less accurate than using an ESA and a frequency extender. Following the reviewer's line of argument, these are also not "bandwidth" measurements. So, are these reported values now relevant or not?
		Even providing these information, which would of course support this complicated measurement, this would just confirm that authors are measuring a device with 430 GHz bandwidth. There is no way that an ESA measurement without VNA and de-embedding devices can exclude parasitic elements limiting the bandwidth.
		We re-iterate here again on the already provided arguments (see our supplement and main text) that the reviewer did not seem to have read:  - The roll-off we observe follows a 60 dB/dec slope. A simple RC circuit forming a low-pass is a typical first-order filter with a 20 dB/dec slope. The third-order filter response points away from an RC limit. - A similar observation to the 60 dB/dec slope was made in literature for the PTE with pump-probe measurements (10.1038/s41566-022-01058-z) - We do not observe a change in the cut-off frequency on the PTE induced directional currents when gating the device and thereby modifying its resistance. - We do observe that the different detection mechanisms have a different frequency cut-off which cannot be explained by an RC limit. - We provided simulation data on the RF pad design. This data also point away from an RC cut-off.
3.		The title of the work starts with the words "Highest-Speed", from which we can derive that the authors consider these as a major achievement of their work. I agree, nevertheless, photo thermoelectric and bolometric detectors based on graphene reaching set-up limited bandwidths have been demonstrated in a plethora of papers in the last years. The ultrahigh speed of these detectors is a fact, supported by theory and by fundamental experiments revealing the ultrafast dynamics of hot electrons in graphene. For these reasons, while showing the record speeds of graphene detectors is very useful for the community, these achievements cannot be ascribed as breakthroughs or novelties anymore, but rather as useful information confirming known facts, as no real scientific/ technical originalities rise from these measurements.
		We thank the reviewer for his comment on the high-speed characteristics. We indeed believe strongly that the high-frequency response – and characterization – are some of the key results of this work. To the best of our knowledge, there is no other device technology able to achieve 420 GHz bandwidth and there is only one other demonstration showing >500 GHz with graphene. Nevertheless, we see that the title might not age well. We therefore changed the title to:
		Controlling Photothermoelectric Directional Photocurrents in Graphene with over 400 GHz Bandwidth
		The reviewer points out that there has been a plethora of setup limited graphene PD papers over the years. While this is true, one has to consider that a setup limit of 110, 70, 40 or even 10 GHz does hardly compare to 500 GHz. Furthermore, we do not experience a setup limitation, but find a device limited bandwidth of 420 GHz. We measure the bandwidth of the detection mechanism and not an RC limited cut-off. We therefore cannot fully agree with the last part of the reviewer's statement. The high speed of graphene PDs is not a fact and requires considerable engineering effort to achieve. We agree that the fast dynamics of hot electrons in graphene are a fact – but even here it is not well understood what the influence of substrate, passivation and material quality is. For example, one can look at a more recent study on graphene carrier dynamics (10.1038/s41566-022-01058-z). Here not only the gate dependence on the relaxation dynamics is tested but also the decay rates across four different

	samples with strongly varying carrier mobilities. The different graphene qualities lead to strongly varying carrier dynamics. Depending on the graphene quality, the carrier dynamics seem further influenced. The topic is therefore far from being entirely understood and additional measurements are required and should be reported. To highlight the novelty in our frequency characterization more strongly, we expand the discussion on the frequency response in the manuscript as follows:
	p10, l26: “When moving from an almost DC signal to 10 GHz (Fig. 5e), the regions previously associated with the PC effect are disappearing. Moving to 300 GHz (Fig. 5f) shows that the BOL effect (corners of the RF response map) is also weakening. At 400 GHz (Fig. 5g) the corner show almost no more features. We estimate a 3 dB cut-off frequency of the BOL effect in the range of 250 to 330 GHz (see Supplementary Information). Finally, at 500 GHz (Fig. 5h), where we are measuring in the roll-off of the response, the device performance is essentially bias independent except for the slope corresponding to the shift of the reference ground. These measurement results indicate a quasi-instantaneous Shockley-Ramo response for this device architecture of the PTE-DC, whereas the PC and BOL effects are limited to lower frequencies. The measurements thereby provide speed limitations of detection mechanisms within graphene and are not hindered by RC-limits or setup limitations. Thus far only one graphene PD architecture consisting of interdigitated electrodes exploiting the photovoltaic effect was able to exceed the frequency response presented here³⁶.” p12, l38: This can be exploited to find the individual frequency responses of the different detection mechanisms revealing that different high-speed photodetection mechanisms within the same graphene device have different bandwidths.
	I agree, this involves the right experimental procedures. There is no way, in my opinion, to probe such fast carriers dynamics using an ESA and subharmonic mixers contacting a device. These measurements are instead performed using other methods involving, e.g. pump-probe measurements. Using a VNA and proper procedures would be interesting, but in my opinion rather complicated.
	We respect the opinion of the reviewer that “there is no way, to probe such fast carrier dynamics using and ESA...” First of all, we are sorry, but we do not measure the carrier dynamics. We report the bandwidth of a photodetector. I.e. we show the performance of a whole device. There has to be a clear distinction between testing purely carrier dynamics and building functioning photodetectors and measuring the performance of a device. Second, measuring the actual device response rather than relying on pump-probe measurements is a much better representation on the device performance. As the reviewer correctly states, using a VNA would be interesting as it would give additional insights into the PD. Yet, these measurements are not only “rather complicated”, but almost – if not completely – impossible to do because the full set of required equipment barely exists.
4.	As the authors surely know, while frequency response is not an issue affecting graphene detectors, responsivity instead is. Specifically, addressing responsivity and linearity at the same time is the only real missing key performance parameter required to compete with SOTA. I completely agree with authors when they assert that “most of these approaches suffer from early optical power saturation onsets due to a high confinement onto a small active area”. Indeed, the big issue of graphene detectors is that when responsivity enhancement is achieved, saturation takes place at very low optical input powers. Instead, linear detectors present very low responsivity. This issue comes from the 2-dimensional nature of graphene and is still not solved. Concerning this, responsivity is not mentioned in the main text, but just in the supplementary information, and is very low compared to graphene SOTA detectors. It should be increased by x1000 to compete with SOTA graphene detectors or UTC-PDs

	exhibiting even higher bandwidths. At the same time, responsivity should be kept linear, with optical power levels > 0dBm before saturation, to compete with SOTA technology. Since addressing just one of these two KPIs is completely unuseful for applications, it would be interesting to give information about device response saturation, and how this is related to responsivity. Would a responsivity increase affect the response saturation against optical power, as happens for all reported devices so far? If they claim that this is not the case, why? Are there theoretical reasons? Do they have experimental evidences?
	We thank the reviewer for his comment on the responsivity. We agree that increasing the responsivity is one of the major challenges for graphene photodetectors. Here, we would like to stress that our paper offers a solution to the very challenge. What we show is that for high-speed applications what matter is the maximum achievable output signal which is the product of responsivity and saturation power or its linear range. Here we are able to compensate in part for the relatively low responsivity as we can operate the device at higher optical input powers. We can use higher optical input powers as we are spreading the power across a larger area in comparison to e.g., waveguide based devices. While the atomical thickness remains the same, the power per area on the photodetector is much lower in such a top illuminated device. So, we believe that even when we increase the responsivity of the device, we will not run into the same saturation issues as many other devices experience. We conducted an additional measurement on the linearity of the device which we added to the supplementary information. We swept the optical input power over more than 3 orders of magnitude. Here we now used a lensed fiber that we procured in the meantime. The smaller illumination spot (~5 μm) ensures that the optical power is coupled to the device's active area. The lensed fiber creates a better overlap compared to a standard single mode fiber, but also creates under-illuminated areas within the device. A slow loss of linearity in the 5-10 mW regime is observed. We comment on this further below on the reviewer's question on saturation. The changes are summarized in the adapted Supplementary Note 8 including an additional figure. We also added the responsivity to the main text:
	p7, l5: "This indicates that the measured total absorption is linked to the absorption in graphene and that the PTE-DC can indeed be enhanced with a MPA stack. The responsivity is found to be ~1.5 V/W and photovoltage signals >18 mV are extractable with the high optical power tolerance, see Supplementary Note 8 for more details."

Supplementary Note 8: Discussion on responsivity

Fig. S15: Voltage and current responsivity and linearity

a Device resistance and normalized photocurrent response as function of gate voltage. The maximum current responsivity is found at a gate voltage of 0.5 V where the device resistance is 2.75 kΩ. **b** Resonator orientation of the specified device and the polarization orientation convention used (blue – 0°, red – 90°). **c,d** Photoresponse as function of optical input power for the 0° polarization orientation and a wavelength of 1480 nm. The corresponding responsivities are extracted from the slope as **c** $\mathcal{R}_I = -0.462 \text{ mA/W}$ and **d** $\mathcal{R}_V = 1.4 \text{ V/W}$. **e,f** Photoresponse as function of optical input power for the 90° polarization orientation and a wavelength of 1550 nm. The current and voltage responsivities are **e** $\mathcal{R}_I = 0.424 \text{ mA/W}$ and **d** $\mathcal{R}_V = -1.6 \text{ V/W}$. The devices have been illuminated with a lensed fiber (focus spot $\sim 5 \mu\text{m}$) to ensure that all light is focused to the active area of the device. A loss of linearity in the 5-10 mW regime is observed.

At this point, what is still open is the performance in terms of responsivity. The presented devices in this work have responsivities of $\sim 1.5 \text{ V/W}$ or $\sim 400\text{-}500 \mu\text{A/W}$, see also Fig. S15. These responsivity values are considerably lower than the 27 V/W or 36.3 mA/W of the first demonstration⁷. However, the responsivity of the device in this paper can be increased by quite a few measures. ...

I do not agree, responsivity is very low.

If one takes a commercial fast detector, and adds 30 dB attenuation, one gets the same linearity and responsivity as the one reported here.

By just adding a low pass filter at the output of a commercial 100 GHz detector, one gets a 500 GHz detector with highest responsivity compared to the reported one. Also linearity would be comparable, maybe better.

So, what is the “offered solution” as authors claim?

Our statements are in comparison with other graphene based PDs and not with respect to commercial PDs. In comparison to most other graphene based PDs that focus the light to an extremely small volume, our solution is able to compensate lower responsivity with higher optical power tolerance due to the large area spread of the power. This, without sacrificing high-speed characteristics of the device.

We're not sure how the reviewer gets to a 500 GHz PD by adding a low pass filter at the output of a commercial 100 GHz detector. This statement makes no sense. More correctly, the reviewer should take a “high pass” filter. But even if he would do so, it would be interesting to see the high-

		pass filter of the reviewer and the photodiode that is only RC limited. The truth is that most photodiodes have more frequency limitations. If it were that easy, the community would have used such detectors for many years.
		I agree. The reported product of responsivity and saturation power is very low due to ultra-low responsivity. Moreover the onset of saturation at 10 mW (observed in this detector) is completely normal in detectors with x1000 responsivity.
		There is no graphene detector with a x 1000 responsivity and that has such a high power tolerance. As outlined in Table S1: the achieved voltage output of our device is larger than what most other PTE based graphene PDs have ever shown in literature.
		How exactly authors quantify this “low loss” of linearity? Usually, the 1 dB compression point is taken.
		We give the power sweep in Fig S15. This plot shows a high saturation point beyond 20 mW. We refrain from using the 1 dB compression point here, because the 1 dB compression point only marks the point at which the linearity is lost but gives no information about the slope of generated photovoltage vs. optical input power, saturation values or damage threshold. Seeing as we did not go to higher optical power values and experienced saturation, we describe the behaviour as low loss of linearity. Seeing that there are almost no graphene PTE based PDs that report rigorous power sweeps, it is difficult to make further statements.
		We believe that the photodetectors have the potential to reach the 0.1 A/W mark (as outlined in the supplementary information). [...]
		I don't see any rigorous argument supporting this belief
		As already discussed before, we provide a rough estimate in the supplementary information. We do not state any absolute value in our paper. We show the line of arguments for higher responsivities that a reader may follow or disregard.
		[...] With these values, they would match or outperform other graphene devices and also other technologies such as UTC-PDs. We also highlight that we provide an external responsivity unlike most WG based devices that deduct coupling losses and only provide an internal responsivity. We further want to note that, to the best of our knowledge, there is no UTC-PD with a bandwidth covering the low GHz up to 420 GHz, especially not without any external bias (SOTA zero-bias: 135 GHz 10.1063/5.0119244). Furthermore, the comparison to other technologies should also take into account other factors than responsivity and bandwidth. A graphene based device has clear

	advantages in manufacturability, integration capabilities and cost. Also, the first UTC-PD demonstration happened ~15 years before the first high-speed graphene PD demonstration and benefitted from decades of PIN-PD developments. We believe, if given time, graphene will be able to compete with the old and established technology which took many years to be developed.
	I agree on this, but it is not reported here.
	We don't know to which part of the section the reviewer refers here with his comment.
5.	Authors propose in the supplementary information some improvements to increase device responsivity. First, they claim that passing from low quality polycrystalline graphene to exfoliated graphene would increase the device responsivity. I agree, and this can be achieved also using single crystal graphene grown by CVD and encapsulated with hBN, since exfoliated graphene has nothing to do with real applications. Then, they compare their device with the one in 10.1038/s41467-020-20115-1. The comparison ends up with a multiplication which is in my opinion not realistic: mobility, number of layers and resistivity are decoupled, and considered separately, which does not make much sense: mobility and resistivity are actually directly connected, number of layers determine resistance and mobility as well... in other words, these parameters should be put together into a simulation and cannot be considered separately. Again, even reaching the (already demonstrated) value of 36 mA/W, what is the maximum optical power that can be coupled before saturation?
	We thank the reviewer for his comment on the responsivity improvement discussion. Indeed, the low quality polycrystalline graphene does not need to be replaced with exfoliated graphene but could also be replaced with monocrystalline graphene grown by CVD. To clarify that we refer to the quality and not the fabrication method, we change the sentence accordingly:
	SI p13, l6: "Moving from high quality mechanically exfoliated few layer graphene to CVD grown polycrystalline monolayer graphene can explain a large fraction of the deficit."
	Regarding the multiplication of the improvement factors. We agree that mobility, number of layers and resistivity are interconnected. We state that mobility influences the Seebeck coefficient and the resistance. We further say that the multilayer graphene influences absorption and resistance. In the end, we multiply: (Seebeck coeff.) × (Absorption enhancement) × (wavelength unit cell change) × (resistance) These coefficients are independent of each other, but as the reviewer pointed out correctly, linked to the graphene properties such as mobility and thickness. We stress the meaning of the multiplication by adding the meaning of each factor to the text:
	SI p13, l24: "Multiplying all estimated contributions leads to a total of (Seebeck coeff.) × (Absorption enhancement) × (wavelength unit cell change) × (resistance factor) $3 \times 4 \times 4 \times 5 \sim 10^2$

	explaining the two order of magnitude lower current responsivity in our devices.”
	I still say that this multiplication is not realistic at all, for the reasons that I have already given. My statement has not been considered by the authors. The resistance depends on the mobility, the Seebeck coefficient as well. scaling the resistance by a factor of 2 does not imply multiplying the responsivity by a factor of two. Also, the seebeck coefficient depends on the resistance (see its formula). Concerning the absorption term, authors are multiplying it by 4. The device as it is now absorbs the 60 % of the incoming radiation. How is it possible to absorb more than 100%?
	We stress here again that we give a rough estimate on the potential of the technology. It is a rough estimate to explain the two order of magnitude difference. We have considered the reviewer's statements and answered accordingly. We again repeat our statement in more detail:  - “Scaling the resistance by a factor of 2 does not imply multiplying the responsivity by a factor of two” – As outlined in our last response: Photovoltages generated by the Seebeck effect are linked to thermoelectric currents across a PD by Ohms law: $I_{ph} = V_{ph}/R$ or $\mathcal{R}_A = \mathcal{R}_V/R$. Therefore, the current responsivity will increase by a factor 2 if the resistance shrinks by a factor 2. The resistance can be controlled either by improving the carrier mobility or by making changes to the graphene and contacting layout. E.g., a rectangular channel or interdigitated fingers can be used to change the resistance of the device. - The photovoltage generated is in essence given by $V_{ph} = \Delta T * S$. We model the Seebeck coefficient with the Mott formula which we discuss in the SI. The equation is: $S(\mu, \varphi) = \frac{2\pi k_B^2 T_e}{3\hbar^2 v_F^2} \times \frac{\mu \mu_c}{\sigma_{min} + \frac{e}{\pi\hbar^2 v_F^2} \mu \mu_c}$ We also plot the calculated Seebeck coefficient in Fig. S3 for various values of the mobility μ and σ_{min}. As stated, if we improve the mobility by a factor of 5x, the Seebeck coefficient will increase by a factor of about 3x. This change in mobility will also decrease the device resistance which benefits the current responsivity of the device as discussed above.  - The ΔT is linked to the optical absorption in graphene. The 60% absorption that the reviewer refers to is the total absorption of the device, not the absorption in the graphene sheet. We already provided in the original SI (Suppl. Note 4) a route to increase the absorption by making changes to the metamaterial layer stack and provided in the last set of review details on how multilayer graphene would influence the absorption per material and element in the layer stack (Fig. S16). Again, we do not disagree with the statement of the reviewer that the properties are not interlinked. Yet, the current responsivity scaling will benefit from an increase in mobility, a decrease in resistance and employing multilayer graphene as we outline it.
	We note again, as mentioned in the manuscript, that this is only a very rough estimate to explain the two orders of magnitude lower responsivity values.

To estimate the saturation power when the responsivity increases is extremely difficult. Some literature reports that the responsivity for the PTE does scale with P_{in}^0 for low powers and later on scales with $P_{in}^{-1/3}$ (10.1038/s41467-021-23436-x). However, power sweeps and maximum output signals are very underreported in graphene based devices. To shed some light on the topic we summarized some of the PTE literature in the new table which we add to the Supplementary Information 8:

An overview of the state of the art of different graphene based PTE photodetectors is provided in the Table S1. The table lists the graphene fabrication technology (exfoliated vs. photonic integrated circuit), the operation wavelength, the voltage responsivity, the maximum reported photovoltage and the bandwidth.

Table S1: Comparison of graphene based PTE photodetectors sorted by publication year. “Fab” stands for the fabrication method of the graphene sheet (Ex. - mechanical exfoliation, CVD - chemical vapor deposition). “Int.” stands for integration scheme (PIC – photonic integrated circuit, FS – free space illuminated). λ is the illumination wavelength in nanometer. \mathcal{R}_V is the reported voltage responsivity. V_{max} is the maximum reported photovoltage signal. The last column summarizes the reported bandwidths (BW) of the devices. The larger than “>” symbol denotes setup limited bandwidths.

Ref.	Year	Fab	Int.	λ (nm)	\mathcal{R}_V	V_{max}	BW
/10.1038/nnano.2014.182	2014	Ex.	FS	100'000	715 V/W	0.0027 mV	3.2 GHz
/10.1021/acs.nanolett.6b03374	2016	Ex.	PIC	1550	3.5 V/W	---	65 GHz
/10.1021/acsp Photonics.8b01128	2018	Ex.	PIC	1550	4.7 V/W	1.5 mV no power sweep	>18 GHz
/10.1021/acs.nanolett.9b02238	2019	CVD	PIC	1550	12.2 V/W	2.9 mV	42 GHz
/10.1038/s41467-020-20115-1	2020	Ex.	FS	4000	27 V/W	0.35 mV	>4 kHz
/10.1021/acsnano.0c02738	2020	CVD, monocryst.	PIC	1550	6 V/W	4 mV no power sweep	>67 GHz
/10.1038/s41467-021-21137-z	2021	CVD, monocryst.	PIC	1550	3.5 V/W	15.1 mV loss of linearity	70 GHz
/10.1038/s41467-021-23436-x	2021	Ex.	PIC	1550	90 V/W	38.6 mV loss of linearity	12 GHz
/10.1038/s41566-021-00819-6	2021	Ex.	FS	4000	15.6 V/W	6.8 mV	0.52 MHz
/10.1038/s41467-022-31607-7	2022	CVD	PIC	5200	1.5 V/W	0.015 mV no power sweep	> 1 MHz
/10.1038/s41566-022-01115-7	2023	Ex.	FS	4000	392 V/W	3 mV	> 0.39 MHz
This work	2024	CVD	FS	1550	1.6 V/W	18.5 mV loss of linearity	420 GHz

With this table and the above new measurements which we provided with respect to reviewer's comment #4, there is evidence that optical inputs beyond several mW of optical power are possible. Evidence that higher saturation powers are possible even with higher responsivity can be seen by the maximum measured signals. Here we measured $> 5 \mu A$ limited by the low current responsivity, but $> 18 mV$ in photovoltage output – the 2nd highest reported value. In 10.1126/science.adg8017 we've demonstrated $\sim 50 \mu A$ output signal with a free-space illuminated graphene PD. Recently we measured generated photocurrents $> 100 \mu A$ in free-space illuminated graphene devices. However, these results are not yet published.

As the PD can be seen as a distribution of smaller PDs over an area, there is also the possibility to increase the active area of the device by adding more of these elements. This in turn would again

		allow higher optical input powers as the power is spread across a larger area. However, how the responsivity and RF response is influenced by the scaling is not yet clear. [...]
		Yes it is, it can be checked in the supplementary information of 10.1038/s41467-021-21137-z, and it is a textbook exercise.
		The model stated by the reviewer as a textbook exercise does obviously not capture the behaviour accurately enough. According to the lumped element modelling of the cited paper their device should reach a 3 dB bandwidth of 300 GHz, whereas measured values are at 70 GHz. Therefore, the model clearly does not capture the behaviour accurately enough to call it a standard textbook exercise. More importantly, the calculations in the cited paper are based on a completely different architecture in form of a waveguide based PD where the light is guided along the graphene. The calculations provided there are in our opinion not directly applicable to our device design. For example, the illumination and heating happen from one side and elongating the device changes the fraction of absorbing graphene to non-absorbing graphene.
		[...]Depending on the application there might ultimately be trade-offs between RF bandwidth, responsivity, sensitivity and saturation. Further research on the topic is required.
		Completely agree on this

6.1		Having said that application relevant performances are in my opinion not considered in the main text and are revealed to be low in the supplementary information (even taking into account the list of improvements in the supplementary file), application relevant concepts and functionalities enabled by this graphene detector could be sufficient to justify a novelty. Nevertheless, the proposed concepts and functionalities are not peculiar to this device and the arguments of the authors about this aspect are quite weak. -Authors claim that their device supports operations in a broadband optical range. They mention many times the MIR window. Moreover, they assert that other solutions like, e.g., waveguide-integrated devices would not afford such "broadband" operation. First, the paper does not show any MIR operation, apart from a supplementary section showing a simulation. Second, the operation of the proposed device is wavelength dependent as for integrated photonic devices based on, e.g., silicon nitride or Si, which design can be optimized to operate at a central wavelength, with optical bandwidths chosen to trade-off efficiency. Integrated photonic waveguides can operate in ranges even larger than those experimentally considered by the authors, in many cases with very low design efforts. Moreover, the wavelength tuneability of the proposed device requires lithographic accuracies in the order of tens of nm, which is not exactly of easy implementation using standard photonic technology. Indeed, the real device central wavelength operation is different from the designed one, as also observed by the authors. Therefore, I really do not see the author's point in this claim. At least, I don't see why integrated solutions are weaker than the proposed one.
		We thank the reviewer for his comments on the applications and wavelength operation range. The MIR operation is indeed only showed in simulation in our work. At the time of writing, we had no access to a MIR light source and did therefore not fabricate any active metamaterial devices for the MIR range. However, in the introduction we refer to many other works that showed detection in the

	MIR as well as our own previous work, which is based on the same metamaterial perfect absorber material stack. We rewrote a sentence in the introduction that might give the impression that we show experimental MIR operation of the photodetector:
	p2, l32: "Here in this work, we investigate the photothermoelectric induced directional currents (PTE-DC) within a metamaterial perfect absorber (MPA) architecture for its viability to serve as infrared photodetector-both the needs of the NIR and MIR detection regime and thereby uncover its quasi-instantaneous frequency response."
	The MIR operation that we do discuss is two-fold; (1) the Dirac cone band structure of graphene which allows broadband operation and (2) the top illuminated metamaterial architecture. The reviewer argues that there is no real advantage in aspect (2) as photonic integrated circuit (PIC) approaches could also offer broadband operation. From our view, the design of a PIC platform allowing for single mode operation over a broad wavelength range is not trivial and does not come with low design efforts. First, we note that the discussion does not include all platforms:  - InP: the SOTA platform for high-speed photodetector technology as it allows for the high quality growth of III-V materials. Here, the wavelength range is limited to a window of ~1000 to 2000 nm. Furthermore, the substrate itself is expensive and scalability of fabrication processes is difficult. They are therefore excluded from the further discussion. - SiN and Si. These platforms can indeed also operate within the MIR regime. To the point of the design effort: Si and SiN PICs have been shown to operate in the MIR range, but the design efforts are considerable. As an example, typical SOI platforms used for NIR applications have Si thicknesses of 220 nm. Such thicknesses are inefficient for MIR operation as optical modes extend largely into the buried oxide in these types of stacks. The oxides induce large losses. One solution is to change the material thicknesses which in turn again limits the spectral range achievable on a single substrate. A good perspective on the topic can be found in 10.1515/nanoph-2017-0085. Other approaches without changing the material layer stack include photonic crystal structures (10.1364/OE.448284). Again, here the design effort is considerable, and the fabrication is lithographically heavy. This discussion further only covers the waveguide itself; additional passives such as couplers, bends, splitters, etc. also require an appropriate design for each target wavelength that they would be used. Each element also introduces losses until the light is guided to the active photodetector region. We want to highlight that we do not make any changes to our layer stack, not in the material composition nor the material thicknesses. The only element that is changed is the resonator shape. We therefore strongly believe that our approach is far easier to tune to a target operation wavelength than a PIC-based photodetector. Regarding the reviewer's comment on the lithographic accuracy: This aspect is true that we did not hit the targeted resonance wavelength in this first iteration. All our active devices have only been designed with one resonator shape to allow us to perform more comparative measurements on properties such as the resonator orientation. We note however that we are only 70 nm off from the target (1480 nm instead of 1550 nm) and that the resonance offers a broad peak with a FWHM of ~200 nm, still allowing us to operate the device with only little deficit at 1550 nm. Furthermore, adapting the simulation framework to include more realistic shapes (e.g., rounded corners) that we can now extract post-fabrication, we strongly believe that we can improve the accuracy of the metamaterial design even further in future works. We highlight that Fig. 2 illustrates the successful

	tunability of the structure and Fig. S10 shows that a size change of 14 nm already yields a very good overlap between measurement and simulation. It has been further shown that nanometre scale metamaterial structures can be efficiently and accurately fabricated with nanoimprint lithography if mass production is a target, e.g., discussed also in this recent review and perspective 10.1021/acsphotonics.3c00457. However, this is clearly beyond the scope of this paper.
6.2	-The discussion of SOTA on polarization/ wavelength demultiplexing is not present in the text. How does their polarization /wavelength demultiplexing compare with already existing systems in terms of efficiency and compactness (also considering the space occupied by the active polarization control)? Polarization/wavelength demultiplexing is a function which can be easily performed not only for two wavelengths. It is implemented for many wavelengths in both CWDM a DWDM systems, also using integrated photonics. ...
	We thank the reviewer on his comment of the demultiplexing application. We first want to point out that the demonstrated device combines the two functionalities (wavelength division demultiplexing and polarization demultiplexing) simply as a proof-of-concept device. The goal of the device is foremost to show that more complex metamaterial unit cells can be employed and do not perturb the overall effect, and in addition provide access to advanced functionalities. We do not claim – or even target – that we would outperform any other wavelength or polarization demultiplexing schemes at this time. We made the following change to the manuscript:
	p10, l47: “We now want to advance the use of the PTE-DC further by showing a proof-of-concept on how two independent detectors can coexist within the same active area.”
	- Authors do not discuss what are these advanced functionalities - Therefore, I respectfully ask to the authors: what is the point?
	The reviewer seems to have missed the respective passage in the paper. In our paper we clearly have stated two advanced functionalities. These are: wavelength division demultiplexing and polarization division demultiplexing.
6.3	... In addition, the scheme proposed by the authors is sensitive to polarization when “demultiplexing” two wavelengths, and sensitive to wavelength when detecting polarization. Moreover, active polarization control is needed. What is the application for such a detector? ...
	We further agree with the reviewer that the combination of both mechanisms in the same device does not lead to any direct advantage. However, we note that there is the possibility to also build devices that you can do either of the two things.  - (1) Devices that work for a single wavelength and can detect two signals with different polarizations or - (2) Devices that work for two different wavelengths but are otherwise polarization independent.

	Number (1) is trivial as one can simply make all resonators identical. Number (2) requires a completely different metamaterial resonator geometry to achieve the polarization independent, wavelength directed photocurrents. We do have theoretical and simulation work that shows that it is possible. However, these preliminary results are beyond the scope of this paper. The applications of these two types of detectors are for example linear polarization state detection or sensing of two different wavelengths at the same time with a single device which can for example be beneficial in environmental sensing applications such as gas detection. Additionally, the high speed of the devices could be used in telecommunication applications. With illumination from a single fiber, it is possible to have essentially two detectors occupying the same space. As such, the bandwidth per area can effectively be doubled. Here either wavelength division demultiplexing or polarization demultiplexing are possible schemes. We note here further that the possibility of the wavelength tunability of the resonators gives access to wider spectral bands, e.g., around 2000 nm which are foreseen to be used in future telecommunication applications.
6.4	... The “small” crosstalk of 3% is 15 dB, which is in my opinion not interesting for applications, where >> 20 dB isolation between the channels are required in systems in which many wavelengths are managed. How can this be solved, also accounting that real applications require >> two wavelengths? How the responsivity and wavelength selectivity scale when considering more than two wavelengths?...
	We thank the reviewer for his comment on the crosstalk and the WDM application. We agree with the reviewer that the crosstalk would ideally be -20 dB or lower. We believe parts of the crosstalk are the relatively broad resonances of the metamaterial structures and the contacting scheme that we employed. With the peaks showing FWHM values of ~200 nm, there is an overlap in the absorption spectra of the two resonators. Ideally one would choose the peaks further apart to show the true crosstalk of the device. However, we did not have any other laser sources available apart from the O- and C-band lasers which is the reason why we chose these specific wavelengths. Secondly, the contacts were designed to be only one period away from the metamaterial structures. Therefore, we have large sections of contact doped graphene at the contact edges which will also contribute to the overall photoresponse which leads to an additional crosstalk. On the question of additional wavelengths: as the device encodes wavelength on a spatial direction in a 2D plane, there is only the possibility to truly support two wavelengths. However, due to the small size and direct top illumination it is of course possible to have space division multiplexing (SDM) of the devices, effectively halving the amount of space required as two photodetectors are built into a single device. If one is interested in simply differentiating between a set of different wavelengths, it is possible to design a polarization independent variant that has four distinct resonances. Then, each λ is encoded in the four spatial directions $+x, -x, +y, -y$. In such a scenario a mixture of wavelengths will however lead to currents that cancel each other out.
6.5	...What is the problem that this scheme could solve and that is not already solved with existing systems?
	As discussed above, there is a variety of applications that could benefit from such a scheme. To achieve the same functionality, one would always require several photodetectors and additional elements. E.g., in free-space illumination the optical signal will need to be split or dispersed to illuminate each PD with the specific functionality. In a PIC approach the same is true with passive

	elements such as polarization beam splitters or wavelength division demultiplexing schemes. We add the following to the conclusion in the main text to put a stronger focus on potential applications:
	p12, l40: "We further leveraged the directionality of the induced currents by introducing a four contact pad design with a supercell metamaterial. This design highlights the flexibility of the PTE-DC. Direct access to demultiplexing schemes is enabled due to the two perpendicular channels. Furthermore, the flexibility in the metamaterial design and graphene's absorption characteristics enable tunability over a large spectral range. The scheme offers thereby the potential of operation in the MIR spectral range where high-speed detectors do not exist. Additionally, they offer the flexibility to provide built-in functionalities such as polarization state detection, dual wavelength detection and two signal demultiplexing within the same active area removing the need to split-up signals or for bulky constructions. Further, the results clearly highlight that future endeavors on graphene photodetectors should not simply focus on the optimization of the absorption but have to rely on a multidisciplinary modelling and design approach. Therefore, photothermoelectric induced directional photocurrents in graphene are an excellent mechanism to further the understanding in the material, physics, and device design, with the promise of strong performing highest-speed photodetectors."
6.6.	In conclusion, my general opinion is that this manuscript shows a work based on known fundamental phenomena. This fact by itself would not be a point for negatively judging the paper, if new application-relevant novelties or ideas would have been shown. But from this standpoint, it presents just a well performed engineering exercise (which certainly could be suggestive for further developments), with no sufficiently relevant application perspectives.
	We thank the reviewer for his detailed comments above and his concluding remarks. We appreciate the positive assessment describing the manuscript as very clear, instructive, and well written. We also value that our engineering work is being complimented. We strongly believe that bringing known fundamental phenomena into a functioning device structure should not be underestimated and can be considered as novelty on its own. A first demonstration of a PTE photodetector with more than 400 GHz bandwidth is worth reporting on as no other technology platform was able to achieve such numbers. Following the above discussions, we also added the relevant applications that could benefit from such a device. While at the moment the device responsivity is still lacking, we hope the reviewer can appreciate that this is in essence a first demonstration of a PTE photodetector with 100s of GHz bandwidth and that there is room to enhance the responsivity. We note that we did not only improve upon the speed of previous demonstrations, but we also introduced a design guideline to include the asymmetric resonators in a passivated metamaterial perfect absorber structure. This allows for operation with low gate voltages (CMOS compatible sub 5V compared to 80-120V of the previous reports) and also stable operation in ambient condition. With the promise of additional functionalities, we strongly believe that this type of high-speed device architecture will be useful. We answer below the additional points the reviewer added:
7.	Authors should put the real image of the RF device with the dimensions.
	We thank the reviewer for pointing out the lack of an image of the RF devices. We're of course happy to add an image of a device. We assume this comment is related to the RC-cut off in the next comment and will add the image in the discussion below.
8.	As far as I understood, the device length is 10 μm. Apart from transit time and quasi-instantaneous response of PTE, one needs to consider the pads capacitance which is most probably responsible for the frequency cut-off. Considering that the resistance of the device is 6000 ohms, I get $\sim 5\text{fF}$ parasitic

pads capacitance to get ~450 GHz, which is impressively good! Maybe the authors could add information about that, and about the RF pads design. Also, information about the substrate resistivity should be included to show that no RF ohmic losses occur due to substrate.

We thank the reviewer for his excited comment about the RF pad design and RC cut-off. First, we comment on the reviewer's calculated capacitance. It is important to consider the full electrical circuit when operating the device in a high-speed application.

The photodetector resistance is essentially in parallel to the load resistance. In a high frequency circuit this corresponds to 50 Ω. The relevant resistance for an RC cut-off of the system is thereby $\left(\frac{1}{6000\ \Omega} + \frac{1}{50\ \Omega}\right)^{-1} \approx 50\ \Omega$.

Further, the RF pads are a travelling-wave design which makes the lumped capacitance of the pads not directly relevant. To discuss the RF pad design, we added a new section in the supplementary information that discusses this in more detail. It reads as follows:

Main text, p9, l3: “Here, we trim a device for high-speed. We have used RF simulations to find an architecture that does not limit the bandwidth of the device but ideally reveals potential speed-limitations of the photodetection effect (see also Supplementary Note 9).”

Supplementary Note 9: Discussion on Frequency Response and RC cut-off

Fig. S17: Radio frequency pad design.

	a Top view of the full device geometry implemented in the simulation environment (CST Studio). The contact pads (gray squares) are connected to an optimized co-planar waveguide section. The active area of the device is represented in red, which is implemented as a lumped element. b Corresponding optical microscope image of a fabricated device. c Side view of the layer stack within the simulation environment. The zoom-in shows the active region layer stack. The substrate is silicon (intrinsic) with a 1μm thick thermally grown silicon oxide layer. Within the silicon oxide the gold gate is buried. A layer of aluminum oxide is on top followed by the active area of the device, and the ground and signal lines. A 50 nm aluminum oxide passivation layer is added on top. The structure is surrounded by air. d Simulated RF response from 0 to 500 GHz for the above depicted geometry. No roll-off behaviour is found. To ensure ideal operation at high frequencies we optimized the contact pad design. Fig. S17 illustrates the design. We implement the structure in CST Studio and perform time domain simulations. The RF pad design is illustrated in Fig. S17a. Three contact pads (gray squares) modelled as perfect electric conductors are connected to the co-planar waveguides (CPW) forming the Ground-Signal-Ground (GSG) structure. The CPW was optimized in a first step to match the wave impedance of the contact structure to the 50 Ω of the RF circuit. For this, only the CPW structure connected terminated with two ports have been simulated separately. The active area of the device was fixed at 10 μm as well as the pitch of the pads which was set to 50 μm. Therefore, the parameters to optimize were the widths of the metal lines, the gaps between the lines and the length of the CPW. These parameters were optimized until the line impedance was close to 50 Ω. We note that the material layer stack illustrated in Fig. S17c is determined by the metamaterial design as illustrated in Fig. S8. Using the optimized CPW geometry we simulated the full device as illustrated in Fig. S17a. The active area of the device is here simply approximated as a lumped element with a resistance of 6000 Ω. At the same spot a port for the excitation signal is implemented mimicking the photoresponse of the PD. The transmitted and received signal is then detected at the gray GSG pads which are terminated with 50 Ω. The resulting simulated RF response is shown in Fig. S17d. No roll-off behaviour is found in the range from 0 to 500 GHz. Oscillation in the spectrum stem from small impedance mismatches between the active area and the probing. Strong drops in the frequency range have been identified as resonant coupling to the gate pad. In the measurements we did not experience these sharp drops which we assume stems from damping in the gate due to resistive losses.
	I assume that CPD is the pad capacitance. This circuit has ~450 GHz bandwidth if CPD is 7fF (I wrote ~5 fF in my previous comment, but the correct number is 7fF). By looking at the image of the device that the authors put in this new version, I get even 9fF pad capacitance (https://www.emissoftware.com/calculator/coplanar-capacitance/) considering just silicon oxide as the substrate, eps=3.9, therefore effective epsilon around 2.45 (calculated from a modal simulation). Considering the silicon substrate, the effective epsilon is higher, thus things get even worst. I consider 2 parallel pads with dimension of 72x72 with 14 um gap. I took these values from the scale on the image that the authors have attached. Capacitance could be even higher in reality which means, lower frequency cutoff. From this considerations, my point is: there is no experimental evidence that the cutoff that authors show is intrinsic to the device, it can be due to the parasitics (and anyway is outstanding), therefore no conclusions can be done on it. This is the reason why carrier dynamics are measured with more reliable methods.
	First off: we don't understand the constant criticism that we're probing carrier dynamics wrongly. We report on a graphene based PD and its performance metric as a function of operating conditions. In no way or form do we argue that our measurement method is superior to other methods to characterize carrier dynamics in a graphene.

		Regarding the discussion on the capacitance: The reviewer probably agrees that our full 3D simulations taking into account the full material stack, material properties and exact dimensions including tapering is a more accurate representation on the device parameters than a web toolbox. Nevertheless, the rough estimates the reviewer provides essentially agree with our measurements and simulations that the pad capacitance is small enough so that we can probe the bandwidths of the devices detections mechanisms.
		How can an RF pad be “travelling wave”? They are lamped by definition. Also the electrodes that connect the pads to the graphene active area do not represent a transmission line (or a “travelling wave” electrode, adopting the authors vocabulary). Considering around 2.5 effective epsilon, one wavelength at 500 GHz is around 400 um, which is longer than the entire electrode.
		The RF pads are designed as co-planar waveguides. Co-planar waveguides are by definition planar transmission lines. There is a quasi-TEM mode supported by such a structures which is propagating. A lumped element consideration is only valid when the characteristic size of the circuit elements is much smaller than the wavelength – which as the reviewer points out, is here clearly not the case. So the full wave treatment and 3D simulations we perform are exactly for said reason.
		Some remarks: -according to the figures and to the device picture, the pitch is not at all 50 um. -What is the meaning of an RF simulation in a simulation area that is not even one wavelength long? -How can the RF response of a 6000 Ohm resistor connected to a 50 Ohm load be 0dB?
		 - The probe pitch of the used GSG probes is 50 um. The points of each RF pad where the probe is in contact is exactly 50 um. - We provide an image on the pad design, I'm not sure how the reviewer can make comments about the simulation domain or simulation setup from an image. Then we again have to wonder what is the reviewer taking offense with? The low frequency range of the simulation where there are long wavelengths or the 500 GHz regime where the wavelengths are actually shortest? We do not look at radiating waves but at quasi TEM modes in a CPW configuration. - The reviewer is correct, we made a mistake in the axis label. The y-axis should of course read Normalized Simulated RF Response (dB) to match the y-axis on the measurement values in the main text.

What are the resistive losses the authors are referring on? Thin metal? This would be present also at the device level and would contribute to frequency cutoff, since losses contribute to a cutoff. These resistive losses are not even modeled, for example the substrate is considered as intrinsic. I asked for the resistivity of the substrate but I received no answers on this.

We refer to the resistive losses of the metallic backplane to the gate contact pad. This connection line was purposefully kept thin and narrow in the regions where we expected coupling of the frequencies dips that we observed in simulation to circumvent the problem. The material values used in simulation appear to be less lossy than for the fabricated devices. This is for example due to roughness in the film.

Our simulations include the modelling of all the materials including losses. We do not use perfect electrical conductors or perfect insulators. The reviewer again makes baseless statements.

Regarding the resistivity of the substrate: indeed, we missed this question in the original review. This value does have actual meaning. The substrate is a wafer bought at MicroChemicals, intrinsic/undoped with a specified resistivity of 10000 - 1000000 Ohm cm. We added this value to the manuscript.

The photodetectors were fabricated on a generic **intrinsic silicon substrate with a specified resistivity of 10000 - 1000000 Ohm cm** on which 1.5 μm of thermal silicon dioxide were grown.

9. Given the low power level of the detector, authors should add more information about the noise of the equipment they use, and the calibration of the same. How did the authors calibrate the measurement? Did they use reference signals generated by, e.g., commercial UTCPDs with known frequency response? The information about the measurement is, in general, not exhaustive, given the large, spanned bandwidth and the different components used in the different frequency ranges.

We thank the reviewer for his remark on the high frequency characterization methods. As discussed in the paper, we essentially use two different setups to characterize the frequency response of the device. (1) In the range of 3-110 GHz directly with an electrical spectrum analyzer (ESA) and (2) with

additional subharmonic mixers in the bands from 110-170, 220-330 and 330-500 GHz. For the calibration of the system:

- (1) The path here consists of a GSG-probe, a bias tee and a RF cable. We calibrate the probe losses with the calibration curve provided by the vendor. The bias tee and RF cable losses are compensated by measuring a high-speed reference PD once directly attached to the ESA and once with the additional components in the RF path.
- (2) The path here consists of a GSG-probe directly attached to a subharmonic mixer. The probe losses are again compensated with the loss calibration provided by the vendor. The loss or respective conversion efficiencies of the mixers are compensated with the calibration data provided by the vendor.

The noise floor strongly depends on the frequency range of interest but is with our equipment and settings typically around or below -90 dBm. As an example, the noise floor at 400 GHz is -94 dBm and the measured RF power of the device is -74 dBm, leaving us a 20 dB span above the noise floor.

We expanded the discussion on the high frequency characterization in the method section to provide more details:

High frequency characterization:

To characterize the RF frequency response of the device we used a laser beating setup as depicted in Fig. 5b. Two lasers are combined in a single fiber where one laser is fixed at 1550 nm and the other is detuned by a difference frequency $\Delta\lambda_{RF}$. The beat-note at $\Delta\lambda_{RF}$ was measured by capturing the device response with an electrical spectrum analyzer (ESA). The ESA (Keysight UXA N9041B) offers a frequency range up to 110 GHz. The device was connected to the ESA with high-frequency GSG-probes, a bias tee and an RF cable. The probe losses are compensated with the loss calibration provided by the vendor. The bias tee and RF cable losses have been calibrated by performing reference measurements using a commercial high-speed photodetector. Frequencies above 110 GHz were accessed by employing sub-harmonic mixers (Virginia Diodes, SAX modules) connected to the ESA. The high-frequency probe losses are again compensated with the loss calibration provided by the vendor. The loss or respective conversion efficiencies of the sub-harmonic mixers are compensated with the calibration data provided by the vendor.

There is something that I really don't understand. Let's consider $500\mu A/W$ responsivity, as declared. Let's consider 1 mW optical input. The generated current is 500 nA. Let's consider the device model, without parasitics:

Authors can easily check that this corresponds to a power of ~ -90 dBm at the subharmonic mixer. Let's consider the best Virginia diode SAX model in terms of SSB conversion efficiency. This has 10 dB loss per sideband, this means 13 dB loss. Let's consider 10, for simplicity. This means that the signal, without any kind of loss (which is physically impossible) should be ~ -100

	dBm at the ESA. Well, authors can state that they used 10 mW optical input, which means ~ -80 dBm at the ESA with no loss. This means that, to get -74 dBm with such extremely low responsivity, even without any kind of loss (and negligible loss are physically impossible at such frequencies), authors used > 10 mW optical input. I would rather say 30-50 mW, even more, realistically speaking. If we then consider SSB losses of the virginia diodeds SAX modules at higher frequencies, well, things become even more complicated. I think that authors must clarify this, in whatever journal they publish this work. The exact optical power coupled to the device has to be declared, and this must correspond to an electrical signal level extracted from the already declared DC responsivity. This signal level must be compared to the noise level of the used equipment chain, also considering losses. Moreover if huge optical power has been used to extract this ultra-low electrical signal, authors must revise the RF measurement which does not correspond anymore to a small-signal regime. At this regard, authors should also revise their experimental conclusions since they are working in a saturation regime according to their linearity measurement
	The reviewer is correct with his calculated estimates but is wrong with how the mixers operate. The reviewer's 0 dBm input corresponds to the mentioned power of ~ -90 dBm. 10 dBm therefore leads to values ~ -70 dBm. We set the optical power to 12.5 dBm or 17.8 mW without using a lensed fiber, i.e., there are some excess losses with the coupling. We report the probe losses above on page 9. They account for ~ 1-5 dB losses depending on the frequency regime. As elaborated above, the mixer and ESA work in unison and any conversion losses are directly taken into account by the instrumentation. A calibrated RF power is thereby directly read-out at the ESA except for the probe losses. The noise floor measured in the ESA is also taking into account any conversion limitations by the subharmonic mixers. We chose the optical power value such that we have sufficient SNR (as mentioned, roughly 20 dB or more over the noise floor) but without saturating the PD. We add the optical power for the RF measurement to the method section:
	Two lasers are combined in a single fiber where one laser is fixed at 1550 nm and the other is detuned by a difference frequency $\Delta\lambda_{RF}$. The optical power at the fiber output was set to 12.5 dBm.
10.1	Apart from the "multiplication" approach discussed in point 5, that is in my opinion not appropriate, I have some concerns on the observations done in the calculation: -Metal ontop of graphene pins the chemical potential but can also degrade graphene quality and so its mobility. Maybe low mobility graphene would not be strongly affected from metal decoration, but high quality could, since high quality graphene is more sensitive to defects. Considering this, are authors sure that higher quality graphene would give 3x mobility if metal is put ontop?
	We thank the reviewer for his comment on the metal pinning. This is indeed a very interesting point. We first want to note that we did not make any comments on what mobility our device could reach. We merely argued that with our relative low mobility of $< 1000 \text{ cm}^2/\text{Vs}$ a moderate improvement in the mobility of factor 5x would lead to an increase in the Seebeck coefficient of a factor 3x. The work we compare ourselves to in this estimation does not report any mobility values and therefore we

		chose a relatively low mobility value for high quality graphene. We are aware of the negative effect of metal contacts on the carrier mobility in graphene. However, there are studies that demonstrate that even with high impurity concentrations it is possible to achieve several 1000 cm^2/Vs in mobility, e.g., (10.1038/s42005-021-00518-2). Therefore, we strongly believe that this improvement in Seebeck coefficient is achievable.
		(with respect to the potential mobility the device could reach). Well, that should be done in my opinion
		As the graphene quality due to fabrication improvements keeps improving, naming an absolute number beyond the factor 5x that we estimate to be achievable seems far fetched which is why we do not want to make such a guess statement.
		I did not speak about impurities concerning metal.
		We ask the reviewer to please re-read his comment. Most certainly the comment was with respect to impurities concerning metal.
		I'm not aware on papers showing > 5000 cm^2/Vs mobility in Coulomb-dominated scattering regime on SiO2 substrates.
		Firstly, our device is not fabricated on SiO2, but on Al2O3. Second, one can find several reports of mobilities on a dielectric substrate that reach such values. Some reports that show these values: /10.1063/1.3483130 – 4500 cm^2/Vs (SiO2) /10.1063/1.3077021 – 8000 cm^2/Vs (SiO2) /10.1088/0957-4484/21/1/015705 – 7400 cm^2/Vs (Al2O3) /10.1126/sciadv.abk0115 – 9500 cm^2/Vs (Al2O3 / Sapphire)
10.2		- Concerning resistance by itself, why does decreasing resistance by 5 means x5 responsivity? Putting low impedance detectors in parallel does not necessarily mean increasing responsivity, this can be easily checked using simple equivalent circuits. Maybe authors refer to the increased electrical power transfer from the detector to the load, which would rise because of impedance matching? Indeed, the electrical power transfer would rise by > 30 dB while passing from 6kohm to 50 ohm. Anyway, this is not related to (DC) responsivity in my opinion but only in power transfer efficiency from the detector.
		We thank the reviewer for his question on the link between resistance and responsivity. As PTE based devices are typically seen as generating a photovoltage and not a photocurrent the device resistance directly also influences the responsivity in terms of photocurrent. Photovoltages generated

	by the Seebeck effect are linked to thermoelectric currents across a PD by Ohms law: $I_{ph} = V_{ph}/R$ or $\mathcal{R}_A = \mathcal{R}_V/R$. This can be seen in various examples in literature:  - $\mathcal{R}_V = 27 V/W$ or $\mathcal{R}_A = 36.3 mA/W$ with a device resistance of 500 – 1000Ω (10.1038/s41467-020-20115-1) - $\mathcal{R}_V = 3.5 V/W$ or $\mathcal{R}_A = 35 mA/W$ with a device resistance of $\sim 110\Omega$ (10.1021/acs.nanolett.6b03374) - Split-gate PTE configuration where ideal voltage responsivity (high resistance) and ideal current responsivity (low resistance) is identified as a function of two gate voltages. (10.1038/s41467-022-31607-7) To clarify this point, we expanded Supplementary Note 8 and added the new Figure S15 showing the above mentioned new recorded power sweep where we measure once the photovoltage and once the photocurrent:
	SI, p13, l23: Lastly, the resistance difference between our device and the previous report is a factor 5x. As photovoltages generated by the Seebeck effect are linked to thermoelectric currents across a PD by Ohms law the resistance has a direct influence on the current responsivity. This effect is seen also in Fig. S15; The voltage responsivity divided by the current responsivity leads to $\sim 3000 \Omega$ matching the device resistance in the operation point.
	There is a clear theory beyond this, relating mobility, resistance, seebeck coefficient and cooling time, ultimately predicting responsivity. This theory can be of course trimmed, but is the only path that triggers designs. I honestly do not approve this method of multiplying things based on a couple of papers.
	The reviewer requested an answer as to why a resistance increase would result in a higher current responsivity. We give a clear answer based on a clear theory: Ohm’s law for PTE based devices. The reviewer’s statement with respect to our answer seems like the reviewer did not read the answer at all. We do not “multiply things based on a couple of papers”

REVIEWERS' COMMENTS

Reviewer #2 (Remarks to the Author):

I have thoroughly reviewed the lengthy peer review comments and the authors' responses. Overall, I still believe this is an outstanding piece of work, and it is one of the few studies in recent years that has genuinely excited me. I consider the quality of this work to be well above the average level of Nature Communications.

Specifically, what intrigued me the most is their systematic investigation of the frequency response of various photoelectric mechanisms in graphene using centrosymmetry-broken metal structures, with the photothermoelectric effect (PTE) showing a working bandwidth exceeding 400 GHz. While many previous studies based on carrier dynamics have predicted that graphene detectors could achieve very high operating bandwidths [Nature Nanotech 7, 114–118 (2012); Nat. Photon. 16, 718–723 (2022)], testing the carrier lifetime in materials with a pump-probe system is entirely different from achieving high-frequency photoelectric conversion in devices!

In fact, I believe the value of this work even surpasses that of the authors' recent publication in Science (Koepfli et al., Science 380, 1169–1174, 2023). Although the Science paper employed an interdigitated electrode structure with impressive results, the photocurrent primarily generated near the electrodes allows photogenerated carriers to be collected without much delay. However, in this paper, the photocurrent is actually generated at locations in the device far from the collecting electrodes, yet it still achieves such a high response frequency. This demonstrates the unique nonlocal nature of graphene's photoelectric response. While this might be anticipated based on the Shockley-Ramo theorem [PRB 90, 075415 (2014)], previous works have been limited by testing systems [\sim 900 ns response time in Nat. Photon. 17, 171–178, 2023], and thus experimental verification was lacking. Therefore, observing a test result as high as 400 GHz in such experiments is truly remarkable.

Additionally, I have evaluated the device characterization and performance:

- **Device Characterization.** High-frequency testing undoubtedly requires exceptional skill. As far as I know, testing beyond 100 GHz involves many factors, and only very few research groups worldwide can conduct tests up to 500 GHz, with Prof. Leuthold's group being one of them. After carefully reading the paper, the peer review comments, and the authors' responses, I believe the test results in this paper are credible. This is not only because the authors are absolute experts in high-frequency photoelectric testing globally but also because they provided sufficient technical details in their responses, including comprehensive descriptions of their testing setup and structural design.
- **Device Performance.** Although the device performance reported in this paper is modest, this is likely due to the lower quality of CVD-grown graphene. As far as I know, similar designs using

mechanically exfoliated high-quality graphene can easily achieve over 30 mA/W, and this result has been independently repeated by multiple research groups. Therefore, the authors' predicted response of 100 mA/W should be achievable with optimization. Furthermore, the performance difference between CVD graphene and mechanically exfoliated graphene devices is indeed an interesting question, but I do not think it is the primary focus of this paper. Additionally, the devices reported in this work operate at zero bias, resulting in very low intrinsic noise. However, many existing circuit designs are tailored for traditional biased devices, leading to a lower signal-to-noise ratio during testing. With more targeted circuit designs in the future, I believe the testing performance will improve.

Reviewer #3 (Remarks to the Author):

The authors have addressed all my concerns in the last round review.